# Three-dimensional structure of kinetochore-fibers in human mitotic spindles

Robert Kiewisz[1]*[†, ‡], Gunar Fabig[1], William Conway[2], Daniel Baum[3], Daniel Needleman[2,4,5,6], Thomas Müller-Reichert[1]*

[1]Experimental Center, Faculty of Medicine Carl Gustav Carus, Technische Universität Dresden, Dresden, Germany; [2]Department of Physics, Harvard University, Cambridge, United States; [3]Department of Visual and Data-Centric Computing, Zuse Institute Berlin, Berlin, Germany; [4]Department of Molecular and Cellular Biology, Harvard University, Cambridge, United States; [5]John A. Paulson School of Engineering and Applied Sciences, Harvard University, Cambridge, United States; [6]Center for Computational Biology, Flatiron Institute, New York, United States

**\*For correspondence:**
robert.kiewisz@gmail.com (RK);
mueller-reichert@tu-dresden.de (TM-R)

**Present address:** [†]Biocomputing Unit, Centro Nacional de Biotecnologia (CNB-CSIC), Darwin, 3, Campus Universidad Autonoma, 28049 Cantoblanco, Madrid, Spain; [‡]Simons Machine Learning Center, New York Structural Biology Center, New York, United States

**Competing interest:** The authors declare that no competing interests exist.

**Abstract** During cell division, kinetochore microtubules (KMTs) provide a physical linkage between the chromosomes and the rest of the spindle. KMTs in mammalian cells are organized into bundles, so-called kinetochore-fibers (k-fibers), but the ultrastructure of these fibers is currently not well characterized. Here, we show by large-scale electron tomography that each k-fiber in HeLa cells in metaphase is composed of approximately nine KMTs, only half of which reach the spindle pole. Our comprehensive reconstructions allowed us to analyze the three-dimensional (3D) morphology of k-fibers and their surrounding MTs in detail. We found that k-fibers exhibit remarkable variation in circumference and KMT density along their length, with the pole-proximal side showing a broadening. Extending our structural analysis then to other MTs in the spindle, we further observed that the association of KMTs with non-KMTs predominantly occurs in the spindle pole regions. Our 3D reconstructions have implications for KMT growth and k-fiber self-organization models as covered in a parallel publication applying complementary live-cell imaging in combination with biophysical modeling (Conway et al., 2022). Finally, we also introduce a new visualization tool allowing an interactive display of our 3D spindle data that will serve as a resource for further structural studies on mitosis in human cells.

## Editor's evaluation

This paper will be an incredible resource for cell biologists. The authors use sophisticated reconstructions of kinetochore–fibers within human metaphase spindles using electron tomography and then analyze their ultrastructure and organization. The findings lead to compelling models with clear implications for kinetochore–fiber and spindle self–organization.

## Introduction

Chromosome segregation during cell division is carried out by microtubule (MT)-based spindles (*Anjur-Dietrich et al., 2021*; *McIntosh et al., 2013*; *Oriola et al., 2018*; *Prosser and Pelletier, 2017*). While mitotic spindles can contain thousands of MTs, only a fraction of those highly dynamic filaments is associated with the kinetochores (*Redemann et al., 2017*). These MTs are called kinetochore microtubules (KMTs) and function to establish a physical connection between the chromosomes and the rest

of the spindle (*Flemming, 1879*; *Khodjakov et al., 1997*; *Maiato et al., 2004*; *Musacchio and Desai, 2017*; *Rieder, 1981*; *Rieder and Salmon, 1998*).

The regulation of KMT dynamics in mitotic spindles has been studied in great detail in a number of different systems, including the early *Caenorhabditis elegans* embryo, *Xenopus* egg extracts and mammalian tissue culture cells (*DeLuca et al., 2006*; *Dumont and Mitchison, 2009*; *Farhadifar et al., 2020*; *Inoué and Salmon, 1995*; *Kuhn and Dumont, 2019*; *Long et al., 2020*). However, our understanding of the ultrastructure of KMTs in mammalian k-fibers is rather limited due to a low number of three-dimensional (3D) studies on spindle organization. Earlier 3D studies on mammalian spindles applied several techniques. Some studies used serial thin-section transmission electron microscopy (TEM) (*Khodjakov et al., 1997*; *Mastronarde et al., 1993*; *McDonald et al., 1992*; *McEwen and Marko, 1998*; *Sikirzhytski et al., 2014*) or partial 3D reconstruction by electron tomography (*O'Toole et al., 2020*; *Yu et al., 2019*). Other studies used scanning electron microscopy to analyze the ultrastructure of mitotic spindles (*Hoffman et al., 2020*; *Nixon et al., 2017*; *Nixon et al., 2015*). However, these prior studies did not present comprehensive 3D reconstructions of mammalian mitotic spindles. Nevertheless, by applying serial thin-section TEM it was reported that k-fibers in PtK1 cells are composed of about 20 KMTs (*McDonald et al., 1992*; *McEwen et al., 1997*). In contrast, tomographic analysis of RPE1 cells revealed 12.6 ± 1.7 KMTs per k-fiber (*O'Toole et al., 2020*). Moreover, different cell types can exhibit a wide range of chromosome sizes, which could be an important factor in modulating the number of attached KMTs (*Moens, 1979*). This variation in the reported numbers of KMTs per k-fiber as well as a lack of complete 3D models of human mitotic spindles motivated us to perform an in-depth analysis of the k-fiber organization and KMT length distribution in the context of whole mitotic spindles in human tissue culture cells.

It was shown that mitotic KMTs exhibited various patterns of organization in different species. Single KMTs are connected to the kinetochores in budding yeast (*Winey et al., 1995*), while multiple KMTs are connected to dispersed kinetochores in nematodes (*Oegema et al., 2001*; *O'Toole et al., 2003*; *Redemann et al., 2017*). Multiple KMTs connected to kinetochores are also observed in human cells. However, KMTs in these cells are organized into bundles, termed 'kinetochore-fibers' (k-fibers), which are attached to a single region on each chromosome (*Begley et al., 2021*; *Godek et al., 2015*; *Inoue, 1953*; *Metzner, 1894*; *Mitchison and Kirschner, 1984*; *O'Toole et al., 2020*).

Three different simplified models of k-fiber organization can be drawn. Firstly, a direct connection between kinetochores and spindle poles can be considered (*Figure 1A*), in which all KMTs in a given k-fiber have approximately the same length and are rigidly connected (*Ris and Witt, 1981*). Secondly, an indirect connection may be considered (*Figure 1B*), In such a model, none of the KMT minus ends would be directly associated with the spindle poles, thus KMTs would show differences in their length and connect to the poles purely by interactions with non-KMTs in the spindle. Such an indirect connection was previously reported for a subset of k-fibers in PtK1 and PtK2 cells (*Sikirzhytski et al., 2014*). Thirdly, the kinetochore-to-spindle pole connection may be neither direct nor indirect, thus

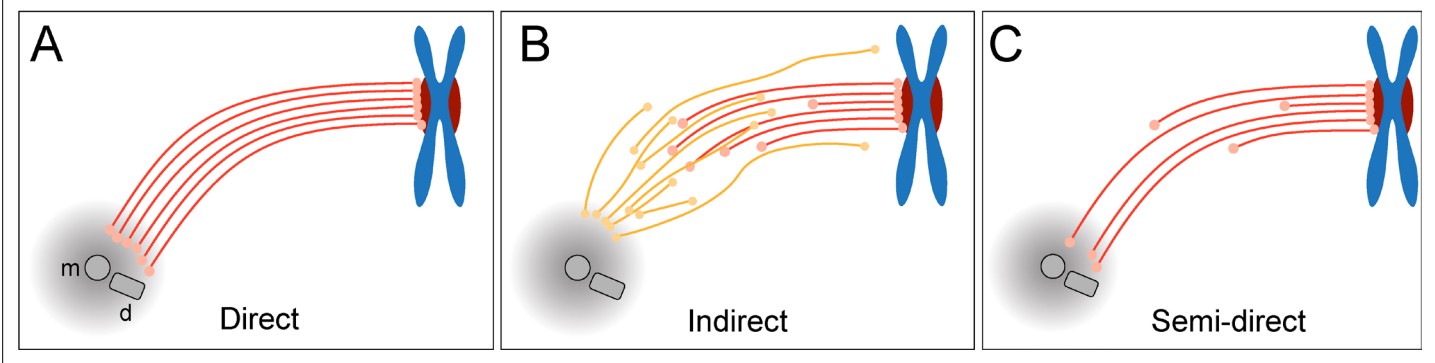

**Figure 1.** Models of k-fiber organization in mammalian mitosis. (**A**) Direct connection with KMTs (red lines) spanning the distance between the kinetochore and the spindle pole. Chromosomes are shown in blue with kinetochores in red. The mother (m) and the daughter centriole (d) of the spindle pole are indicated. All KMTs are assumed to have similar lengths. (**B**) Indirect connection showing KMTs linking the kinetochore and the spindle pole by association with non-KMTs (yellow lines). K-fibers in this model are composed of KMTs with different lengths, and none of the KMTs is directly associated with the spindle pole. (**C**) Semi-direct connection showing KMTs of different lengths. Some KMTs are directly associated with the spindle pole, while others are not. In this model, KMTs show a difference in length.

showing a semi-direct pattern of connection, in which only some of the KMTs in a given k-fiber are associated with the spindle pole while others are not (*Figure 1C*). Previously, we have shown such a semi-direct pattern of KMT anchoring into the spindle network for the first embryonic mitosis in *C. elegans* (*Redemann et al., 2017*). Some KMTs in this nematode system are indeed directly associated with the spindle poles, while others are not. As far as the length of the KMTs in mammalian cells is concerned, a difference in their length had previously been reported for PtK1 cells (*McDonald et al., 1992*; *Sikirzhytski et al., 2014*). We, therefore, wondered how the anchoring of k-fibers into the spindle network is achieved in mammalian cells.

Here, we aimed to determine the number and length of KMTs and the positioning of their putative minus ends in human HeLa cells. We further aimed to analyze the organization of k-fibers and the interaction of KMTs with non-KMTs in whole mammalian spindles. Focusing on the metaphase stage, we applied serial-section electron tomography to produce large-scale reconstructions of entire mitotic spindles in HeLa cells. To achieve this, we developed software tools for a quantitative in-depth analysis of both KMTs and non-KMTs (*Kiewisz and Müller-Reichert, 2021*; https://github.com/RRobert92/ASGA). We found that k-fibers in HeLa cells display a previously unexpected variable morphology. The k-fibers indeed contain KMTs of different lengths (a semi-direct type of connection with the spindle pole) and show an uncoupling of KMT minus ends at the site of preferred interaction with the spindle poles. For better visualization of KMT organization and k-fiber morphology, we introduce here a new 3D visualization tool that allows the interested reader to interactively display the 3D data (https://cfci.shinyapps.io/ASGA_3DViewer/).

## Results

### K-fibers are composed of approximately nine KMTs

For our large-scale analysis of mammalian k-fibers, we acquired data on metaphase spindles in HeLa cells by serial-section electron tomography (*Figure 2A–B*). To visually inspect the quality of our samples, we extracted slices of regions of interest (*Figure 2—figure supplement 1*). We also used the tomogram data to reconstruct full spindles in 3D for quantitative analysis of the spindle morphology (*Figure 2—videos 1–3*). In preparation for this quantitative analysis, we applied a Z-factor to our 3D models to correct for a sample collapse that had occurred during the acquisition of the tomographic data (*Figure 2—figure supplement 2*). In our three full reconstructions, we segmented all MTs, the chromosomes and the spindle poles (including the centrioles). Each of these metaphase spindles was composed of approximately 6300 MTs (6278 ± 1614 MTs, mean ±STD; *Figure 2C–E*; *Table 1*, *Table 2*) and had an average pole-to-pole distance of 9.0 ± 1.7 µm (mean ±STD; *Figure 2—figure supplement 3A-B*; *Table 1*).

We then annotated the KMTs in our reconstructions based on the association of the putative MT plus ends with kinetochores. MTs that were arranged in parallel and made end-on contact at a single 'spot' on the chromosomes were defined as KMTs being part of the same k-fiber. For this publication, these bundled KMTs were considered the 'core' of the k-fibers. Possible interactions of these KMTs with other MTs (referred to as non-KMTs) in the spindle were subject to subsequent steps of our in-depth spindle analysis. In our tomographic data sets, we identified between 90 and 110 k-fibers per cell, which included on average 859 ± 218 KMTs (mean ±STD, n=3; *Figure 2F–H*; *Figure 2—videos 4–6*; *Table 1*) in each spindle. Thus, only ~14% of all MTs in the reconstructed spindles were KMTs. The majority of annotated KMTs displayed open flared ends at the kinetochore (*Figure 2—figure supplement 4*), consistent with previous observations on the morphology of KMT plus ends in mammalian cells (*McIntosh et al., 2013*). We took advantage of these extracted k-fibers to further analyze the distance between the sister k-fiber ends in each data set. For this, we calculated the median position of the KMT plus ends at each k-fiber and then determined the distance between the median KMT plus-end positions of sister k-fibers (*Figure 2—figure supplement 3C-D*; *Table 1*). The average distance between the sister k-fiber ends was 1.13 ± 0.24 µm (mean ±STD, n=292). The similarity in the median distance between sister k-fiber ends in the three reconstructions indicated to us that the selected pre-inspected spindles were indeed cryo-immobilized at a similar mitotic stage, thus allowing a further comparative quantitative analysis of our 3D models.

Next, we extracted individual k-fibers from our full 3D reconstructions to visualize their overall morphology (*Figure 3A*; *Figure 3—videos 1–6*). Our serial-section approach enabled us to follow

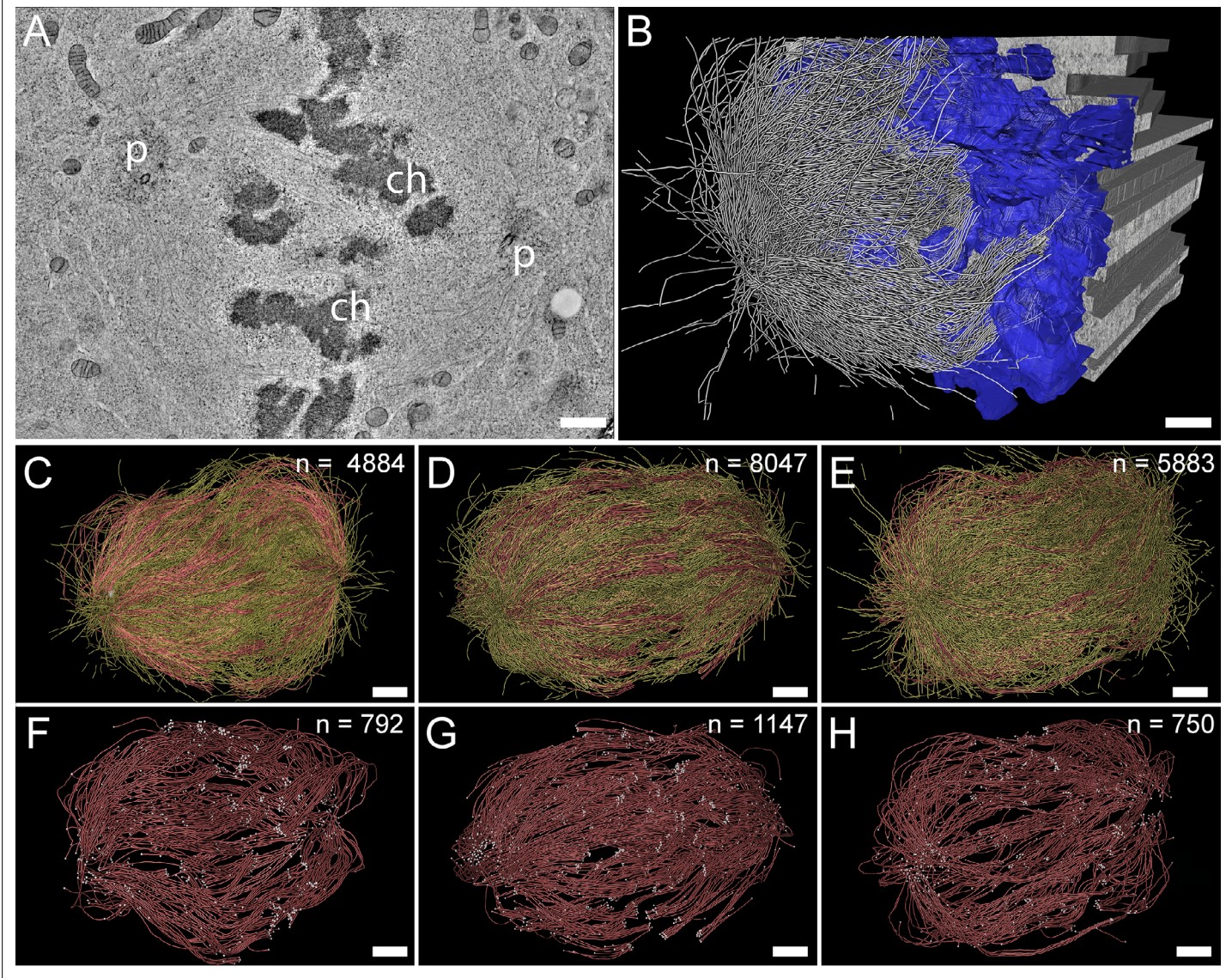

**Figure 2.** Three-dimensional reconstruction of metaphase spindles by large-scale electron tomography. (**A**) Tomographic slice showing a HeLa cell (spindle #1) in metaphase. The chromosomes (ch) and the spindle poles (p) are indicated. (**B**) Three-dimensional reconstruction of the same spindle as shown in A. The stacking of the serial tomograms used to generate a three-dimensional model of the spindle with the MTs (white lines) is visualized. The segmented chromosomes are shown in blue. (**C**) Three-dimensional model of the spindle as shown in A. The total number of all MTs is given in the upper right corner. The non-KMTs (yellow lines) and KMTs (red lines) are shown. (**D**) Full 3D model of metaphase spindle #2. (**E**) Full 3D model of metaphase spindle #3. (**F**) Extraction of KMTs from the 3D reconstruction as shown in C. The number of KMTs is given in the upper right corner. KMT plus and minus ends are shown by white spheres. (**G**) KMTs extracted from spindle #2. (**H**) KMTs extracted from spindle #3. Scale bars, 1 µm.

The online version of this article includes the following video and figure supplement(s) for figure 2:

**Figure supplement 1.** Illustration of metaphase in HeLa cells.

**Figure supplement 2.** Correction of sample collapse caused during data acquisition by electron tomography.

**Figure supplement 3.** Analysis of pole-to-pole and sister k-fiber-to-sister k-fiber distances.

**Figure supplement 4.** Morphology of KMT plus ends.

**Figure 2—video 1.** Generation of a 3D model from joined serial electron tomograms displaying spindle #1.
https://elifesciences.org/articles/75459/figures#fig2video1

**Figure 2—video 2.** Generation of a 3D model from joined serial electron tomograms displaying spindle #2.
https://elifesciences.org/articles/75459/figures#fig2video2

**Figure 2—video 3.** Generation of a 3D model from joined serial electron tomograms displaying spindle #3.

*Figure 2 continued on next page*

*Figure 2 continued*

https://elifesciences.org/articles/75459/figures#fig2video3

**Figure 2—video 4.** Organization of KMTs in spindle #1.

https://elifesciences.org/articles/75459/figures#fig2video4

**Figure 2—video 5.** Organization of KMTs in spindle #2.

https://elifesciences.org/articles/75459/figures#fig2video5

**Figure 2—video 6.** Organization of KMTs in spindle #3.

https://elifesciences.org/articles/75459/figures#fig2video6

each KMT in each k-fiber in 3D. This was achieved by semi-automatic stitching of the corresponding ends over section borders (*Figure 3—figure supplement 1*; *Lindow et al., 2021*). In addition to this semi-automatic stitching, each KMT in our reconstructions was manually checked for a proper end identification. The individual k-fibers showed remarkable variability in their overall shape. Some k-fibers were rather straight, while others were very curved. At the kinetochores, k-fibers showed a compacted appearance, while k-fibers were considerably broader at their pole-proximal end. Interestingly, some KMT minus ends extended beyond the position of the centrioles (*Figure 3A*, k-fibers #I - #III).

We further investigated the number of KMTs associated per kinetochore (*Figure 3B*; *Figure 3—figure supplement 2A*; *Table 3*) and found that the k-fibers were composed of around nine KMTs (8.5 ± 2.2, mean ±STD, n=292). To exclude the possibility that the average number of KMTs attached to kinetochores is influenced by a possible stretch of the sister kinetochores, we plotted both the number of attached KMTs and the difference (delta) in the number of KMTs associated with the respective sister kinetochore against the distance between the kinetochore-proximal ends of k-fiber pairs. We did not observe a correlation between these parameters (*Figure 3C–D*; Pearson's correlation coefficients were 0.04 and 0.29) and concluded that the number of KMT attachments to kinetochores in metaphase is not influenced by a variation in the inter-kinetochore distance. Another variable with a possible influence on the number of attached KMTs to the outer kinetochores could be the position of the k-fibers within the metaphase spindle. Because spindles show a rounded appearance at metaphase, a difference in the number of attached MTs to the outer kinetochores could be influenced by the overall spindle shape. To analyze such a possible positional effect, we considered the cross-section of the metaphase plate as an ellipse and defined a central, an intermediate and a peripheral zone on this ellipse (*Figure 3—figure supplement 3A*). By determining the position of the kinetochores on the 3D-reconstructed metaphase plate, we then annotated each k-fiber in our three data sets to one of these regions (*Figure 3—figure supplement 3B-M*). Keeping the roundedness of spindles at metaphase in mind, we indeed observed that k-fibers positioned in the center are rather straight, while peripheral k-fibers are more curved. However, we did not find a difference in the number of attached KMTs for these three different regions (*Figure 3—figure supplement 3N*; Table 5) and concluded that also the position of the k-fibers within the spindle has no effect on the average number of KMTs per k-fiber.

We were also interested in measuring the density and spacing of KMTs at the kinetochore, thus allowing subsequent analysis of KMT density along the k-fiber length. Because kinetochores show lower contrast in high-pressure frozen material compared to conventionally prepared samples (*McEwen et al., 1998b*), we indirectly measured the size of the kinetochores in our spindles by determining the cross-sectional area of the k-fibers (i.e. by encircling the KMTs) close to the outer kinetochore plate. The measured average kinetochore area was $0.10 \pm 0.07 \ \mu m^2$ (mean ±STD; *Figure 3—figure supplement 2B-C*). We then analyzed the density of KMTs at the outer kinetochores by counting the number of KMTs within the determined areas, which was $112 \pm 60$ KMTs/$\mu m^2$ (mean ±STD, n=292; *Figure 3—figure supplement 2D*; *Table 3*). In addition, we observed an average center-to-center distance between neighboring KMTs of 74 ± 22 nm (mean ±STD, n=292; *Figure 3—figure supplement 2E*; *Table 4*). Considering an MT diameter of 25 nm, this corresponds to an average wall-to-wall spacing of about 50 nm between the KMTs at the outer kinetochore. Thus, following our initial visual inspection of k-fibers, the KMTs tend to be highly compacted at the outer kinetochore.

We also measured the length of the KMTs in our reconstructed k-fibers and observed a broad distribution of KMT lengths with an average value of 3.87 ± 1.98 μm (mean ±STD, n=2579; *Figure 4A*;

**Table 1.** Characterization of the 3D-reconstructed metaphase spindles in HeLa cells.

| Data set | Spindle pole distance [μm] | Inter-kinetochore distance [μm]* | No. of MTs in the tomographic volume | No. of kinetochores | No. of KMTs | No. of non-KMTs | No. of k-fibers |
|---|---|---|---|---|---|---|---|
| Spindle #1 | 7.16 | 1.08 ± 0.20 (n=43) | 4884 | 92 | 797 (16.3%) | 4087 (83.7%) | 92 |
| Spindle #2 | 10.39 | 1.24 ± 0.21 (n=50) | 8047 | 110 | 1,102 (13.7%) | 6945 (86.3%) | 110 |
| Spindle #3 | 9.48 | 1.03 ± 0.27 (n=40) | 5904 | 90 | 680 (11.5%) | 5224 (88.5%) | 90 |

*Numbers are given as mean ± STD.

**Table 2.** Tomographic data sets as used throughout this study.

| Data set | Original data set | Montage (X/Y) | No. of serial sections [300 nm each] | Estimated tomographic volume [μm³] | Data set size [Gb] |
|---|---|---|---|---|---|
| Spindle #1 | T_0475 | 2 × 3 | 22 | 598 | 46.5 |
| Spindle #2 | T_0479 | 2 × 3 | 29 | 996 | 77.9 |
| Spindle #3 | T_0494 | 2 × 3 | 35 | 904 | 71.9 |

*Figure 4—figure supplement 1A*; *Table 3*). Our analysis revealed the existence of relatively short KMTs in central, intermediate and peripheral k-fibers that were not associated with the spindle poles (*Figure 4—figure supplements 2–3*; *Table 5*). Indeed, about 20 ± 4% of the KMTs had lengths less than 2 μm. Our analysis also showed relatively long KMTs (about 39 ± 10%) that were longer than the half spindle length. Some of these long KMTs showed a pronounced curvature at their pole proximal end, thus connecting to the 'back side' of the spindle poles (see also *Figure 3A*, k-fiber #I - #III; *Figure 3—videos 1–3*).

We continued our study by further analyzing the pole proximal ends (from now on called minus ends). As a first step, we annotated each KMT minus end in our spindle reconstructions. The development of appropriate software allowed us then to determine both the distance of the KMT minus ends to the nearest spindle pole and the relative position of the KMT minus ends along the pole-to-kinetochore axis (*Figure 5A*; *Kiewisz and Müller-Reichert, 2021*). In addition, we were also interested in the percentage of the KMT minus ends that were directly associated with the spindle poles. Similar to our previously published analysis of spindle morphology in the early *C. elegans* embryo (*Redemann et al., 2017*), we defined a MT-centrosome interaction area. For this, we plotted the distribution of all non-KMT minus-end distances to the nearest spindle pole. The distribution peaked ~1 μm from the pole and then fell before plateauing in the spindle bulk. To find the edge of this MT-centrosome interaction area, we fit a Gaussian to the distribution peak and defined the cutoff distance for the edge of the MT-centrosome interaction area as twice the half-width, which was 1.7 μm from the mother centriole. (*Figure 5B*, gray area). In other words, KMTs with their minus ends positioned at 1.7 μm or less to the center of the nearest mother centriole (i.e. inside this MT-centrosome interaction zone) were defined to be directly associated with a pole, while KMT minus ends positioned farther than this cut-off distance of 1.7 μm were called indirectly associated with the spindle pole. We then measured the distance of each KMT minus end to the nearest mother centriole (*Figure 5C*; *Figure 5—figure supplement 2A*). Taking our determined cut-off value into account, we found that only 49% (±15.5%, ±STD, n=3) of the KMT minus ends were positioned within the defined MT-centrosome interaction area. This is in accord with our observation that the average number of KMTs per k-fiber at the spindle pole (4.1 ± 2.0, mean ±STD; *Figure 5—figure supplement 3*; *Table 3*) was lower compared to the average number of KMTs per k-fiber at the kinetochore (8.5 ± 2.2, mean ±STD; *Figure 3B*). All in all, this suggested to us that only half of the KMTs in HeLa cells are directly connected to the spindle pole, while the other half of the KMTs are indirectly connected.

Interestingly, we also observed that the number of KMT minus ends associated with the spindle poles was significantly higher in k-fibers positioned in the center compared to those at the periphery of the mitotic spindle. In addition, the average length of KMTs in central k-fibers and their minus-end distance to the spindle pole were significantly lower compared to those observed in peripherally positioned k-fibers (*Figure 5—figure supplement 4*; *Table 5*). This suggested to us that the position of the k-fibers within the spindle affects the ultrastructure of the individual KMTs.

We next investigated the relative position of the KMT minus ends on the pole-to-kinetochore axis. For this, we defined the approximate relative position of the MT-centrosome interaction area on the pole-to-kinetochore axis (*Figure 5D*; *Figure 5—figure supplement 1*; *Figure 5—figure supplement 2B*; *Table 3*). The approximated relative position was calculated as an average for all KMTs and ranged from –0.2 to 0.2. We found that the KMT minus ends that were positioned within the MT-centrosome interaction zone showed a peak position close to the center of the spindle poles. In contrast, KMT minus ends outside this interaction area did not show a preferred position but rather displayed a flat relative distribution on the pole-to-kinetochore axis. This analysis confirmed our initial visual 3D

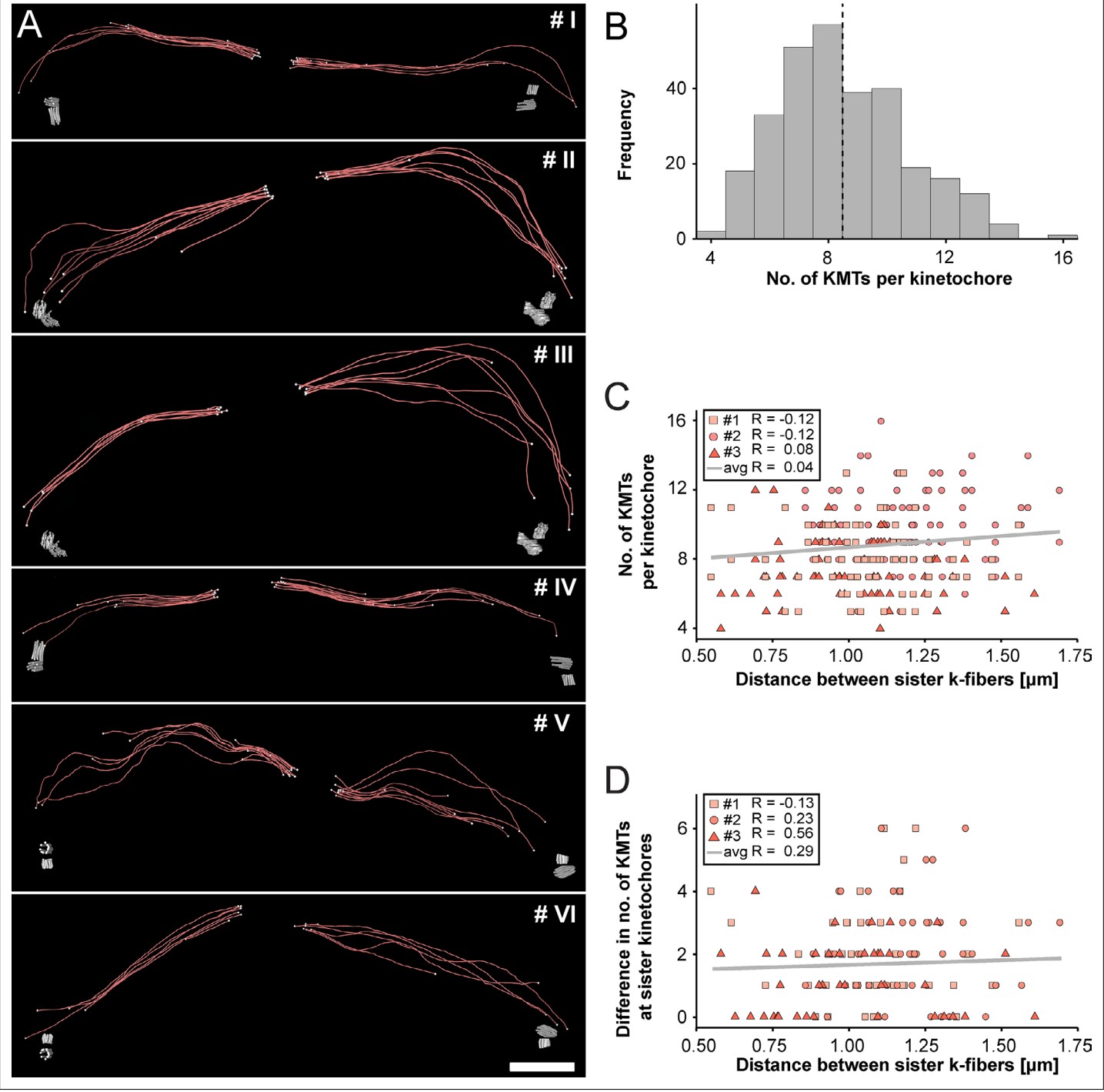

**Figure 3.** Morphology of k-fibers and number of KMTs associated per kinetochore. (**A**) Examples of individual sister k-fibers extracted from the full 3D reconstruction of metaphase spindle #1. The numbering of these examples (corresponding to the supplementary videos) is given in the upper right corners. KMTs are shown as red lines. The ends of the KMTs are indicated by white spheres, centrioles are shown as cylinders (gray). Scale bar for all examples, 1.5 μm. (**B**) Histogram showing the frequency of detected KMTs per kinetochore. This plot includes data from all three spindle reconstructions. The dashed line (black) indicates the average number of KMTs per kinetochore (n=292). (**C**) Graph showing the number of KMTs associated per kinetochore plotted against the distance between sister k-fibers (n=292). The Pearson's correlation coefficient for each data set and the average coefficient for all data sets are given. (**D**) Graph showing the difference (delta) in the number of KMTs associated with the respective sister kinetochores plotted against the distance between the kinetochore-proximal ends of k-fiber pairs (n=292). The Pearson's correlation coefficient for each data set and the average coefficient for all data sets are given.

The online version of this article includes the following video and figure supplement(s) for figure 3:

*Figure 3 continued on next page*

*Figure 3 continued*

**Figure supplement 1.** MT segmentation and stitching across consecutive serial sections.

**Figure supplement 2.** Correlation of k-fiber circumference and number of attached KMTs.

**Figure supplement 3.** Positioning of k-fibers in the mitotic spindle.

**Figure 3—video 1.** 3D reconstruction of a single k-fiber pair.
https://elifesciences.org/articles/75459/figures#fig3video1

**Figure 3—video 2.** 3D reconstruction of a single k-fiber pair.
https://elifesciences.org/articles/75459/figures#fig3video2

**Figure 3—video 3.** 3D reconstruction of a single k-fiber pair.
https://elifesciences.org/articles/75459/figures#fig3video3

**Figure 3—video 4.** 3D reconstruction of a single k-fiber pair.
https://elifesciences.org/articles/75459/figures#fig3video4

**Figure 3—video 5.** 3D reconstruction of a single k-fiber pair.
https://elifesciences.org/articles/75459/figures#fig3video5

**Figure 3—video 6.** 3D reconstruction of a single k-fiber pair.
https://elifesciences.org/articles/75459/figures#fig3video6

inspection of the KMTs, revealing that the k-fibers in HeLa cells are not composed of compact bundles of KMTs of the same length but rather show KMTs of different lengths, thus confirming previously published data (*McDonald et al., 1992*; *O'Toole et al., 2020*).

For comparison, we also analyzed the length distribution of non-KMTs in the spindles. Non-KMTs had an average length of 2.0 ± 1.7 µm (mean ±STD; n=14458; *Figure 4B*; *Figure 4—figure supplement 1B*) showing a high number of very short (<2 µm) and a low number of long MTs (>half spindle length). In addition, 38 ± 9% of the non-KMT minus ends were localized in the defined MT-centrosome interaction zone and the remaining ~60% were located in the bulk of the spindle (*Figure 5E*; *Figure 2—figure supplement 2C*; *Table 3*). In addition, the distribution plot of the relative position of the non-KMT minus ends on the pole-to-pole axis showed two peaks at the spindle poles (*Figure 5F*; *Figure 5—figure supplement 2D*). Overall, this indicated to us that the non-KMTs show a very high number of very short MTs that is different from the flatter length distribution of KMTs.

## KMT tortuosity is higher at the spindle poles than at the kinetochores

Previous work on the flexibility and the rigidity of MTs indicated that these polymers are able to search the spindle space for a binding partner, bend and continue to grow in a modified direction to avoid obstacles or react to pushing/pulling forces. It was further shown that the flexibility of MTs is dependent on their length (*Pampaloni et al., 2006*). Therefore, we were interested in whether long KMTs are more curved compared to short KMTs. As a measure, we decided to analyze the tortuosity of individual KMTs in our 3D models. Tortuosity is the ratio of the total length of a curve (the spline length of a given KMT) to the distance between its ends. Straight KMTs, therefore, have a tortuosity of 1, while a quarter circle has a tortuosity of around ~1.1 and a half-circle of around ~1.6 (*Figure 6A*). Because the tortuosity of KMTs might not be homogeneous throughout the spindle, we aimed to

**Table 3.** Quantitative analysis of KMTs and non-KMTs.

| Data set | Length of KMTs [µm]* | Length of non-KMTs [µm]* | No. of KMTs per kinetochore* | No. of KMTs in the MT-centrosome interaction area* | Mean KMT minus-end distance to poles [µm] | % of KMTs associated with poles | % of non-KMTs associated with poles |
|---|---|---|---|---|---|---|---|
| Spindle #1 | 3.59 (±1.57) | 2.13 (±1.67) | 8.04 (±1.86) | 5.0 (±1.8) | 1.72 | 61.2 | 44.3 |
| Spindle #2 | 3.82 (±1.97) | 1.95 (±1.60) | 9.75 (±2.18) | 3.1 (±2.3) | 2.87 | 31.5 | 28.6 |
| Spindle #3 | 4.27 (±1.93) | 2.07 (±1.93) | 7.49 (±1.91) | 4.1 (±2.0) | 2.12 | 54.2 | 41.9 |

*Numbers are given as mean ±STD.

**Table 4.** Quantitative analysis of k-fiber organization.

| Data set | KMT density at the kinetochore [KMT/µm²]* | KMT-KMT distance at the kinetochore [nm]* | Global tortuosity of KMTs* | % of curved KMTs | Area of k-fibers [µm²]* | % of KMTs in a k-fibers* |
|---|---|---|---|---|---|---|
| Spindle #1 | 122 (±62) | 67 (±20) | 1.11 (±0.11) | 39.8 | 0.08 (±0.1) | 64 (±27) |
| Spindle #2 | 99 (±45) | 78 (±23) | 1.07 (±0.07) | 28.4 | 0.09 (±0.11) | 70 (±25) |
| Spindle #3 | 117 (±72) | 76 (±23) | 1.13 (±0.13) | 47.1 | 0.12 (±0.24) | 59 (±29) |

*Numbers are given as mean ±STD.

measure both their global and local tortuosity in our 3D reconstructions, that is, the tortuosity of the KMTs along their entire length and also in defined segments of a length of 500 nm along the k-fibers, respectively (*Figure 6B–C*).

Firstly, we analyzed the global tortuosity of the KMTs. For this, we applied a color code to our 3D models to visualize differences in the curvature of individual KMTs (*Figure 6D*; *Figure 6—videos 1–3*). For all data sets, we observed an average value of KMT tortuosity of 1.1 ± 0.1 (mean ±STD, n=2579). We found that 62 ± 8% of the KMTs showed a tortuosity of lower than 1.1 and 38 ± 10% of the KMTs displayed a tortuosity higher than 1.1 (*Figure 6E*; *Table 4*). We also observed that straight KMTs (tortuosity <1.1) were predominantly located in the center of the spindle, while curved KMTs (tortuosity >1.1) were located more at peripheral spindle positions (*Figure 6F*; *Figure 6—figure supplement 1*, *Table 5*). Furthermore, the global tortuosity of KMTs was correlated with their length. As expected, short KMTs were straighter, while long KMTs were more curved ($R = 0.68$; $p = 2.2e{-}16$; *Figure 6G*). In addition, 75 ± 6% of the KMTs with a tortuosity higher than 1.1 were longer than the half-spindle length. Secondly, we also investigated the local tortuosity of the KMTs. For each KMT, we applied the same color code as used for the analysis of global tortuosity (*Figure 6H*). Then we plotted the tortuosity value for each 500 nm segment against the position on the pole-to-kinetochore axis (*Figure 5—figure supplement 1*). Our analysis revealed that the tortuosity of KMTs was not uniform

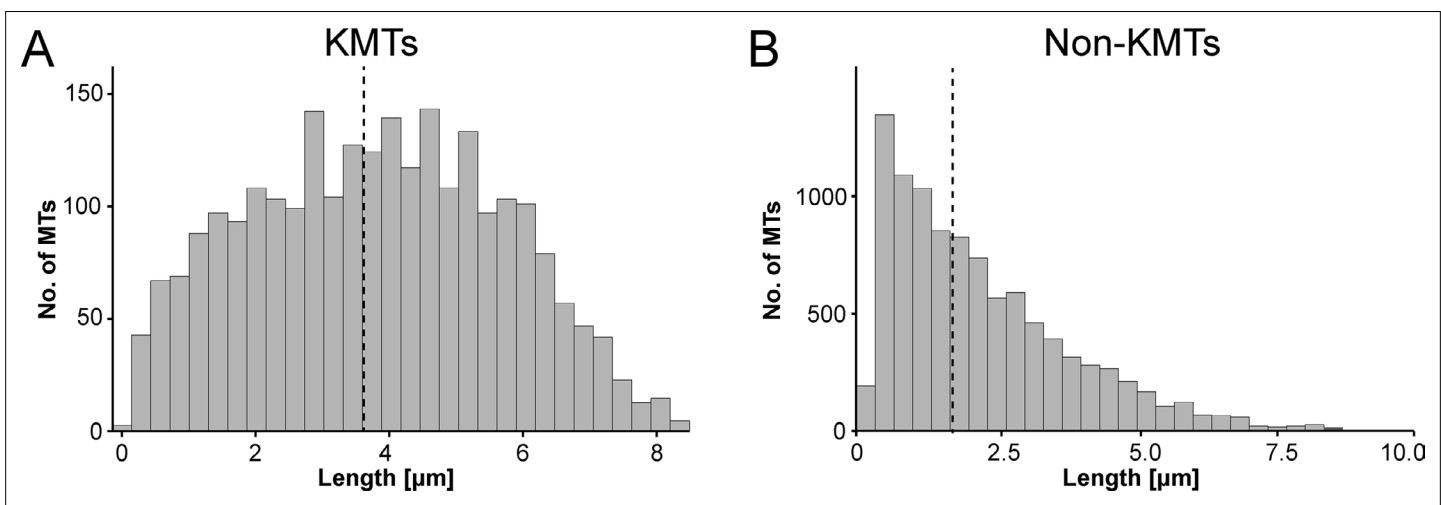

**Figure 4.** Analysis of MT length distribution. (**A**) Histogram showing the length distribution of KMTs from all data sets (n=2579). The dashed line indicates the average length of KMTs. (**B**) Histogram showing the length distribution of non-KMTs (n=14458). The dashed line indicates the average length of non-KMTs.

The online version of this article includes the following figure supplement(s) for figure 4:

**Figure supplement 1.** Length distribution of KMTs and non-KMTs.

**Figure supplement 2.** KMT length distribution based on the k-fiber position in the spindle.

**Figure supplement 3.** KMT length distribution based on the position of individual k-fibers within the spindle.

**Table 5.** Quantitative analysis of k-fiber positioning in the spindle.

| Region | Length of KMTs [μm]* | No. of KMTs per kinetochore* | No. of KMTs at MT-centrosome interaction area† | Mean KMT minus-end distance to poles [μm]* | No. of KMTs associated with poles* | Global tortuosity of KMTs* |
|---|---|---|---|---|---|---|
| Central | 3.5 (±1.7) | 8.2 (±2.4) | 162 (~48%) | 2.0 (±1.3) | 4.3 (±2.3) | 1.08 (±0.08) |
| Intermediate | 3.6 (±1.7) | 8.6 (±2.1) | 266 (~49%) | 2.1 (±1.3) | 4.6 (±1.9) | 1.11 (±0.12) |
| Peripheral | 3.9 (±2.0) | 8.6 (±2.4) | 730 (~45%) | 2.5 (±1.6) | 4.1 (±2.0) | 1.10 (±0.10) |

*Numbers are given as mean ±STD.
†Number and percentage of KMTs is shown.

along the pole-to-kinetochore axis. Importantly, the local tortuosity of the KMTs was weakly correlated with the relative position of the KMT segments on the pole-to-kinetochore axis. The local tortuosity slowly and constantly increased from the kinetochores towards the spindle poles ($R = –0.13$; $p = 2.2e-16$; *Figure 6I*). Extending previously published knowledge, we concluded that KMTs have a higher tortuosity at the spindle poles compared to the kinetochores.

## K-fibers are broadened at spindle poles

Our tortuosity measurements revealed that individual KMTs in the mitotic spindle are rather curved at positions close to the spindle poles. Therefore, we were also interested in analyzing how the curvature of individual KMTs might shape the overall structure of the k-fibers, particularly at their pole-proximal ends. For this, we determined the cross-section areas of k-fibers along their entire length (*Figure 7A*; *Figure 7—figure supplement 1*). In the interest of precision, we analyzed the cross-sections of k-fibers by calculating polygonal areas, allowing a quantitative geometrical analysis without a prior assumption about their shape. Cross-sections of k-fibers showed an average polygonal area of 0.097 ± 0.161 μm² (mean ±STD, n=292). We then continued by plotting the values for these polygonal areas against the relative position on the pole-to-kinetochore axis (*Figure 7B*; *Table 4*). We measured an average polygonal area of 0.034 ± 0.019 μm² at the kinetochores, 0.149 ± 0.210 μm² in the middle of the spindles, and 0.092 ± 0.146 μm² at the spindle poles. Compared to the position at the kinetochore, the average polygonal area of the k-fibers was about fourfold higher in the middle of the spindles and roughly threefold higher at the spindle poles. Moreover, the cross-section polygonal area of the k-fibers showed a higher spread of values at the spindle poles compared to the kinetochores, thus reflecting the observed broadened appearance of the k-fibers at the spindle poles.

To further characterize the arrangement of the KMTs in the k-fibers, we also set out to measure the number of the KMTs along the length of the k-fibers (*Figure 7C*). For each k-fiber, we defined a circle enclosing all KMTs at the kinetochore. We then measured the number of KMTs that were included in this defined k-fiber circle and plotted the percentage of the enclosed KMTs against the relative position along the pole-to-kinetochore axis. We observed a variation in the percentage of enclosed KMTs along the k-fiber length. As defined, the highest percentage of enclosed KMTs was observed at the outer kinetochore. However, at the spindle poles, roughly only 64% of the KMTs were enclosed (*Table 4*). Thus, the density of KMTs in the k-fibers at the spindle poles was decreased compared to the one observed at the outer kinetochore (*Figure 7D*). From all these analyses, we concluded that k-fibers display a higher tortuosity and a lower KMT density close to the spindle poles compared to the kinetochore positions, thus leading to a broadened appearance of their pole-proximal ends.

## KMTs primarily associate with non-KMTs at spindle poles

So far, we had concentrated only on an analysis of KMT morphology and considered these bundled MTs as the 'core structure' of the k-fibers. Likely, the observed organization of KMTs in k-fibers is the result of KMTs interacting with other non-KMTs in the spindle, thus contributing to the maturing of k-fibers (*Almeida et al., 2021*). Therefore, we also aimed to investigate patterns of association of KMTs with the neighboring non-KMTs in our 3D reconstructions. Moreover, we were particularly

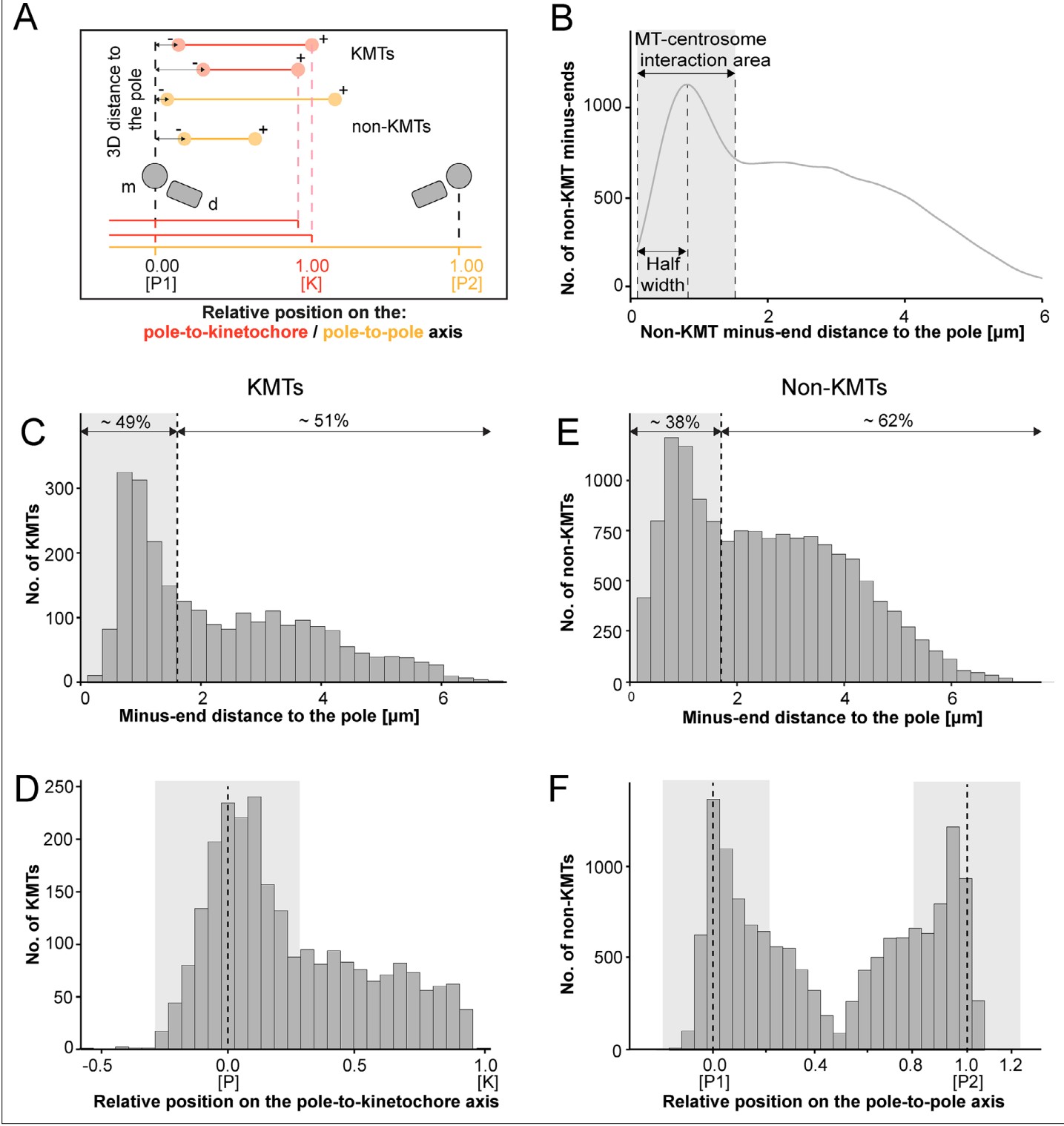

**Figure 5.** Analysis of MT minus ends. (**A**) Measurement of MT minus-end positioning. A KMT (red line) with its ends (red circles) and a non-KMT (yellow line) with its ends (yellow circles) are shown. The distance of both the KMT and the non-KMT minus ends to the center of the mother centriole was calculated. The relative position of the KMT minus ends along the pole-to-kinetochore axis and the non-KMT minus ends along the pole-to-pole axis was also determined (P1, pole 1; P2, pole 2; K, kinetochore). (**B**) Determination of the MT-centrosome interaction area. Graph showing the number of non-KMT minus ends plotted against their distance to the pole (i.e. to the center of the mother centriole). The determined area of the interaction of non-KMTs with the centrosome and the half-width of this area is indicated in gray. The border of the MT-centrosome interaction area (right dashed line) was determined by identifying twice the half-width of the distribution peak of the minus-end distances. (**C**) Histogram showing the distribution of the

*Figure 5 continued on next page*

*Figure 5 continued*

KMT minus-end distances to the center of the mother centriole (n=2579). The MT-centrosome interaction area as defined in B is indicated by a gray area (dashed line shows the border of this area). (**D**) Histogram showing the relative position of the KMT minus ends on the pole-to-kinetochore axis (n=2579). The position of the spindle pole (p = 0, dashed line) and the kinetochore (K = 1) is indicated. The approximated MT-centrosome interaction area is indicated in gray. (**E**) Histogram showing the distribution of the non-KMT minus-end distances to the center of the mother centriole (n=14458). The MT-centrosome interaction area is indicated in gray. (**F**) Plot showing the relative position of the non-KMT minus ends on the pole-to-pole axis (n=14458). The position of the spindle poles (P1 = 0, P2 = 1). The approximated MT-centrosome interaction area is shown in gray.

The online version of this article includes the following figure supplement(s) for figure 5:

**Figure supplement 1.** Normalization of minus-end positioning on the pole-to-kinetochore axis.

**Figure supplement 2.** Minus-end distribution of KMTs and non-KMTs.

**Figure supplement 3.** Analysis of KMT minus ends reaching the pole.

**Figure supplement 4.** KMT minus-end distribution based on the k-fiber position in the spindle.

interested in localizing such KMT/non-KMT associations in the spindles to map the detected positions of MT-MT interaction on the pole-to-kinetochore axis. In general, we considered two types of interactions between MTs. Firstly, we analyzed potential interactions between MT ends with neighboring MT lattices, which could be mediated by MT minus-end associated molecular motors such as dynein (*Tan et al., 2018*) or kinesin-14 (*Molodtsov et al., 2016*)**,** by other MT-associated proteins such as HDAC6 (*Ustinova et al., 2020*), Tau (*Bougé and Parmentier, 2016*), or by ɣ-tubulin (*Rosselló et al., 2018*). Secondly, we considered MT-MT lattice interactions, which might be established by molecular motors such as kinesin-5 (*Falnikar et al., 2011*) or PRC1 (*Mollinari et al., 2002*; *Polak et al., 2017*).

Both types of interactions are also shown here by using our new 3D visualization tool (*Kiewisz and Müller-Reichert, 2022*; https://cfci.shinyapps.io/ASGA_3DViewer). The aim of applying this tool is to enable an illustration of the 3D complexity of such KMT/non-KMT associations. Readers are encouraged to visit this website for a display of our spindles in 3D, thus allowing to view k-fibers, KMTs and also non-KMTs in an interactive way.

We started our analysis by investigating possible KMT minus-end associations with either KMT or non-KMT lattices (*Figure 8*). For this, we annotated all KMT minus ends in our 3D reconstructions and measured the distance of each minus end to a neighboring MT lattice. We then determined association distances (i.e. 25, 30, 35, 45, 50, 75, and 100 nm) to quantify the number of associations occurring within these given interaction distances (*Kellogg et al., 2016*; *Redwine et al., 2012*). From this, we further determined the percentage of all KMT minus ends that were associated with non-KMT lattices according to selected association distances (*Figure 8—figure supplement 1*; *Table 6* and *Table 7*). As expected, we observed that the number of KMT minus ends associated with adjacent MT lattices increased at larger association distances. Considering 35 nm as an example of a possible interaction distance between two MTs connected by a single dynein motor (*Amos, 1989*), we observed that only 32.6 ± 5.5% of all KMT minus ends were associated with other MTs (for a visualization of the pattern of association see *Figure 8—figure supplements 1–2*; *Figure 8—video 1*). Moreover, all KMT minus ends that were not associated with the spindle poles (i.e. those positioned farther than 1.7 μm away from the centrioles) only 32.8 ± 24.9% showed an association with other MT lattices at a given distance of 35 nm (*Figure 8—figure supplement 1*). This suggested that for an interaction distance of 35 nm roughly only 30% of the KMT minus ends in k-fibers were associated with the MT network. Further considering larger distances of association between KMT minus ends and neighboring MT lattices, we also observed that not all KMT minus ends were associated with neighboring MTs even at a value of 100 nm (*Table 6* and *Table 7*).

Next, we sought to map the positions of the detected associations of KMT minus ends with either KMT or non-KMT lattices within the reconstructed spindles. We determined the position of such associations in our spindles and then plotted the data against the relative position on the pole-to-kinetochore axis. For this, we normalized the pole-to-kinetochore axis by the MT density at each given position. We first plotted the normalized number of KMT minus-end associations with MT lattices against the relative position on the pole-to-kinetochore axis (*Figure 8A*; *Figure 8—figure supplement 3A*). KMT minus ends were distributed along the pole-to-kinetochore axis with a preference for positions at the spindle poles. As an example, for a given association distance of 35 nm, 60.7 ± 9.4% of the total number of associations were observed at the spindle poles. We then also determined the

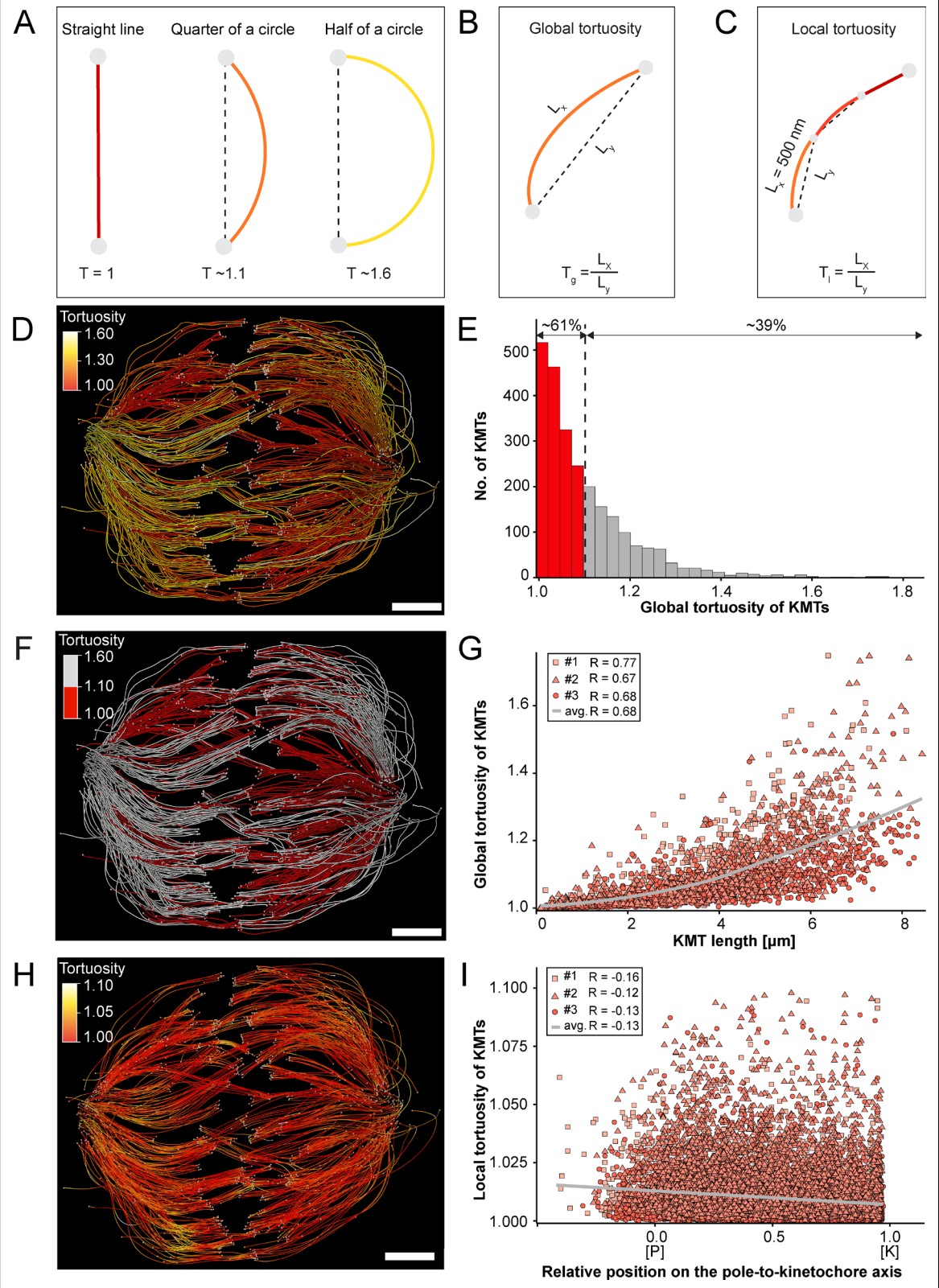

**Figure 6.** Global and local tortuosity of KMTs. (**A**) Schematic illustration of tortuosity (T) as given for a straight line, a quarter of a circle, and a half of a circle. (**B**) Schematic illustration of global tortuosity ($T_g$) of KMTs given by the ratio of the spline length ($L_x$) to the 3D distance between the KMT ends illustrated by gray circles ($L_y$). (**C**) Schematic illustration of KMT local tortuosity ($T_l$) as given by division segments with a length of 500 nm. (**D**) Three-dimensional model of k-fibers (spindle #1) showing the global tortuosity of KMTs as indicated by color coding (top left corner). (**E**) Histogram showing

*Figure 6 continued on next page*

*Figure 6 continued*

the frequency of tortuosity for KMTs (n=2579). The Pearson's correlation coefficient is given for each reconstructed spindle. The black dashed line indicates the average KMT tortuosity. The percentage ratio of 'straight' to 'curved' KMTs is also given. (**F**) Perspective view as shown in D. Straight KMTs (tortuosity of 1.0–1.1; red) and curved KMTs (tortuosity ≥1.1; white) are highlighted. (**G**) Correlation of global tortuosity and length of KMTs (n=2579). The Pearson's correlation coefficient is given for each reconstructed spindle. The gray line indicates the local regression calculated by the loess method. (**H**) Three-dimensional model of k-fibers (from spindle #1) showing the local tortuosity of KMTs as indicated by color-coding. (**I**) Correlation of the local tortuosity of KMTs with the relative position along the pole (P)-to-kinetochore (K) axis (n=2579). Scale bars, 1 μm.

The online version of this article includes the following video and figure supplement(s) for figure 6:

**Figure supplement 1.** Global tortuosity of KMTs based on the k-fiber position in the spindle.

**Figure 6—video 1.** Analysis of k-fiber global tortuosity in spindle #1.

https://elifesciences.org/articles/75459/figures#fig6video1

**Figure 6—video 2.** Analysis of k-fiber global tortuosity in spindle #2.

https://elifesciences.org/articles/75459/figures#fig6video2

**Figure 6—video 3.** Analysis of k-fiber global tortuosity in spindle #3.

https://elifesciences.org/articles/75459/figures#fig6video3

---

relative position of the KMT minus-end associations with non-KMT lattices (*Figure 8B*; *Figure 8—figure supplement 3B*). Similarly, the majority of the associations of KMT minus ends with non-KMT lattices were observed at the spindle poles. For the chosen distance of 35 nm, 44.7 ± 5.2% of these associations were observed at the spindle poles. Thus, the spindle poles appeared as the major sites for interaction of KMT minus ends with neighboring MT lattices.

*Vice versa*, we also determined the occurrence of either KMT or non-KMT minus ends in the vicinity of KMT lattices (*Figure 8*; *Figure 8—figure supplement 4*; *Figure 8—video 2*). At 35 nm or closer to the KMT lattice, we observed that on average 42 ± 8% of KMTs were associated with either KMT or non-KMT minus ends, with the majority of associations with non-KMT minus ends (*Figure 8—figure supplement 3C-D*; *Table 8* and *Table 9*). Moreover, we also determined the relative position of these associations on the spindle axis. Again, more than half of the KMT lattices (59.8 ± 6.7%) associated with other MT minus ends were preferentially found at spindle poles (*Figure 8C*; *Figure 8—figure supplement 3E*). In contrast, only 39.1 ± 4.6% of non-KMTs associated with other MT minus ends were found at the poles (*Figure 8D*; *Figure 8—figure supplement 3F*). Again, this analysis indicated that the interaction of KMTs with other MTs preferentially takes place at the spindle poles regardless of the association distance. Notably, we could observe a peak of association between the KMT lattices and the non-KMT minus ends at a relative position of around 0.3 (*Figure 8D*), suggesting that the KMT lattices at this position are important for interactions with non-KMTs.

In addition, we were also interested in mapping the number and the length of MT-MT associations on the pole-to-pole axis in order to recognize specific patterns of interactions within the mitotic spindle. For a pairing length analysis as previously applied (*McDonald et al., 1992*; *Winey et al., 1995*), we defined 20 nm as a minimal length of interaction. For each MT, we also counted the number of continuous interaction segments over which they retained this minimal association proximity (*Figure 9A*). In addition, we also varied the distance between associated MTs by choosing values of 25, 30, 35, 45, and 50 nm. As expected, the peaks in the number of KMTs changed rapidly with an increase in the number and length of associations (*Table 9* and *Table 10*). We then analyzed the association of KMTs with other MTs in the spindle by plotting the number of associations against the relative position on the pole-to-pole axis. We also normalized the number of associations by the MT density. With an increase in the considered association distance between MTs, we observed an increase in the number of associations at the spindle poles and a drastic decline in the number of these associations at positions in the middle of the spindle (*Figure 9B*; *Figure 9—figure supplement 1A*; *Figure 9—video 1*). We then also analyzed the association of non-KMTs with other MTs. In contrast to the previous analysis, by increasing the association distances we detected a considerable increase in the number of interactions near the spindle midplane. (*Figure 9C*; *Figure 9—figure supplement 1B*; *Figure 9—video 2*). This peak is of functional importance, most likely representing the region, where kinesin motors generate pushing forces (*Shimamoto et al., 2015*). We concluded from all these analyses that KMTs and non-KMTs differ in their spatial pattern of MT-MT association. KMTs strongly interact with

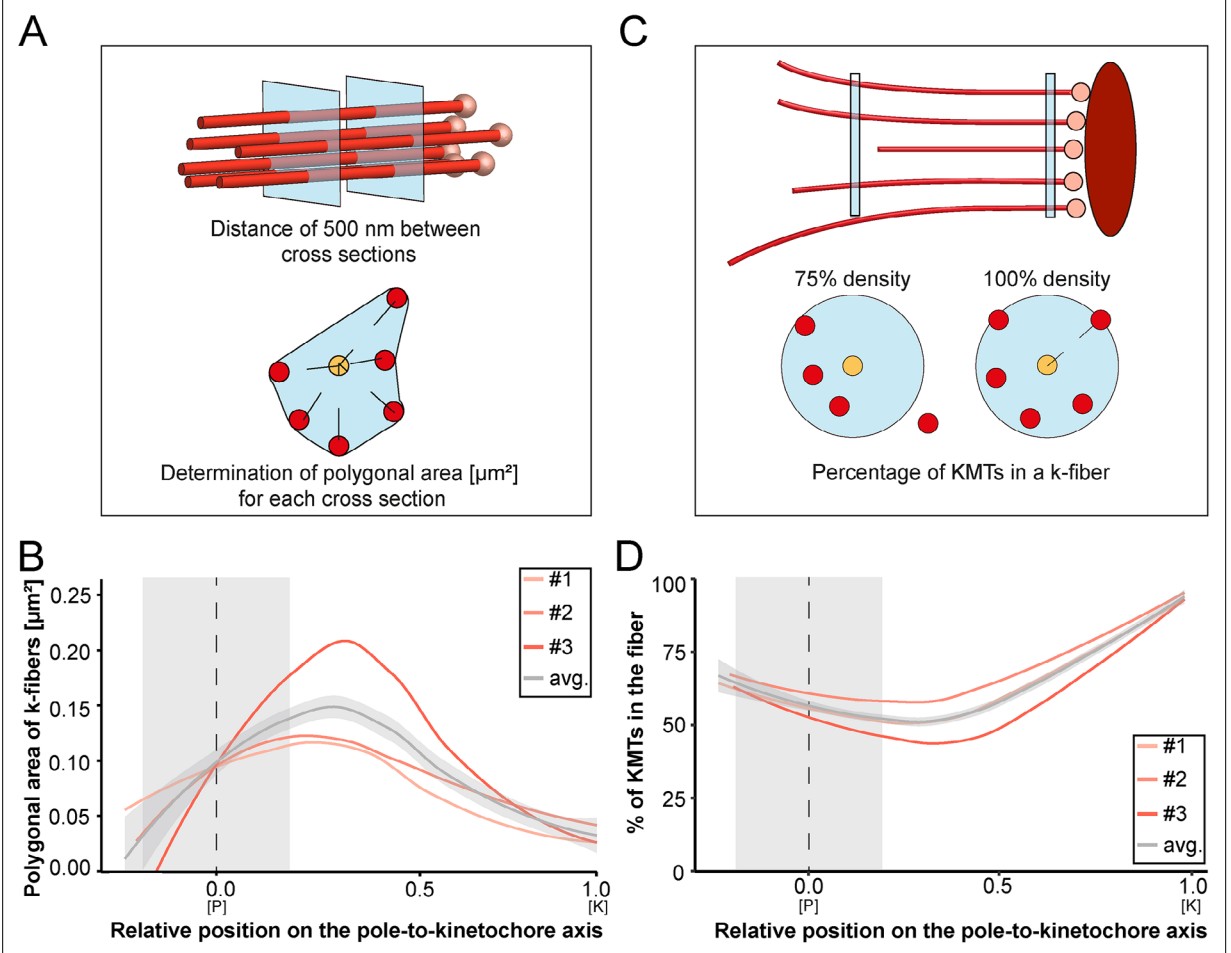

**Figure 7.** Shape of k-fibers. (**A**) Schematic illustration of the analysis of polygonal areas as obtained from k-fiber cross-sections. KMTs are shown as lines (red), KMT ends as spheres (light red). Cross-sections of the given k-fiber are shown as blue squares (top). The median position of all KMTs in a cross-section is indicated as a yellow circle (bottom). (**B**) Distribution of the k-fiber polygonal area along with the relative position on the pole [P]-to-kinetochore [K] axis (n=292). (**C**) Schematic illustration of the k-fiber density analysis. For each k-fiber, a radius at the kinetochore was estimated by calculating a minimum circle enclosing all KMTs (top). The determined radius was then enlarged by factor 2 to account for k-fiber flexibility. Along with the k-fiber, the number of KMTs enclosed in the selected radius was then measured (bottom). (**D**) Distribution of the percentage of KMTs enclosed in the k-fiber along with the relative position along the pole [P]-to-kinetochore [K] axis (n=292). For each reconstructed spindle, data sets are presented as polynomial lines showing local regression calculated with the loess method. Average values with standard deviations are shown in gray. The approximated MT-centrosome interaction areas are shown in gray with the position of the poles indicated by dashed lines (B and D).

The online version of this article includes the following figure supplement(s) for figure 7:

**Figure supplement 1.** Schematic illustration of the analysis of k-fiber area and KMT density.

neighboring MTs at the spindle poles, while non-KMTs show a broad region of MT-MT interaction within the middle of the spindle, potentially forming interpolar bundles (*Mastronarde et al., 1993*).

Finally, we were interested in how the distribution patterns of MT-MT associations change in relation to the position in the spindle. With our high-resolution 3D data sets covering all MTs in the spindle, we decided to investigate the number and the length of associations for both KMTs and non-KMTs as a function of the distance between MTs. Firstly, we analyzed the association of KMTs with any MT in the spindle (*Figure 9—figure supplement 2A-D*). As expected, with an increase in the considered distance between MTs, KMTs showed an increase in the number and also in the average length of interactions (*Table 10* and *Table 11*). For a given MT-MT distance of 35 nm, each KMT associates on average with 10.6 ± 3.2 (mean ±STD, n=2579) other MTs in the spindle with an average association length of 145 ± 186 nm (±STD, n=2579). Secondly, we also analyzed the association of non-KMTs with any MT in the spindle. Non-KMTs showed a similar pattern of increase in the number and length of associations with increasing distances between individual MTs. For 35 nm, each non-KMT

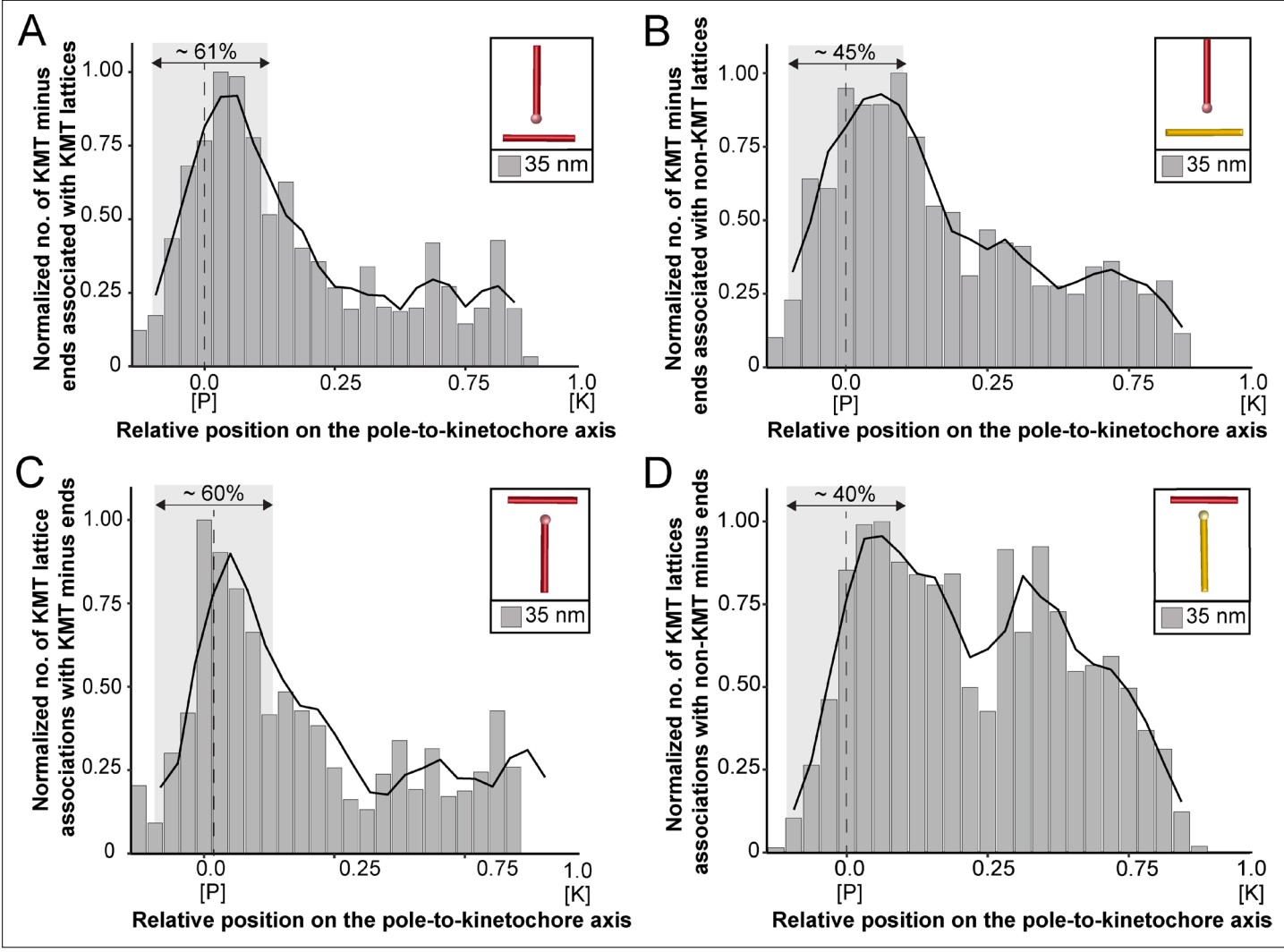

**Figure 8.** Association of KMTs with the MT network. (**A**) Graph showing the number of KMT minus ends associated with KMT lattices within 35 nm of interaction (n=2579). Numbers of KMT minus ends are normalized by the density of surrounding MTs and plotted against the relative position on the pole-to-kinetochore axis (P, pole; K, kinetochore). The approximated MT-centrosome interaction area is shown in gray with the position of the pole indicated by a dashed line. The percentage of KMT associations located in the MT-centrosome interaction area is given. (**B**) Bar plot showing the normalized number of KMT minus ends associated with non-KMT lattices within 35 nm distance (n=2579). (**C**) Graph showing the number of KMT lattices associated with other KMT minus ends plotted along the relative position on the pole-to-kinetochore axis and normalized by the spindle density (n=2579). (**D**) Graph displaying the number of KMT lattices associated with non-KMT minus ends (n=2579). Moving averages with a period of 0.05 along the pole-to-kinetochore axis are shown as black lines.

The online version of this article includes the following video and figure supplement(s) for figure 8:

**Figure supplement 1.** Association of KMT minus ends with other MT lattices.

**Figure supplement 2.** Association of KMT minus ends with KMT and non-KMT lattices at an interaction distance of 35 nm.

**Figure supplement 3.** Association of KMTs with other MTs.

**Figure supplement 4.** Association of KMT lattices with other MT minus ends at an interaction distance of 35 nm.

**Figure 8—video 1.** Associations of KMT minus ends with MT lattices in spindle #1.
https://elifesciences.org/articles/75459/figures#fig8video1

**Figure 8—video 2.** Association of KMT minus end with MT lattices.
https://elifesciences.org/articles/75459/figures#fig8video2

**Table 6.** Analysis of the association of KMT minus ends with other neighboring KMT lattices.

| Data set | Analysis | Interaction distances [nm] | | | | | | |
|---|---|---|---|---|---|---|---|---|
| | | 25 | 30 | 35 | 45 | 50 | 75 | 100 |
| Spindle #1 | No. of KMTs | 37 | 68 | 112 | 204 | 238 | 306 | 330 |
| | % of KMTs | 4.9 | 9.1 | 15.0 | 27.3 | 32.0 | 40.9 | 44.1 |
| Spindle #2 | No. of KMTs | 20 | 37 | 68 | 142 | 177 | 266 | 290 |
| | % of KMTs | 1.9 | 3.5 | 6.3 | 13.2 | 16.5 | 24.8 | 27.1 |
| Spindle #3 | No. of KMTs | 13 | 27 | 66 | 116 | 135 | 199 | 218 |
| | % of KMTs | 1.9 | 4.0 | 9.8 | 17.2 | 20.0 | 29.5 | 32.3 |

associates on average with 7.4 ± 3.1 (mean ±STD, n=16256) other MTs in the spindle with an average association length of 103±118 nm (mean ±STD, n=16256). With an increase in the distance between MTs, we observed that KMTs tend to show a higher number and a higher average length of associations compared to non-KMTs. Importantly, these results were consistent for all selected association distances (*Figure 9—figure supplement 2E-F*).

## Discussion

Large-scale reconstruction by serial-section electron tomography (*Fabig et al., 2020*; *Redemann et al., 2018*; *Redemann et al., 2017*) allowed us to quantitatively analyze KMT organization in individual k-fibers and in the context of whole mitotic spindles.

### Methodological considerations

For generating 3D reconstructions of spindles, we applied electron microscopy of plastic sections. The use of plastic sections suffers from the fact that samples undergo a collapse in the electron beam during imaging, and this is obvious by a reduction in the section thickness (*Luther et al., 1988*; *McEwen and Marko, 1998*; *O'Toole et al., 2020*). By expanding the complete stack of serial tomograms (*Figure 2—figure supplement 2*), it is possible to correct this loss in Z, and we did so for our three data sets covering whole metaphase spindles in HeLa cells.

Here, we used serial, semi-thick sections of plastic-embedded material for a 3D tomographic reconstruction of whole spindles. Although serial sectioning is never perfect, in that the section thickness within ribbons always shows some variability, we were able to produce data sets of remarkable similarity. This is true for our analysis of MT length distribution (*Figure 4*; *Figure 4—figure supplement 1*) and our measurements of minus-end distance to the spindle poles and minus-end positioning (*Figure 5*; *Figure 5—figure supplement 2*). In combination with a semi-automatic segmentation and stitching of MTs (*Lindow et al., 2021*; *Weber et al., 2012*), our approach enabled us to reliably model individual MTs over section borders, thus allowing a quantitative study of MT length and end-positioning in whole spindles. In the future, we will use this routine approach to quantify MT organization also in other mammalian systems, such as RPE1 and U2OS cells.

**Table 7.** Analysis of the association of KMT minus ends with neighboring non-KMT lattices.

| Data set | Analysis | Interaction distances [nm] | | | | | | |
|---|---|---|---|---|---|---|---|---|
| | | 25 | 30 | 35 | 45 | 50 | 75 | 100 |
| Spindle #1 | No. of KMTs | 37 | 82 | 132 | 217 | 248 | 353 | 384 |
| | % of KMTs | 4.9 | 11.0 | 17.6 | 29.0 | 33.2 | 47.2 | 51.3 |
| Spindle #2 | No. of KMTs | 245 | 313 | 353 | 469 | 525 | 677 | 732 |
| | % of KMTs | 22.9 | 29.2 | 33.0 | 43.8 | 49.0 | 63.2 | 68.3 |
| Spindle #3 | No. of KMTs | 28 | 64 | 107 | 198 | 230 | 355 | 410 |
| | % of KMTs | 4.2 | 9.5 | 15.9 | 29.4 | 34.1 | 52.7 | 60.8 |

**Table 8.** Analysis of the association of KMT lattices with other neighboring KMT minus ends.

| Data set | Analysis | Interaction distances [nm] | | | | | | |
|---|---|---|---|---|---|---|---|---|
| | | 25 | 30 | 35 | 45 | 50 | 75 | 100 |
| Spindle #1 | No. of KMTs | 39 | 71 | 117 | 210 | 236 | 336 | 403 |
| | % of KMTs | 5% | 10% | 15% | 28% | 31% | 45% | 54% |
| Spindle #2 | No. of KMTs | 24 | 46 | 86 | 179 | 237 | 401 | 470 |
| | % of KMTs | 2% | 4% | 8% | 17% | 22% | 37% | 43% |
| Spindle #3 | No. of KMTs | 14 | 27 | 61 | 127 | 148 | 227 | 284 |
| | % of KMTs | 2% | 4% | 9% | 19% | 22% | 34% | 43% |

In electron microscopic images, centrosomes or spindle poles are visible by pairs of centrioles surrounded by electron-dense pericentriolar material (PCM). Since these membrane-less organelles do not show a clear boundary in thin sections or in electron tomograms, it is not immediately obvious how to define the edge of the spindle pole. Inspired by earlier studies on the early *C. elegans* embryo (*O'Toole et al., 2003*; *Redemann et al., 2017*; *Weber et al., 2012*), we determined the edge of the spindle pole from the density distribution of non-KMT minus ends in the spindle. The non-KMT minus-end density peaked a micron away from the pole and then fell before leveling off at constant non-KMT minus-end density in the spindle bulk. We defined the edge of the spindle pole as twice the half-width from the center of the non-KMT minus-end density peak. In the HeLa spindles, this was 1.7 µm from the mother centriole. We applied the same cutoff in a parallel study on the dynamics of mammalian k-fibers (see Figure 1 in *Conway et al., 2022*).

In this parallel study, we supplemented our electron tomography data on the KMT length distribution with light microscopic data. Essentially, our 3D reconstructions show a distribution of KMT length in metaphase that is strikingly similar to the distribution plot of KMT length as obtained by biophysical modeling in combination with light microscopy (see Figure 8B-D in *Conway et al., 2022*). All this shows that light and electron microscopy produces truly complementary data, although completely different methods of sample preparation and data analysis have to be applied.

## KMT organization

Counting the total number of KMTs and non-KMTs in our spindles, we show that only ~14% of all MTs in the reconstructed spindles were KMTs. However, this percentage in the total number of all MTs corresponds to ~25% of the tubulin mass as measured in parallel by light microscopy (*Conway et al., 2022*). Comparing the average length of KMTs and non-KMTs, we also find that KMTs are on average twice as long as non-KMT. Thus, a higher value in the average length of KMTs *versus* non-KMTs contributes to a higher percentage in the tubulin mass of KMTs compared to all other MTs in the spindle.

The length distribution of KMTs in HeLa cells shows striking similarities to the distribution of KMTs observed in the early *C. elegans* embryo (*Redemann et al., 2017*). Both human KMTs attached to monocentric kinetochores and also nematode KMTs associated with dispersed holocentric kinetochores

**Table 9.** Analysis of the association of KMT lattices with other neighboring non-KMT minus ends.

| Data set | Analysis | Interaction distances [nm] | | | | | | |
|---|---|---|---|---|---|---|---|---|
| | | 25 | 30 | 35 | 45 | 50 | 75 | 100 |
| Spindle #1 | No. of KMTs | 81 | 151 | 223 | 362 | 415 | 534 | 577 |
| | % of KMTs | 11% | 20% | 30% | 48% | 55% | 71% | 77% |
| Spindle #2 | No. of KMTs | 51 | 100 | 173 | 351 | 433 | 640 | 717 |
| | % of KMTs | 5% | 9% | 16% | 33% | 40% | 59% | 67% |
| Spindle #3 | No. of KMTs | 34 | 93 | 176 | 301 | 348 | 471 | 507 |
| | % of KMTs | 5% | 14% | 26% | 44% | 51% | 69% | 75% |

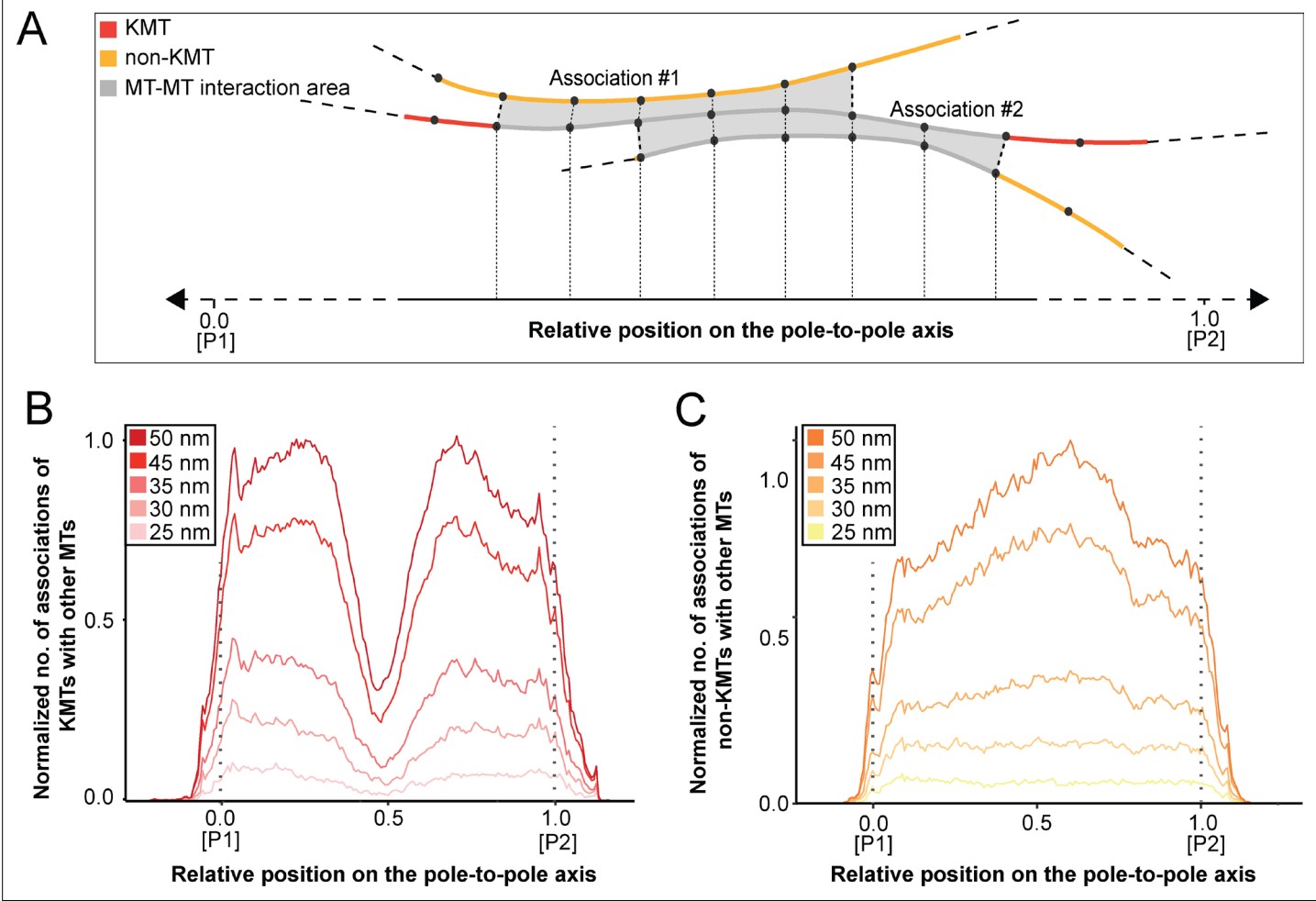

**Figure 9.** Positions of MT-MT associations. (**A**) Schematic illustration showing the mapping of the number of MT-MT associations on the pole-to-pole axis (P1, position = 0; P2, position = 1). The number of associations is measured in defined segments (20 nm). KMTs are illustrated in red, non-KMTs in yellow, and areas of MT-MT association in gray. (**B**) Graph showing the number of KMTs associated with other MTs plotted against the relative position on the pole-to-pole axis. KMT number is normalized by the MT density. The defined association distances for KMTs with other MTs in the spindle are given in the insert. (**C**) Number of non-KMTs associated with neighboring MTs plotted against the relative position on the pole-to-pole axis.

The online version of this article includes the following video and figure supplement(s) for figure 9:

**Figure supplement 1.** Association of KMT lattices with other MT lattices at an interaction distance of 35 nm.

**Figure supplement 2.** Association of KMT lattices with MT minus ends.

**Figure 9—video 1.** Associations of KMT lattices with other MTs.

https://elifesciences.org/articles/75459/figures#fig9video1

**Figure 9—video 2.** Associations of non-KMTs with other MTs.

https://elifesciences.org/articles/75459/figures#fig9video2

show a rather flat length distribution and a rather low number of both very short and very long KMTs. In contrast, non-KMTs in both systems show an exponential length distribution with a very high occurrence of very short MTs (around 57% of the non-KMTs and ~21% of KMT in HeLa cells were less than 2 μm in length). Exponential length distributions as found for non-KMTs are typical of dynamic instability kinetics (*Burbank et al., 2007*; *Loiodice et al., 2019*). The observed length distribution of KMTs, however, indicates a difference in dynamics and possibly higher stability of the plus-ends against MT depolymerization. Taken together, all this argues that KMTs in both spindles have distinct properties different from those of non-KMTs.

A difference in the properties between KMTs and non-KMTs is also obvious after a cold treatment of cells. Such treated cells show cold-stable k-fibers, while most of the non-KMTs undergo

**Table 10.** Average number of associations of KMTs and non-KMTs with MT lattices.

| Data set | MT type | Interaction distances [nm] | | | | |
| | | 25 | 30 | 35 | 45 | 50 |
|---|---|---|---|---|---|---|
| Spindle #1* | KMTs | 4.8 (±1.5) | 8.6 (±2.5) | 12.4 (±3.4) | 18.8 (±4.9) | 1.3 (±5.5) |
| | Non-KMTs | 4.4 (±1.5) | 7.0 (±2.6) | 9.6 (±3.7) | 13.0 (±5.4) | 16.2 (±6.2) |
| Spindle #2* | KMTs | 4.2 (±1.3) | 5.8 (±1.8) | 8.0 (±2.6) | 13.3 (±4.1) | 16.0 (±4.8) |
| | Non-KMTs | 3.2 (±0.9) | 4.0 (±1.3) | 5.2 (±1.8) | 8.2 (±3.1) | 9.8 (±3.7) |
| Spindle #3* | KMTs | 4.2 (±1.2) | 8.0 (±2.3) | 12.4 (±3.4) | 18.6 (±4.8) | 21.2 (±5.4) |
| | Non-KMTs | 3.6 (±1.2) | 5.2 (±2.3) | 8.0 (±3.3) | 11.6 (±4.7) | 13.2 (±5.3) |
| All spindles* | KMTs | 4.4 (±1.3) | 7.4 (±2.3) | 10.6 (±3.2) | 16.4 (±4.7) | 19.0 (±5.3) |
| | Non-KMTs | 3.6 (±1.2) | 5.4 (±2.2) | 7.4 (±3.1) | 10.8 (±4.5) | 12.4 (±5.1) |

*Numbers are given as mean ±STD.

depolymerization upon exposure to cold (*Maiato et al., 2004*). Here, we can only speculate about this resistance to cold temperature. Likely, KMTs are stabilized by interaction with the kinetochores (*Brinkley and Cartwright, 1975*; *DeLuca et al., 2006*; *Warren et al., 2020*) and/or by KMT-KMT/ KMT-non-KMT associations, possibly mediated by several MT-associated proteins (*Agarwal, 2018*). It is also possible that non-KMTs, involved in k-fiber maturation during mitosis (*Maiato et al., 2004*), contribute to such stabilization of k-fibers in mammalian cells.

Electron tomography revealed that on average nine KMTs are attached to each kinetochore in HeLa cells in metaphase. This result differs from previous observations in PtK1 cells (*McEwen et al., 1997*; *O'Toole et al., 2020*). In this marsupial cell line, about 20 KMTs were reported to connect to

**Table 11.** Average length of associations of KMTs and non-KMTs with MT lattices.

| Data set | MT type | Interaction distances [nm] | | | | |
| | | 25 | 30 | 35 | 45 | 50 |
|---|---|---|---|---|---|---|
| Spindle #* | KMTs | 81.3 (±88.8) | 119.7 (±151.1) | 163.9 (±207.5) | 241.3 (±301.9) | 271.0 (±335.9) |
| | Non-KMTs | 58.3 (±54.1) | 78.3 (±79.9) | 107.7 (±116.9) | 165.1 (±195.1) | 187.9 (±227.0) |
| Spindle #2* | KMTs | 69.5 (±69.9) | 93.2 (±107.0) | 124.3 (±146.0) | 207.8 (±252.3) | 252.2 (±314,9) |
| | Non-KMTs | 59.2 (±53.2) | 73.2 (±71.9) | 92.3 (±97.3) | 145.8 (±170.5) | 175.1 (±213.0) |
| Spindle #3* | KMTs | 66.1 (±63.4) | 97.3 (±117.8) | 143.2 (±191.7) | 231.3 (±321.2) | 263.4 (±362.2) |
| | Non-KMTs | 54.3 (±51.7) | 74.5 (±86.4) | 104.6 (±133.7) | 165.6 (±218.7) | 191.3 (±252.0) |
| All spindles* | KMTs | 73.0 (±76.2) | 104.6 (±129.2) | 145.1 (±186.0) | 225.6 (±292.3) | 261.9 (±336.8) |
| | Non-KMTs | 57.2 (±53.1) | 75.4 (±80.5) | 102.2 (±118.9) | 159.1 (±197.2) | 184.9 (±232.1) |

*Numbers are given as mean ±STD.

the kinetochores. This difference in the number of attached KMTs could be related to kinetochore size. As previously observed by light microscopy, kinetochores in HeLa cells are about half the size of kinetochores in PtK1 cells (*Cherry et al., 1989*). Similarly, kinetochore size in PtK$_1$ cells was 0.157 ± 0.045 μm$^2$ (mean ±STD) as observed by electron tomography (*McEwen et al., 1998a*), whereas kinetochores in HeLa cells, as determined indirectly in this study, have an estimated size of about 0.107 ± 0.075 μm$^2$ (mean ±STD). Possibly, the area of the outer kinetochore might indirectly define the size and/or the number of available free binding sites for MTs (*Drpic et al., 2018*; *Monda and Cheeseman, 2018*). Concerning the number of kinetochore-attached MTs, it is interesting to note here that the number of KMTs per k-fiber is not related to the position of these KMTs in the spindles. In fact, central, intermediate and peripheral kinetochores show similar average numbers of attached KMTs. Thus, the peripheral position of k-fibers within the spindle accompanied by an increase in the global tortuosity has no effect on the number of KMTs in the k-fibers.

KMTs in our reconstructed k-fibers are of different lengths, confirming previous observations (*McDonald et al., 1992*; *O'Toole et al., 2020*; *Sikirzhytski et al., 2014*). In fact, many KMTs are relatively short (~20% of KMTs were shorter than 2 μm; *Figure 4A*), and half of the KMT minus ends are not positioned in the defined MT-centrosome interaction area. Per definition, these short KMTs in k-fibers are not directly associated with the spindle poles. Interestingly, only 5% of the analyzed

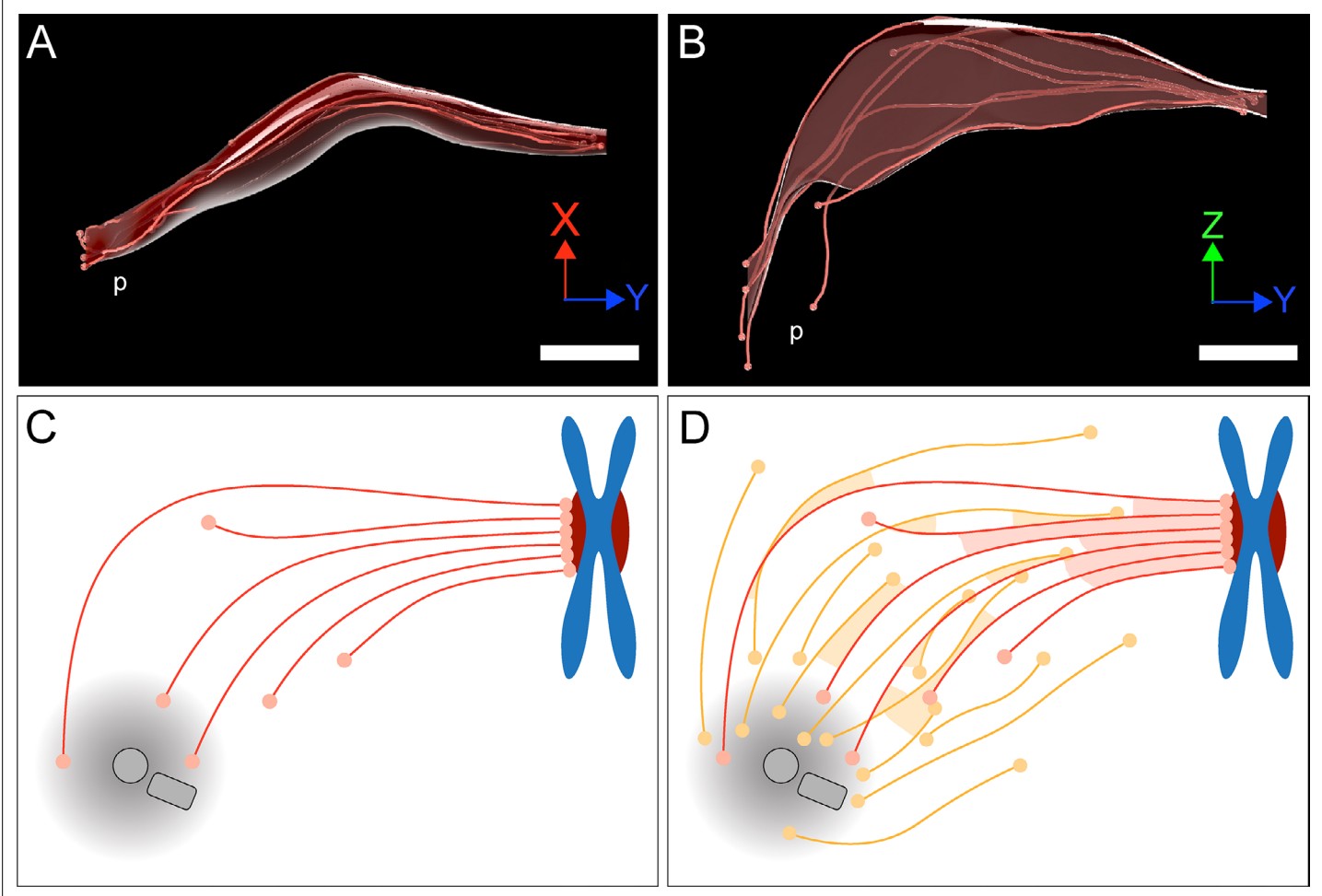

**Figure 10.** Model of a k-fiber showing a semi-direct connection between a kinetochore and a spindle pole. (**A–B**) Three-dimensional views of a selected 3D-reconstructed k-fiber with an overlay area drawn around KMTs using the alpha shape method. The KMTs are shown as red lines and the ends are marked with red dots. The approximate position of the pole is indicated (p). The same k-fiber is shown from two different perspectives (X/Y view in A; Z/Y view in B). Scale bars, 1 μm. (**C**) Schematic model of a semi-direct connection between a kinetochore (chromosome in blue, paired kinetochores in dark red) and a spindle pole (MT-centrosome interaction area and centrioles in gray) as established by a single k-fiber. KMTs are shown as red lines, KMT ends as light red circles. (**D**) Model of the k-fiber (as shown in C) associated with the surrounding non-KMT network. Non-KMTs are shown as yellow lines, non-KMT ends as yellow circles. KMT-KMT interactions are indicated by light red areas, KMT-non-KMT interactions by light yellow areas.

k-fibers show a length distribution in which none of the analyzed KMTs is positioned in the MT-centrosome interaction area (*Figure 5—figure supplement 3*). When analyzing KMTs in the one-cell *C. elegans* embryo, we found that only about 20% of the KMT minus ends were located within 2 μm of their corresponding mother centriole. This suggested that the majority of KMTs in *C. elegans* do not contact the centrosomes. In agreement with previously published data (*McDonald et al., 1992*; *O'Toole et al., 2020*), our tomographic analysis of mammalian KMTs thus suggests that the k-fibers in HeLa cells mediate a semi-direct connection with the spindle poles, in which at least one KMT of the k-fibers is directly connected to the poles, while the other KMTs of the fiber are indirectly linked to non-KMTs (*Figure 10*). Thus, spindles in nematode embryos and in mammalian cells are similar in that anchoring of KMTs into the spindle network can be observed.

Interestingly, we observed a difference in KMT length and their minus-end distance to the pole in central *versus* peripheral KMTs. Centrally located KMTs were shorter, and their minus ends showed a shorter distance to the mother centriole compared to peripheral KMTs. This difference is most likely related to the roundedness of the mitotic spindles (*Taubenberger et al., 2020*). To test whether the roundedness of spindles and the organization of KMTs in terms of KMT length and minus-end distribution are directly related, it would be interesting to analyze the organization of KMTs in spindles showing a lower degree of rounding up during mitoses such as in PtK1 (*McDonald et al., 1992*) and RPE1 cells (*O'Toole et al., 2020*).

Extending previous knowledge, we have shown that k-fibers in our reconstructions show a remarkable morphological variability, as obvious by a change in the circumference of the k-fibers along their entire length (*Figure 10A–B*). This variability in the circumference of the k-fibers is reflected in an increase in the local tortuosity of KMTs at positions close to the spindle poles. An increase in the tortuosity of KMTs at spindle poles might promote the anchoring of the broadened k-fibers into the non-KMT network through MT-MT interactions (*Figure 10C–D*).

Here, we consider the bundled KMTs as the 'core' of the k-fibers (*Figure 10C*). We used the annotated KMTs in our reconstruction to identify other non-KMTs associated with these KMTs. In other words, we annotated the KMTs in the spindles to 'fish out' other non-KMTs out of more than 6000 MTs to identify those non-KMTs that were positioned in the vicinity of the reconstructed KMTs. Explicitly, the results obtained from our approach do not exclude models of KMT organization, in which the k-fiber is a tight bundle that continues to the pole with changing composition of KMTs and associated non-KMTs along its length. In this sense, differences in either the length of KMTs or in the loss of KMTs from the k-fiber might simply reflect a MT exchange with the spindle (*Figure 10D*). Our consideration of KMTs as the cores of k-fibers is also not in disagreement with a dynamic change in k-fiber composition during the maturing of those fibers in metaphase (*Begley et al., 2021*; *Maiato et al., 2004*). Unfortunately, our 3D reconstructions can deliver only snapshots of the very dynamic mitotic process.

While both KMTs and non-KMTs show a clear correlation in the number and the average length of associations (*Figure 9—figure supplement 2*), both MT populations show differences in the position of these associations. In contrast to non-KMTs, KMTs show a high tendency to associate with non-KMTs at the spindle poles (*Figure 8F–G*; *Figure 9C–D*). This tendency to interact at spindle poles is independent of the chosen distance of MT interaction. In accord with the previously discussed broadening of the k-fibers at their pole-facing end, our results suggest that KMTs preferably associate with other MTs at the spindle poles. In contrast, non-KMTs show a flat pattern of interaction with other MTs at association distances of 25 and 35 nm. Moreover, an increase in the association distance from 35 nm to 50 nm, shows a higher tendency of non-KMTs to associate with MTs in the center of the spindle, very likely related to the organization of interpolar MTs in the center of the spindle (*Figure 8—figure supplement 1F*; *Kajtez et al., 2016*; *Mastronarde et al., 1993*; *Vukušić et al., 2017*). In general, it would be interesting to analyze the organization of these interpolar MTs, the structure of the KMTs in the k-fibers, and also the recognized patterns of MT-MT interaction during other stages of mitosis, for instance at anaphase. Patterns of the interaction of KMTs with non-KMTs might be more obvious during the segregation of the chromosomes.

## Implications for models on spindle organization

As previously noted, we have combined our 3D reconstructions with additional live-cell imaging and biophysical modeling in a parallel publication (*Conway et al., 2022*). Combining data on the length and the position of KMT minus ends in spindles (as obtained here by electron microscopy), and the

turnover and movement of tubulin in KMTs as generated by light microscopy, a model was proposed in which KMTs predominantly nucleate de novo at kinetochores, with KMTs growing towards the spindle poles. A major outcome of this parallel study is that KMTs in spindles grow along the same trajectories as non-KMTs and that both the KMTs and non-KMTs are well aligned throughout the spindle, leading to the assumption that spindles can be considered as active liquid crystals (*Brugués and Needleman, 2014*; *Oriola et al., 2020*). This might apply to both centrosomal mitotic as well as acentrosomal female meiotic spindles (*Redemann et al., 2018*; *Redemann et al., 2017*). Such liquid crystals can be characterized by the degree of local MT alignment, expressed by the nematic order parameter. Interestingly, the analyzed spindles show a high nematic order parameter (S = 0.81 ± 0.02) near the chromosomes, whereas the nematic order parameter (S = 0.54 ± 0.02) is lower at the spindle poles (*Conway et al., 2022*). Along this line, KMTs in our electron tomography study are well aligned in the middle of the spindle, while the order of the KMTs in the k-fibers is progressively lost at positions closer to the spindle poles. While KMTs are growing out from the kinetochores towards the centrosomes, the observed broadening of the k-fibers at the spindle poles might be a direct consequence of a decrease in the internal structural organization of the spindle trajectories (i.e. of the surrounding non-KMTs). In the future, it will be important to analyze k-fibers in other fully 3D-reconstructed mammalian spindles to advance the developed model on KMT outgrowth in the context of such well-defined trajectories.

## Materials and methods

**Key resources table**

| Reagent type (species) or resource | Designation | Source or reference | Identifiers | Additional information |
|---|---|---|---|---|
| Strain, background (HeLa, Kyoto) | Gerlich Lab | IMBA, Vienna, Austria | - | - |
| Software, algorithm | SerialEM Boulder Laboratory for 3-Dimensional Electron Microscopy of cells Colorado, USA | https://bio3d.colorado.edu/ *Mastronarde, 2003* | - | - |
| Software, algorithm | IMOD Boulder Laboratory for 3-Dimensional Electron Microscopy of cells Colorado, USA | http://bio3d.colorado.edu/ *Kremer et al., 1996* | - | - |
| Software, algorithm | Amira Thermo Fisher Scientific, USA | https://www.zib.de/software/amira *Stalling et al., 2005* | - | - |
| Software, algorithm | ASGA Robert Kiewisz / Müller - Reichert Lab Dresden, Germany | https://github.com/RRobert92/ *Kiewisz and Müller-Reichert, 2021* | - | https://kiewisz.shinyapps.io/ASGA |
| Software, algorithm | ASGA - 3D Viewer Robert Kiewisz / Müller - Reichert Lab Dresden, Germany | https://github.com/RRobert92/ *Kiewisz and Müller-Reichert, 2022* | - | https://cfci.shinyapps.io/ASGA_3DViewer/ |

## Cell line

For all experiments, we have used a HeLa Kyoto cell line obtained from Dr. Daniel Gerlich (IMBA, Vienna), which was given to the Gerlich lab by S. Narumiya (Kyoto, Japan; RRID: CVCL_1922) and validated using the Multiplex Human Cell Line Authentication test (MCA). Furthermore, the HeLa Kyoto cell line was checked for mycoplasma with a PCR test kit. This cell line was not on the list of commonly misidentified cell lines as maintained by the International Cell Line Authentication Committee.

## Cultivation of cells

HeLa cells (*Guizetti et al., 2011*) were grown in Dulbecco's Modified Eagle's Medium (DMEM) supplemented with 10% fetal bovine serum (FBS) and 100 units/ml of penicillin/streptomycin (Pen/Strep). Flasks were placed in a humidified incubator at 37°C with a supply of 5% $CO_2$. For electron microscopy, cells in mitosis were enriched by applying the shake-off technique (*Kiewisz et al., 2021*). Flasks with cell confluency of 60–80% were shaken against the laboratory bench. The medium with detached cells was then collected, centrifuged at 1200 rpm for 3 min at room temperature, and resuspended in 1 ml of pre-warmed DMEM medium.

## Electron tomography

### Specimen preparation for electron microscopy

Cultures enriched in mitotic HeLa cells were further processed for electron microscopy essentially as described (*Guizetti et al., 2011*; *Kiewisz et al., 2021*). Briefly, sapphire discs with a diameter of 6 mm were cleaned in Piranha solution (1:1 $H_2SO_4$ and $H_2O_2$, v/v), coated with poly-L-lysine (0.1% in dd$H_2O$, w/v), and dried for 2 hrs at 60°C. Furthermore, the discs were coated with fibronectin (1:10 dilution in 1 x PBS, v/v) for 2 hr and stored in a humidified incubator until further used. The sapphire discs were then placed into custom-designed 3D-printed incubation chambers (*Kiewisz et al., 2021*). Subsequently, cells were seeded on the coated sapphire discs and incubated for 10 min in a humidified incubator at 37°C supplied with 5% $CO_2$. This allowed the mitotic cells to re-attach to the surface of the coated sapphire discs and continue to divide.

### High-pressure freezing and freeze substitution

Cells were cryo-immobilized using an EM ICE high-pressure freezer (Leica Microsystems, Austria). For each run of freezing, a type-A aluminum carrier (Wohlwend, Switzerland) with the 100 μm-cavity facing up was placed in the specimen loading device of the EM ICE. The cavity of the type-A carrier was filled with 5 μl of DMEM containing 10% BSA. The carrier was then immediately closed by placing a 6 mm-sapphire disc with attached cells facing down on top of the type-A carrier. Finally, a spacer ring was mounted on top of the closed carrier, and freezing was started. Samples were frozen under high pressure (~2000 bar) with a cooling rate of ~20000°C/s (*Reipert et al., 2004*). Frozen samples were then opened under liquid nitrogen and transferred to cryo-vials filled with anhydrous acetone containing 1% (w/v) osmium tetroxide (EMS, USA) and 0.1% (w/v) uranyl acetate (Polysciences, USA). Freeze substitution was performed in either a Leica AFS or a Lecia AFS II (Leica Microsystems, Austria). Samples were kept at –90°C for 1 hr, warmed up to –30°C with increments of 5°C/hr, kept for 5 hrs at –30°C, and then warmed up to 0°C (increments of 5°C/hr). Finally, samples were allowed to warm up to room temperature. After freeze substitution, samples were washed three times with pure anhydrous acetone and infiltrated with Epon/Araldite (EMS, USA) using increasing concentrations of resin (resin:acetone: 1:3, 1:1, 3:1, then pure resin) for 1 hr each step at room temperature (*Müller-Reichert et al., 2003*). Samples were infiltrated with pure resin overnight and then embedded by using commercial flow-through chambers (Leica Microsystems, Austria) designed for sapphire discs of a diameter of 6 mm. Samples were polymerized at 60°C for 36 hr.

### Pre-selection of staged cells

To select cells in metaphase, resin-embedded samples were pre-inspected using an Axiolab RE upright brightfield microscope (Zeiss, Germany) with a 5 x and a 40 x objective lens (Zeiss, Germany). Selected cells in metaphase were sectioned using an EM UC6 ultramicrotome (Leica Microsystems, Austria). Ribbons of semi-thick (~300 nm) serial sections were collected on Formvar-coated copper slot grids, post-stained with 2% (w/v) uranyl acetate in 70% (v/v) methanol, followed by 0.4% (w/v) lead citrate (Science Services, USA) in double-distilled water. In addition, 20 nm-colloidal gold (British Biocell International, UK) was attached to the serial sections, serving as fiducial markers for subsequent electron tomography. The selected cells were then pre-inspected at low magnification (~2900 x) using either an EM906 (Zeiss, Germany) or a TECNAI T12 Biotwin (Thermo Fisher Scientific, USA) transmission electron microscope operated at either 80 or 120 kV, respectively.

## Acquisition and calculation of tomograms

Serial sections of the selected cells were then transferred to a TECNAI F30 transmission electron microscope (Thermo Fisher Scientific, USA) operated at 300 kV and equipped with a US1000 CCD camera (Gatan, USA). Using a dual-axis specimen holder (Type 2040, Fishione, USA), tilt series were acquired from –65° to +65° with 1° increments at a magnification of 4700 x and a final pixel size of 2.32 nm applying the SerialEM software package (*Mastronarde, 2005*; *Mastronarde, 2003*). For double-tilt electron tomography, the grids were rotated for 90 degrees and the second tilt series were acquired using identical microscope settings (*Mastronarde, 1997*). The tomographic A- and B-stacks were combined using IMOD (*Kremer et al., 1996*; *Mastronarde and Held, 2017*). For each spindle reconstruction, montages of 2×3 frames were collected. Depending on the orientation of the spindles during the sectioning process, between 22 and 35 serial sections were used to fully reconstruct the volumes of the three selected spindles (*Table 9*).

## Segmentation of MTs and stitching of serial tomograms

As previously published (*Redemann et al., 2014*; *Weber et al., 2012*), MTs were automatically segmented using the ZIB Amira (Zuse Institute Berlin, Germany) software package (*Stalling et al., 2005*). After manual correction of MT segmentation, the serial tomograms of each recorded cell were stitched using the segmented MTs as alignment markers (*Lindow et al., 2021*) Following this pipeline of data acquisition and 3D reconstruction, three complete models of HeLa cells in metaphase were obtained (*Table 9*). As also done in our previous study on mitosis in *C. elegans* (*Redemann et al., 2017*), we discarded MTs with one endpoint found within 100 nm from the border of a reconstructed tomogram. With high probability, these MTs were leaving the tomographic volume. These discarded MTs account for <1% of all traced MTs in all datasets. Therefore, we do not expect a relevant error in this analysis.

## Z-correction of stacked tomograms

Each stack of serial tomograms was expanded in Z to correct for a sample collapse during data acquisition (*McEwen and Marko, 1998*). We corrected this shrinkage by applying a Z-factor to the stacked tomograms (*Figure 2—figure supplement 2*; *O'Toole et al., 2020*). Taking the microtome setting of 300 nm, we multiplied this value by the number of serial sections. For each spindle, we also determined the thickness of each serial tomogram and then calculated the total thickness of the reconstruction. The Z-factor was then determined by dividing the actual thickness of each stack of tomograms by the total thickness as determined by the microtome setting. Such calculated Z-factors (1.3 for spindle #1, *Figure 2B, C and F*; 1.4 for spindle #2, *Figure 2D and G*; and 1.42 for spindle #3, *Figure 2E and H*) were then applied to our full spindle reconstructions. All quantitative data in this publication are given for the Z-expanded spindles. For comparison, values for the non-expanded spindles are also given in *Table 12* and *Table 13*.

## Software packages

We used the ZIB extension of the Amira software (Zuse Institute Berlin, Germany) for further quantitative analyses (*Stalling et al., 2005*). In addition, an automatic spatial graph analysis (ASGA) software tool was created for the quantification of KMT length and minus-end distribution (*Kiewisz and Müller-Reichert, 2021*). The ASGA software tool was also used to quantify the position of each k-fiber in the mitotic spindles and determine the tortuosity, the cross-section area, the shape and the density of KMTs in the k-fibers and the MT-MT interactions.

## Staging of spindles

For staging of the three reconstructed metaphase spindles, we determined the inter-kinetochore distance for each k-fiber pair. More precisely, we analyzed the distance between the paired outer kinetochores. For this, the closest neighboring sister kinetochores were determined. The center of each kinetochore was then defined as a median position of all KMT plus ends associated with each selected kinetochore, and the inter-kinetochore distance was then calculated as the 3D distance between the defined median centers of each kinetochore pair. For each mitotic spindle, the inter-kinetochore distance is given as the mean value (±STD). As an additional criterion for mitotic staging, the pole-to-pole distances were measured. For this, we analyzed the 3D distance between the centers

**Table 12.** Quantification of KMT ultrastructure before and after application of Z-expansion to the 3D models.

| Data set | Length of KMTs [µm]* | | Length of non- KMTs [µm]* | | No. of KMTs per kinetochore* | | No. of KMTs in the MT-centrosome interaction area* | | Mean KMT minus-end distance to poles [µm] | | No. of KMTs associated with poles [%] | | No. of non-KMTs associated with poles [%] | |
|---|---|---|---|---|---|---|---|---|---|---|---|---|---|---|
| | **Before** | **After** | **Before** | **After** | **Before** | **After** | **Before** | **After** | **Before** | **After** | **Before** | **After** | **Before** | **After** |
| **Spindle #1** | 3.23 (±1.49) | 3.59 (±1.57) | 2.03 (±1.6) | 2.13 (±1.67) | 8.04 (±1.86) | 8.04 (±1.86) | 4.1 (±1.8) | 5.0 (±1.8) | 1.16 | 1.72 | 62.2 | 61.2 | 44.5 | 44.3 |
| **Spindle #2** | 3.69 (±1.87) | 3.82 (±1.97) | 1.85 (±1.55) | 1.95 (±1.60) | 9.75 (±2.18) | 9.75 (±2.18) | 2.4 (±2.0) | 3.1 (±2.3) | 2.47 | 2.87 | 53.6 | 31.5 | 28.8 | 28.6 |
| **Spindle #3** | 4.03 (±1.79) | 4.27 (±1.93) | 1.91 (±1.80) | 2.07 (±1.93) | 7.49 (±1.91) | 7.49 (±1.91) | 3.4 (±1.8) | 4.1 (±2.0) | 1.35 | 2.12 | 62.0 | 54.2 | 42.3 | 41.9 |

*Numbers are given as mean ±STD.

**Table 13.** Quantification of k-fiber organization before and after application of Z-expansion to the 3D models.

| Data set | Density of KMTs at the kinetochore [KMT/μm²]* | | KMT-KMT distance at the kinetochore [nm]* | | Global tortuosity of KMTs* | | % of curved KMTs* | | Area of k-fibers [μm²]* fibers* | | % of KMTs in a k-fibers* | |
|---|---|---|---|---|---|---|---|---|---|---|---|---|
| | **Before** | **After** | **Before** | **After** | **Before** | **After** | **Before** | **After** | **Before** | **After** | **Before** | **After** |
| **Spindle #1** | 151 (±74) | 122 (±62) | 61 (±11) | 67 (±20) | 1.09 (±0.10) | 1.11 (±0.11) | 36.1 | 39.8 | 0.063 (±0.09) | 0.08 (±0.1) | 34 (±27) | 64 (±27) |
| **Spindle #2** | 137 (±68) | 99 (±45) | 65 (±12) | 78 (±23) | 1.06 (±0.06) | 1.07 (±0.07) | 21.4 | 28.4 | 0.068 (±0.10) | 0.09 (±0.11) | 70 (±25) | 70 (±25) |
| **Spindle #3** | 175 (±123) | 117 (±72) | 66 (±12) | 76 (±23) | 1.11 (±0.11) | 1.13 (±0.13) | 39.5 | 47.1 | 0.080 (±0.15) | 0.12 (±0.24) | 59 (±39) | 59 (±29) |

*Numbers are given as mean ±STD.

of the manually segmented mother centrioles in each data set. This read-out was used to determine the spindle size at metaphase.

## Classification of MTs

MTs with their putative plus end associated with the chromosomes were defined as KMTs (*Figure 2—figure supplement 2*). Characteristically, these KMTs showed a parallel arrangement at the site of attachment to the chromosomes. Unfortunately, identification of individual kinetochores in our electron tomograms was hindered by the fact that prominent single and electron-dense KMT attachment sites, as described previously for conventionally fixed cells (*McEwen et al., 1998b*), were not always clearly visible after cryo-fixation by high-pressure freezing. All other MTs in our 3D reconstructions were classified as non-KMTs.

## MT-centrosome interaction area

For each non-KMT, the end closest to the nearest mother centriole was defined as the minus end. The absolute distance of each putative non-KMT minus end to the nearest mother centriole was measured in 3D. The number of the non-KMT minus ends was then plotted against their distance to the pole. We then fit a Gaussian distribution to the non-KMT minus-end density. We also defined the peak of the Gaussian distribution to determine its half-width. The border of spindle poles, termed here the border of the MT-centrosome interaction area, was defined as twice the half-width, which was 1.7 μm from the centrosome.

## Position of MT minus ends

To analyze the position of KMT and non-KMT minus ends in the metaphase spindles, two measurements were performed. Firstly, the 3D distance between the nearest mother centriole and the KMT and the non-KMT minus ends was determined. Secondly, the relative position of these ends on the pole-to-kinetochore and the pole-to-pole axis was determined. For each KMT minus end, the relative position is given as the normalized position between the mother centriole (position = 0) and the kinetochore (position = 1; *Figure 5—figure supplement 1*). For each non-KMTs minus end, the relative position is given as the normalized position between two spindle poles (pole1 = 0, and pole2 = 1; *Figure 5A*). The distribution of the relative positions of KMT and non-KMT minus ends (mean ±STD) is given for each data set. The number and percentage of KMT and non-KMT ends not associated with the spindle pole were defined as minus ends detected farther than the calculated MT-centrosome interaction area. To visualize an approximated MT-centrosome interaction area on both the pole-to-kinetochore and the pole-to-pole axis, we defined the relative position of the average border of this interaction area. The average border of this interaction area was defined as the average relative position of all KMTs and ranged from –0.2 to 0.2.

## Length distribution of MTs

The full length of each reconstructed KMT and non-KMT was measured, and the average (±STD) is given for each data set. We also analyzed the percentage of short *versus* long KMTs. For each data set, short KMTs were defined as those shorter than 1.7 μm in length. This threshold was chosen based on the MT-centrosome interaction area. Long KMTs were identified as KMTs longer than the half-spindle length for each given data set.

## Defining kinetochore position

To determine the position of each k-fiber in the mitotic spindle, a position model was created that is based on the location of each kinetochore on the metaphase plate. For this, the kinetochores of each spindle were projected in 2D space on the X/Z axis and an ellipse with a semi-major (called a-axis) and a semi-minor axis (called b-axis) was fitted onto all projected kinetochores. The fitted ellipse was then divided into three regions ranging from 0 to 50% (central region), 50 to 75% (intermediate region), and 75 to 100% (peripheral region). Kinetochores with associated k-fibers were then assigned to these three regions.

## Global tortuosity of KMTs

For the analysis of global KMT tortuosity, the ratio of the KMT spline length and the 3D distance between the plus and the minus end for each KMT was measured. The distribution of KMT tortuosity (mean ±STD) is given. In addition, the correlation of the tortuosity of KMTs with their length is given as a fitted polynomial line calculated as a local polynomial regression by the locally estimated scatterplot smoothing 'loess' method. A confidence interval for the created polynomial line was calculated with the t-based approximation, which is defined as the overall uncertainty of how the fitted polynomial line fits the population of all data points. Local polynomial regressions and confidence intervals for all data sets were calculated using the stat 4.0.3 R library (*R Development Core Team, 2021*).

## Local tortuosity of KMTs

For the calculation of the local tortuosity, each KMT was subsampled with segments of a length of 500 nm. Both the tortuosity and the relative position along the pole-to-kinetochore axis were measured for each segment. In addition, the correlation of local KMT tortuosity against the relative position is given. Local polynomial regressions and confidence intervals for all data sets were calculated using the stat 4.0.3 R library (*R Development Core Team, 2021*).

## The polygonal cross-section area of k-fibers

The cross-section area was calculated every 500 nm along each k-fiber. For each defined k-fiber cross-section, the KMT positions were mapped on a 2D plane, and the polygonal shape of the k-fiber cross-sections was calculated based on the position of the KMTs. The polygonal shape was calculated with the alpha shape algorithm ($\alpha$ = 10) using the 'ashape3d' function of the alphashape3d 1.3.1 R library (*Lafarge and Pateiro-Lopez, 2020*). The alpha shape is the polygonal shape formed around a given set of points (KMTs from a cross-section) created by a carving space around those points with a circle of a radius defined as $\alpha$. The polygonal shape was then built by drawing lines between contact points. In order to calculate the area from the polygonal shape of a k-fiber cross-section, a polygonal prism was created by duplicating and shifting a polygonal shape 1 µm in the X/Y/Z dimension. This created a prism with a height of 1 µm. The volume of the created 3D object (prism) was then calculated using the alphashape3d 1.3.1 R library (*Lafarge and Pateiro-Lopez, 2020*). From this, a polygonal area could be calculated by dividing the prism volume ($V_{pp}$) by prism high ($h_{pp}$ = 1 µm). The distribution of the k-fiber polygonal area along the pole-to-kinetochore axis is given as a fitted polynomial line of local polynomial regression using the 'loess' method. Confidence intervals were calculated with the t-based approximation using the stat 4.0.3 R library (*R Development Core Team, 2021*).

## Density of KMTs in k-fibers

The density of KMTs in the k-fibers was calculated in segments of 500 nm length along the entire path of each fiber. To determine the percentage of KMTs that were enclosed in the k-fiber for each cross-section, the number of KMTs enclosed in the given k-fiber section and the circular area were determined. The radius of the circular area was calculated for each k-fiber at the position of KMT attachment to the kinetochores. The distribution of the k-fiber density along the pole-to-pole axis is given as a fitted polynomial line and a confidence interval calculated with the t-based approximation using the stat 4.0.3 R library (*R Development Core Team, 2021*).

## Interaction of KMTs with non-KMTs

A possible association between KMT minus ends and other MT lattices was measured by calculating the 3D distance between KMT ends and every MT lattice in the reconstructed spindle. An interaction between KMT minus ends and a MT lattice was identified when KMT minus ends were found within a given interaction distance to any MT lattice. The defined interaction distances were 25, 30, 35, 45, 50, 75, and 100 nm. To account for differences in the density of MTs along the pole-to-pole axis, each KMT interaction was normalized by calculating the local MT density around each KMT end. This was achieved by selecting a voxel of 0.001 µm$^3$ with the KMT end in its center and calculating the local MT density by dividing the number of potential interactions by the voxel volume. For visualization, each KMT was labeled based on the type of detected interaction with KMTs or non-KMTs. KMTs without any interaction were also labeled. The percentage of KMTs with any interaction was measured and the average value for all data sets is given (mean ±STD).

To identify possible MT minus-end associations with KMT lattices, the 3D distances of the MT minus ends to KMT lattices were calculated. An association between MT minus ends and KMT lattices was detected when MT minus ends were positioned within defined interaction distances to the KMT lattices. Again, we considered the following interaction distances: 25, 30, 35, 45, 50, 75, and 100 nm. In addition, each interaction was normalized by the local MT density, as described above. The percentage of KMTs with any interaction was measured and the average from all datasets is given (mean ±STD).

To analyze the position of MT-MT associations, the relative position of MT minus ends on the pole-to-kinetochore axis was calculated. The relative position of each minus end is given as the position between the kinetochore (position = 1) and mother centriole (position = 0) along the spindle axis, normalized by the MT density.

## Analysis of KMT-KMT distances

The KMT-KMT distances at given k-fiber cross-sections were measured by a K-nearest neighbor estimation. An estimation was achieved by calculating a distance matrix between all selected KMTs. Each KMT-KMT connection was ranked according to its distance. Finally, for each KMT in a k-fiber, the closest KMT neighbors were selected. For each k-fiber, the mean KMT-KMT distance and the standard deviation were calculated.

## Interaction of MTs

The interaction between MTs was calculated in steps of 20 nm along each MT. For each MT segment, the distance to a neighboring MT was calculated. In addition, the length of interaction was analyzed for each detected MT-MT interaction. The length of interaction between MTs was calculated as a sum of the 20 nm segments. This analysis was performed for defined interaction distances of 25, 30, 35, 45, and 50 nm. The frequency plots for the average number of interactions per MT and the average length of interaction are given for each interaction distance. Each MT segment is labeled based on the number of interactions.

## Error analysis

For the tracing of MTs, the error associated with our approach was previously analyzed for the 3D reconstructions of mitotic centrosomes in the early *C. elegans* embryo using serial semi-thick plastic sections (*Weber et al., 2012*). Although the data on mammalian spindles is larger, the tomogram content of this current study is similar to the published centrosome data sets, and thus we assume that the error MT tracing lies in the same range of 5–10%. All traced MTs were manually verified. This was achieved by using the 'filament editor' tool in the ZIB extension of the Amira software that allowed us to create a flattened overview of the entire MT track, which was instrumental for quick validation of each MT. Both false-positive and negative tracings were corrected.

However, it is more difficult to estimate the error of the matching algorithm. Our standardized automatic stitching method has been described in detail in previous publications (*Lindow et al., 2021*; *Redemann et al., 2014*; *Weber et al., 2012*). In general, the stitching depends on the local density and properties of the MTs. For this reason, the stitched MTs were manually verified and corrected (*Lindow et al., 2021*). In particular, all KMTs in our reconstructions were checked for correct stitching across section borders. Examples of correct stitching of MTs at section borders are given in *Figure 3—figure supplement 1*. The quality of the analysis of the MTs, especially the KMTs, should therefore be influenced by minor errors. In our previous publications (*Redemann et al., 2014*; *Weber et al., 2012*), we estimated the overall quality of the stitching by analyzing the distribution of MT endpoints in the Z-direction (i.e. normal to the plane of the slice). We expect to find approximately the same density of MT endpoints along the Z-direction of each serial-section tomogram. This distribution is visualized in the *Serial Section Aligner* tool previously presented (*Lindow et al., 2021*). Therefore, if the density of endpoints after matching is approximately the same along the Z-direction of the serial-section tomograms, we can assume that the number of artificial points that have been introduced at the interfaces of the serial sections are negligible. This was visualized by projecting each spindle along the Y/Z axis (*Figure 3—figure supplement 1*).

## Custom-designed software for the visualization of 3D data

For better visualization of the 3D organization of KMTs in k-fibers, a platform was developed using the WebGL library (rgl 0.106.8 R library; *Adler et al., 2021*). This platform was implemented for the public and will allow readers to choose data sets from this publication for an interactive visualization of selected spindle features. For instance, users may choose to visualize the organization of k-fibers or KMTs and select for the analysis of MT-MT interactions. For an analysis of KMTs, users can select the following features of analysis such as length distribution, minus-end positioning, curvature, and number at the kinetochore. For the MT-MT interaction analysis, users can select different interaction distances. This platform is designed for the continuous addition of 3D reconstructions of spindles obtained from different systems and can be accessed as follows: https://cfci.shinyapps.io/ASGA_3DViewer/.

## Data availability

Tomographic data before and after the z-expansion has been uploaded to the TU Dresden Open Access Repository and Archive system (OpARA) and is available as open access: http://doi.org/10.25532/OPARA-128; http://dx.doi.org/10.25532/OPARA-177.

We released all datasets in Amira format. The tomographic data are also available in tiff format, which can be opened either with the ImageJ Fiji (*Schindelin et al., 2012*) or the IMOD (*Kremer et al., 1996*) open-source software packages. The MT-track files containing information about the segmented MTs were released in binary and ASCII format. To make this task easier for interested readers, the ASGA (*Kiewisz and Müller-Reichert, 2021*) open-source software, which is part of this publication, is supplied with small scripts written in R language, which allows users to read the ASCII format into an array. https://github.com/RRobert92/ASGA/blob/main/R/bin/Utility/Load_Amira.R.

The code used to perform quantitative analysis and visualization of MT organization in spindles has been uploaded to the GitHub repository and is available as open access under the GPL v3.0 license: https://github.com/RRobert92/ASGA; https://github.com/RRobert92/ASGA_3DViewer.

The supplementary high-resolution videos have also been uploaded to YouTube. l These movies can be found at this URL: https://youtube.com/playlist?list=PL-L6a60L11laVrVBFZqGi0wmULXD1b4Px.

## Acknowledgements

The authors thank Dr. Tobias Fürstenhaupt (Electron Microscopy Facility at the MPI-CBG, Dresden, Germany) for technical support. We are also grateful to Drs. Reza Farhadifar, Stefanie Redemann, Alejandra Laguillo Diego and Isabelle Vernos for a critical reading of the manuscript. We also thank Felix Herter for implementing a module in the ZIB extension of Amira that we used in this paper for visualizing different cross-section plains of KMT ends. Research in the Müller-Reichert laboratory is supported by funds from the Deutsche Forschungsgemeinschaft (MU 1423/8–2). RK received funding from the European Union's Horizon 2020 research and innovation program under the Marie Skłodowska-Curie grant agreement No. 675737 (grant to TM-R). This work was supported by the NSF-Simons Center for Mathematical and Statistical Analysis of Biology at Harvard (award number #1764269), and the Harvard Quantitative Biology Initiative.

## Additional information

### Funding

| Funder | Grant reference number | Author |
|---|---|---|
| Deutsche Forschungsgemeinschaft | MU 1423/8-2 | Robert Kiewisz<br>Gunar Fabig<br>Thomas Müller-Reichert |
| Horizon 2020 Framework Programme | 675737 | Robert Kiewisz<br>Thomas Müller-Reichert |
| Harvard University | | William Conway<br>Daniel Needleman |

| Funder | Grant reference number | Author |
|---|---|---|
| Nick Simons Foundation | 1764269 | William Conway<br>Daniel Needleman |

The funders had no role in study design, data collection and interpretation, or the decision to submit the work for publication.

## Author contributions

Robert Kiewisz, Conceptualization, Resources, Data curation, Software, Formal analysis, Validation, Investigation, Visualization, Methodology, Writing – original draft, Writing – review and editing; Gunar Fabig, Formal analysis, Validation, Methodology, Writing – review and editing; William Conway, Data curation, Formal analysis, Validation, Visualization; Daniel Baum, Validation, Writing – review and editing; Daniel Needleman, Conceptualization, Formal analysis, Writing – original draft, Writing – review and editing; Thomas Müller-Reichert, Conceptualization, Supervision, Funding acquisition, Writing – original draft, Project administration, Writing – review and editing

## Author ORCIDs

Robert Kiewisz (ID) http://orcid.org/0000-0003-2733-4978
Gunar Fabig (ID) http://orcid.org/0000-0003-3017-0978
William Conway (ID) http://orcid.org/0000-0001-7532-4331
Thomas Müller-Reichert (ID) http://orcid.org/0000-0003-0203-1436

## Decision letter and Author response

Decision letter https://doi.org/10.7554/eLife.75459.sa1
Author response https://doi.org/10.7554/eLife.75459.sa2

# Additional files

## Supplementary files

• MDAR checklist

## Data availability

All Datasets were uploaded and ara available in OpaRA server: http://doi.org/10.25532/OPARA-128; http://doi.org/10.25532/OPARA-177. The code used to perform quantitative analysis and visualization of MT organization in spindles has been uploaded to the GitHub repository and is available as open access under the GPL v3.0 license: https://github.com/RRobert92/ASGA; (copy archived at swh:1:rev:142dcb882134954b9dc98f26044dd04a3893f181); https://github.com/RRobert92/ASGA_3DViewer, (copy archived at swh:1:rev:7594a85728f7fce05562d4f75fefee4d0f1935e4).

The following datasets were generated:

| Author(s) | Year | Dataset title | Dataset URL | Database and Identifier |
|---|---|---|---|---|
| Kiewisz R | 2021 | HeLa metaphase spindle tomographic data sets and analysis | https://opara.zih.tu-dresden.de/xmlui/handle/123456789/1970 | OpaRA, 10.25532/OPARA-128 |
| Kiewisz R | 2022 | HeLa metaphase spindle tomographic data sets and analysis for z-expansion data | https://opara.zih.tu-dresden.de/xmlui/handle/123456789/5675 | OpaRA, 10.25532/OPARA-177 |

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
