## [Editor Report]

This paper will be an incredible resource for cell biologists. The authors use sophisticated reconstructions of kinetochore–fibers within human metaphase spindles using electron tomography and then analyze their ultrastructure and organization. The findings lead to compelling models with clear implications for kinetochore–fiber and spindle self–organization.

---

## [Decision Letter]

**Decision letter after peer review:**

Thank you for submitting your article "Three–dimensional structure of kinetochore–fibers in human mitotic spindles" for consideration by *eLife*. Your article has been reviewed by 3 peer reviewers, and the evaluation has been overseen by a Reviewing Editor and Anna Akhmanova as the Senior Editor. The following individual involved in review of your submission has agreed to reveal their identity: J Richard McIntosh (Reviewer #1).

Essential revisions:

1) The authors do not take account of the collapse in section thickness that occurs when plastic–embedded samples are treated with the beam of an electron microscope. This shape–change is not isotropic, because the film upon which the sections sit shrinks little, if at all, supporting the dimensions of the embedded material in two dimensions. The dimension perpendicular to the section plane is, however, unsupported, and the section collapses along that axis by about 40%. There is confidence that this situation pertains for the current data for two reasons. The full metaphase spindles, seen in cross–section, appear either elliptical or flattened, whereas both EM cross–sections of HeLa metaphases and light microscopic images of living spindles viewed down their axes are circular. Second, each kinetochore appears elliptical in this study, whereas in previous work with spindle cross–sections, human kinetochores are circular. The ellipticity of both the kinetochore and the metaphase plates is just about the degree expected, given the 30 – 40% collapse that others have measured for ET of plastic embedded material. The potential error caused by this distortion must be resolved in the paper. One way to address this would be to apply the asymmetric expansion to their model necessary to make it a better representation of an uncollapsed spindle, as other have previously done, then recalculate all their near–neighbor distributions on the data that have been brought into true. While we do not insist on this for this quite large amount of work, the authors must resolve this artefact in their reconstruction.

2) Another issue that should be considered is that the preference for KMTs to have non–KMTs as neighbors near the pole, which they report, is uncorrected for two structural issues: the increased density of MTs near the poles, which is a natural result of the essentially spherical shape of the metaphase spindle, and the fact that there are many more non–KMT in this region of the spindle than KMTs, so naturally KMTs have more non–KMTs as neighbors. This situation says nothing about the possibility of MT–MT interactions unless the proximity data are normalized for these two considerations.

3) When the observer is excising the tiny wafer of plastic that contains the cell for study from the thin layer of plastic in which it is embedded, it is all too easy to bend the plastic. This distorts the biological material in the plastic and may contribute to the torsion observed in the 3D reconstructions of spindle MTs. This small but important issue deserves mention.

4) The "textbook" models proposed and challenged in the text are not representative of the common understanding in the state of the field and the authors do not do enough to discuss the implications of their results or fit their results in the context of previous literature. Previous EM studies in mammalian spindles have already shown that k–fibers form semi–direct connections to the pole. The authors could instead emphasize other key findings in the paper, or provide more context as to why their main point about a semi–direct connection between kinetochores and poles in human cells is novel and exciting.

5) HeLa spindle lengths were previously reported to be ~12 µm, however, here they are reported to be significantly shorter in all spindles analyzed. The authors should clarify the origin of this discrepancy.

6) Figures 5, 7, and 8 should be better motivated. It would also help to discuss implications of findings, and to add summaries/models to the text or figure.

7) The paper lacks any discussion or estimation of potential errors –– tracking MTs or erroneous end declared/missed. An estimation/discussion of the errors should be provided so that the reader knows how much confidence there should be in this analysis.

8) The authors need to provide a better explanation of the use of the term K–fiber. The paper uses the term K–fiber to refer to the bundle of KMTs. This is making an unjustifiable assumption. The paper demonstrates that the KMTs often do not retain a tight cluster in going from KT to pole through premature termination of the KMTs and splay. Is this then the K–fiber? Is this consistent with the known structural integrity (eg under laser cutting) and high stability (cold stable) of K–fibers? An alternative model would be that the K–fiber is a tight MT bundle that continues to the pole with a changing composition of MTs along its length. The loss of KMTs from the bundle is then simply MT exchange with the spindle. The analysis is unclear on this point – proximity to MT lattices of the KMTs has weak dependence on distance along the spindle axis, suggesting bundles are present, but this needs more analysis.

9). How are so many minus ends generated in the spindle – 50% for KMTs, 30% for non–KT MTs. These terminated MTs are presumably crosslinked, and the fast dynamics of astral MTs suggests they must be nucleated in the spindle itself. This should be discussed.

10) Inclusion of more EM images to back up and support the conclusions.

Please also take note of specific points raised by individual reviewers below.*Reviewer #1 (Recommendations for the authors):*

Abstract: While K–fibers do connect kinetochore directly with the poles in many cells, in others the k–fiber connects with other spindle MTs, rather than the material at the pole. Even in cells with clearly defined poles, connection with other MTs may contribute to spindle mechanics (as is shown in this study). Therefore, it might set the reader's mind on the right course by replacing the first sentence with something like, "… provide a physical linkage between the chromosomes and the rest of the spindle."

Ln 112 I think you mean "Figure 2–movie supplements 1–3".

Ln 114 The observations of 14% of the MTs being KMTs, but those MTs accounting for 24% of the tubulin mass make an interesting discrepancy. I presume the difference in these numbers comes from the fact that KMTs are, on average longer than non–KMTs, but whatever the reason, it should be explained.

Figure 2 Supplement 1 Note that the legend to this figure is a repeat of the legend for figure 2, and there is no legend for this Figure In Figure 3B you call these kinetochores sisters 1 and 2. If you are confident that they are a pair, it might be more meaningful to your readers to call this pair of kinetochores sisters #1 and call the individual kinetochores 1A and 1B. I certainly don't insist on this nomenclature change, but it might help readers to follow you.

Figure 2 Supplement 2 In the legend (ln 884) you use the term "cross–section" to refer to tomographic slices that cut through a MT parallel to its axis. I find this confusing, because the cross–section of a MT is usually a slice cut perpendicular to the MT axis. Rephrase?

Ln 116 Were the Kinetochore–proximal ends of ALL KMTs flared, as shown? In previous publications many KMT plus ends were flared, but not all.

Ln 119 O'Toole et al. have examined the pole–proximal ends of KMTs in *C. elegans* and found that about half of these ends are open, half appeared to be capped, perhaps by the γ–tubulin ring complex. I wonder of your data would allow you to make comparable observations on KMTs in Hela cells? Mammalian spindle MTs flux toward the spindle pole, whereas comparable data for the C.e. embryonic spindle are (to my knowledge) not available. It would be very interesting to know if KMT minus ends in HeLa are capped or not. If capped, how can they flux?

Ln 120 Since there is no legend for Figure 2 Sup 1, I am making an inference about panel C on that figure, but it looks as if you use the 3D positions of each KMT associated with a single kinetochore to find a 3D average position, then call that the position of the kinetochore. This is not strictly true, because the plus ends of the KMTs are not AT the kinetochore, they commonly end outside the outer plate in the corona material. It would be valuable to find a wording that states more explicitly what you are doing, because the distances you present for the spacings of sister kinetochores are likely to be bigger than those measured by others with different definitions of the kinetochore.

Ln 128 "Site facing the pole" is imprecise and undefined.

Ln 134 and following. If I have understood you correctly, you estimated the size of each kinetochore by the area occupied by the KMTs that ended at approximately on region of the spindle because you did not see extra staining density in the chromatin or some other structural indicator of the kinetochore itself. That's OK, but you then go on to calculate the correlation between the number of KMTs and the size of the area they occupy and get a great correlation. But this is not meaningful. Of course, you get a good correlation because you are estimating the size of the kinetochore by the area occupied by the MTs. You then say that this is in good agreement with Cherry's observations on HeLa by LM, without saying what Cherry saw. I find this set of statements unhelpful for understanding K size.

Ln 902 There is something funny about the font for G of Graph and then P of Plot. Similar problems throughout this legend and others, so check the PDF carefully before it is posted.

Ln 903 Panel B does not show the distribution mentioned in the legend. It shows a related fact, which is important, but the distribution of minus ends is the subject of Figure 4. Have you computed it relative to a pole–to–pole axis, or the like? Perhaps relative to distance from the pole? The fact that so many KMTs do actually reach the pole is important and surprising to me. Based on K–fibers in PtK cells, I would have thought that more KMTs would have ended farther from the pole. Interesting.

Ln 910 As indicated by my comments relevant to the text derived from this figure, I don't see much value in this analysis, given the way you defined the position and size of the kinetochores.

Ln 160 and following. Your definition of the distance from the pole within which you call MT ends pole–interacting is clear, but your reasons for choosing this distance are not. In a sense this is a discussion point, and so how either you or Steffie decided this was the right distance might best be presented in that part of the paper. However, something should be said about it here, because in its current form, the distance chosen sounds completely arbitrary. Perhaps add a discussion paragraph and say here that this point is clarified in the discussion?

Ln 919 The end markers are circles, not spheres.

Ln 923 You map non–KMT end positions onto the pole–to–pole axis, which makes sense. You say that you map KMT end positions onto a kinetochore–to–pole axis, but that implies that you used a different axis for every kinetochore. Is that true? If so, you should say how these axes were determined, and a figure to show exactly what you mean should be added.

Figures 4D and G: Because you say "distance" in the text and on the abscissa, I presume this is what you have previously called "Euclidian distance", as suggested in 4A. Please be specific here because it is important for thinking about your data. In G, you again say distance, but here, I presume it is distance along the pole–to–pole axis, since that is how you describe it above. Thus, in your current labeling, the word distance means different things in these two panels. If I have this right, you should clean up the terminology. Note that again there are font problems at several places in the PDF.

Ln 942 Here again distance of KMT minus ends to poles is referred to simply as "distance". If this is a 3D Euclidian distance, as previously implied please change wording. Now in panel C, you are presenting distances as mapped onto the spindle axes, so these should be directly comparable to the distances you present for nonKMTs. If I'm right about this, I think you should say so explicitly. For example, Panels C and E are directly comparable, and the penetration of KMTs beyond the position of the mother centriole is nicely demonstrated. But now you state that E represents real distances (Euclidian?) and F is position on the axes. This implies that you measured distances from minus ends of nonKMTs to the centriole, which has not previously been mentioned. This needs clarification and explicit description with clearer words. Moreover, the shapes of these plots shown in Figure 4 sup1 F are remarkable for their symmetry, and this deserves comment and explanation.

Ln 180 This is the only reference to Figure 4 supple. 2, which is an interesting figure. I wonder why this information doesn't get more attention in the text.

Also relevant to this figure: here better than anywhere, one sees the elliptical shape of the cross sections of your spindle reconstructions. Given what one knows from light microscopy, this is almost certainly an artifact of electron tomography. I presume it is a result of the collapse of section thickness in the beam of the microscope. Some mention and explanation of this feature of your reconstructions belongs in the paper somewhere.

Ln 196 I find this title to be a bit of an exaggeration. Splay implies really flaring out. Yes, the area of the k–fiber encompassing polygons increases toward the pole, but I would describe it as a widening, not flaring or splaying.

Ln 225 You are using polygon here and below as an adjective. Should be polygonal.

General comment on the study of MT associations. Some of these statements and graphs need a bit more thought. Figures 7B and 7D show that the greater the distance between an MT end and an MT wall that is considered to be an "interaction", the greater the number of interactions there are. This point is almost trivial; it must be true. I don't see that it is worth two graphs. Said another way, the looser the criterion use for MT–MT interaction, the more interactions are seen. I don't think this needs to be demonstrated.

Ln 245 It might be worth mentioning MT–interacting proteins that are not motor enzymes, since some of these have been shown to be important for spindle structure and function.

Ln 255 The term association distance is introduced without any intellectual background. This may be a bit cryptic for people who have not been reading papers about spindle MTs and the ways structural studies have tried to identity MT–MT interactions. Perhaps add a sentence that rationalizes your interest in "interaction distance"? Perhaps a reference to others who have used this approach?

Ln 261 You say that most of these interactions occur near the pole, and you also say that many of the KMTs end before getting near the pole. Thus, the 41.4% of KMTs that have an end interacting with the wall of a non–KMT might be quite a large fraction of the KMTs that get into the polar area and are therefore in a position to have this kind of interaction. Can you look at this and clarify the point? For example, if you find a distance from the mother centriole that defines a hemisphere in which MT–MT distances are small, what fraction of the KMTs that get into that region have a tip interaction with a non–KMT.

Ln 270 and following. The observation that most of the MT–MT interactions (i.e., sites of close proximity) occurred near the spindle pole might be important, but a qualitative look at the 3D reconstructions suggests that the MT density is greater near the poles than farther away. By this I mean in a given spindle cross–section, how many MTs are there/unit area. If this number is higher near the poles, then the MTs must be closer to one another, so the increase in the number of MT proximities is a necessity, not a novel fact. This issue should be considered in the presentation of these data. Perhaps you could compute the MT density as a function of position along the spindle, then use this number at each position to normalize the distributions of interactions, thereby asking whether there is something in the data to indicate that the MTs are binding with one another, rather than simply being packed in. Also, the mechanical significance of KMT ends near KMTs is not clear to me.

Ln 276 You say, "as a result", but that phrase could be omitted. You are just describing what you saw.

Ln 290 Note that the number of non–KMTs/spindle cross–section exceeds the number of KMTs/cross section, and this is particularly true near the spindle poles. Thus, the fact that there are more associations between KMT ends and non–KMTs than with KMTs may simply be a consequence of this population difference. This could be calculated on the bases of the relative densities of the two kinds of MTs.

Ln 295 The fact that association frequencies are higher near the pole regardless of association distance may again be based simply on MT density.

Ln 296 In this context, I'm particularly interested in Figure 7, supple. 1E. This distribution of interactions between the walls of KMTs and the ends of non–KMTs does show its biggest peak at the pole, as stated, but there is a good indication of a second peak, not far from the kinetochores. This peak would be clearer if the data were presented as a running average, since the fluctuations seen make the bar graphs very spikey. Moreover, if the normalization, based on MT density were applied, it would reduce the polar peak but not the rise in frequency that is seen nearer to the kinetochores. As you know, this position along the spindle was identified structurally by O'Toole et al. as a place where the minus ends of overlapping MTs from the interzone (a subset of your non–KMTs) made association with the walls of KMTs. It is also the place where two groups using microbeams have identified places where KMTs receive a push from the spindle interzone. Thus, the gentle rise in your data may be very significant for the functional interactions between different classes of spindle MTs.

Ln 297 This discussion sounds just like what Mastronarde and others have called pairing length in their studies of yeast spindles. It might be well to use similar terminology to minimize confusion. Also, I think there might be a clearer way to describe what you have done. Here is my effort: "To measure the paraxial length over which MTs interacted, we defined 20 nm along the axis as a minimal length of interaction. For each pair of MTs that lay within a chosen interaction distance, we measured the number of 20 nm segments over which they retained this proximity."

Ln 305 I'm a little uncertain about what the words here mean. Are the "peaks of association" peaks in the number of 20 nm paring lengths that describe the pairing distances? Please clarify.

Figure 8 It seems as if the legends to 8B and C are switched.

Ln 305 and following. These words don't do justice to the very interesting distribution graphs you show in Figure 8. In the text, you don't discriminate between the classes of MTs that you break out in your graphs. For example, the words here describe what is shown in Figure 8B but not 8D. Please be specific about which MTs your words refer to.

Ln 309 You see no "hotspots" in your data for interactions between non–KMTs, but in Figure 8D there is a significant peak near the spindle midplane. (The label on the ordinate of this graph says non–KMTs interacting with all other MTs. I presume that you mean both other non–KMTs and KMTs. If this is not right, please be more specific.) Granted, the peak is higher for greater interaction distances, but studies in fission yeast suggest that the mean distance between MTs in the zone of overlap is greater than near the poles, and this the region where kinesin motors generate pushing forces. This peak may be of functional significance, and I think deserves mention in the text.

Ln 327 The term "spindle MTs" appears here without definition. Please clarify.

Ln 369 In the text and in Figure 9 it seems to me that you are under–representing the extent to which some KMTs reach the polar area. Your summary diagram shows only one MT of five making to the polar region, whereas both the images you show in the several movies for Figure 3 and your earlier text suggest that the fraction is closer to one–half. This diagram is important, because it is likely to be what gets cited and maybe even reaching textbook figures. Make sure it shows what you have actually seen.

Ln 383 This statement about strong indications of KMTs preferring to interact with non–KMTs is subject to the concern raised above that your data are not corrected for the greater number of non–KMTs. Make sure to do this normalization before making this claim.

Ln 472 No mention is made in the methods section of coping with the fact that plastic sections collapse during the early stages of exposure to the electron beam. Although there is some shrinkage in the plane of the section, the collapse of section thickness can be as much as 40%, and this affects 3–D geometry of the reconstruction. In my opinion, the fact that both the over–all shapes of your spindles and the shapes of each kinetochore fiber are oval, not circular, is a result of this collapse. Some labs expand the dimension perpendicular to the section plane to correct for this collapse. Was this correction applied here? If not, do you think it has impact on the distances observed between spindle MTs? Some discussion would be appropriate.

Ln 535 "was found within the interaction distance" would be clearer.

Ln 1099 The words, "This video shows" could be omitted from each of these legends with no loss.

*Reviewer #2 (Recommendations for the authors):*

Figure 1

–The authors should include the semi–direct connection between kinetochores and poles, since this has previously been shown.

Figure 2

– Figure legend 2C should indicate that microtubule number is in the upper right corner.

Figure 2–supplement 1.

– The figure legend is missing.

– Figure 2–supplement 1B indicates pole–to–pole distances of > 8µm for each spindle, with an average around 10.5 µm. However, Table 1 reports spindle pole distances of ~7, 10, and 9 µm. The authors should clarify this discrepancy.

Figure 3

– Line 140 in the text reports a KMT packing density of 0.07 KMT/µm2, and Table 3 reports similar KMT packing densities (0.06–0.09 KMT/µm2). However, Figure 3–supplement 1E reports KMT densities in the range of 25–75 KMTs/µm2. The correlation coefficient on lines 137–8 also doesn't match the correlation coefficient on Figure 3–supplement 1D.

– Authors should clarify the motivation for comparing inter–kinetochore distance and the number of MTs at the kinetochore, and what previous evidence might suggest a correlation (Figure 3C).

– Authors should define "neighborhood KMT–KMT" in the figure legend for Figure 3–supplement 1F.

Figure 4

– The authors make several conclusions about the composition of k–fiber bundles, for example, that k–fiber bundles are made up of varying KMT lengths. However, they only present gross KMT data. To directly conclude that individual bundles are made up of varying KMT lengths, they should show their KMT data grouped by k–fiber bundle.

– Relating to Figure 4–supplement 2N, the authors could explain the significance of finding more KMT minus–ends at spindle poles in central k–fibers compared to peripheral k–fibers, expanding on what the motivations were for this analysis and what might it say about global k–fiber organization. The authors could also consider plotting KMT length versus kinetochore region.

– The color of the outer kinetochore region does not match the colors of the images or plots, similarly in Figure 5.

Figure 5

– The authors could expand on their motivations for studying tortuosity of KMTs, and speculate on what their findings could imply about spindle organization.

Figure 6

– The polygon area analysis shown in Figure 6–supplement 1D is unclear and difficult to follow. The authors could clarify why this method was necessary.

– Line 229 confusingly states that the polygon area of the k–fiber was 5–fold lower at the pole–facing end of the fibers. It is unclear whether this is relative to the mid–position or to the kinetochore.

Figure 7 and Figure 8

– Some of the text in Figure 7 is too small to read.

– The authors could condense and clarify these figures, as well as compare their findings to previous work on KMT and non–KMT minus–end distributions (O'Toole et al. 2020).

– The authors could discuss more on the key differences found between kMTs and non–kMTs, specifically how the distributions of these two populations are related to their lifetime, organization in the spindle, and mechanistic differences.

– This overall section would benefit from a general summary in the text and model figure(s).

*Reviewer #3 (Recommendations for the authors):*

I would recommend a number of additions. Potentially too many for this paper. The most important are the error analysis and the K–fiber identification issue, whilst the interpretation of Figure 8 needs clearing up. Adding more context/discussion and showing tomograph images to support your analysis would be very helpful to your readers.

Additions to support analysis claims:

1. Measurement error analysis and discussion. The manuscript fails to address any issue with respect to errors – all biological data has errors, the question is how much and if it is biased (errors on average not zero). A discussion of accuracy and cautions for interpretation is thus essential. The key issues are whether:

a. MTs are tracked accurately and along their full length. In particular are microtubules cut prematurely. This could be addressed by analysis of the location of identified MT ends in the sections, in particular errors are more likely at the boundaries of the 300nm slices. Thus, if end density is uniform across the section this provides some reassurance that sectioning (and image morphing) is not causing an error in declaring a MT end when there is none. Since sectioning of the 3 cells is likely to have different orientations with respect to the spindle axis, error rates across sections could be assessed with regard to MT orientation. Secondly, can MTs be lost between sections – what is the invisible volume width between sections? Thirdly, examine both the + and – ends and ascertain if they are consistent with MT ends in EM. Since + ends at KTs are curved, it would be good to determine if all + ends in the spindle are curved; otherwise, there may be a signature of an erroneously declared end. Please show examples from the EM. Why wasn't end structure used to determine which are the + and – ends?

b. Are MT ends located accurately, and can you estimate the accuracy?

c. Are there lost volumes within sections, or rips, where the sample was destroyed/distorted by the slicing procedure. This would give blind spots in the analysis. What proportion of cell volume is lost and how is it handled in the analysis? Are MT ends called near these problematic volumes?

2. Confirm basic statistics with other modalities. To reassure the reader, quantities such as intersister distances, spindle length, KT number, sister–sister twist (wrt metaphase plate), metaphase plate width, kinetochore size, etc could be shown to be consistent with live imaging of the Kyoto HeLa cell line. Perhaps in a Table. It was good to see in Conway et al. the polarisation microscopy comparison.

3. Show EM/tomographs to illustrate results. Given this data has 2nm resolution, there is a lack of illustrations with the imaging data. Only examples of KMT + ends are shown in Figure 2 Supplement 2. A sequence of cross sections through a 'K–fibre bundle' would be helpful, marking the KMTs and nonKMTs. This would also help visualise the 'area' and KMT density quantification. An image (sequence) showing an example MT along its length might also be useful. These would allow the reader to see the MT/bundle environment. High resolution sections and/or movies through the spindle moving from the metaphase plate to pole would be good to see, annotated with KMTs in pseudo–colour for instance. Adding this ability to their visualisation tool might be helpful, and taking an idea from Google maps – flip between tomograph section image and analysis would be great. This could be made 3D showing MT orientations as a vector field.

Additional analyses, which this data should be amenable to, and likely add to the paper's conclusions:

4. Identify K–fiber bundles. You seem to identify the K–fiber with the bundle of KMTs. I don't think it can be used in this limited way. We know that K–fibres are mechanically distinct MT bundles attached to the KT, bent by forces created by KT attachments (rebounding when cut), MTs in the bundle are crosslinked and K–fibers are cold stable. Hence, can the authors identify nonKMTs that join the bundle (K–fibre) or demonstrate they are not there. These should be parallel to the KMTs with a long association length and be physically close to KMTs. You could use your transverse circle around KMTs to also compute the density of MTs – is that density constant despite loss of KMTs. Movies/image series along the bundle center line could then be constructed showing the bundle tomography image section with distance. Going beyond the KTs would also enable visualisation of bridging fibers, if any – this should complement the analysis of O'Toole et al. on mcMTs. You could then separate MTs into KMTs, mcMTs, and neither (you mention another paper on interdigitating MTs but this paper is weakened without addressing this point). If you find bridging fibers, what is typical geometry of merger to K–fibre and distance from KT?

5. Quantify + ends. There must be a substantial number of + ends of KMTs in the spindle. It would be good to confirm this. This could be used to examine if MT ends could be generated by breaking MTs – are there + and – ends in close proximity and along the spindle MT orientation, in particular do the KMTs – ends have a + end nearby?

6. The analysis of peripheral and central KTs (I don't like the word inner and outer) you do for tortuosity can be extended to other statistics – length, fraction KMTs meeting the pole, splay etc. It seems evident that peripheral KMTs will be longer and more likely to fail to reach the spindle pole.

7. You could analyse the angles of interaction of MT end–MT lattice interactions (within 35 nm). Your length association would suggest these are parallel.

8. You don't account for the change in MT density as you move towards the spindle poles. For instance, the increased MT associations towards the pole can reflect simply this increased MT density. It might also be useful to ascertain the angular dependence of the MTs around the spindle pole (at 1.53 microns). Is it homogeneous as one moves from the pole–pole axis and behind the pole – I suspect not? Is there evidence of clusters (possibly use a statistical test here). It might also be helpful to illustrate the density of MTs. I calculated that MTs would be separated at the spindle pole sphere on average by 148nm (if homogeneous), whilst your tightest bundling of KMTs is 60nm at KTs. Distances are thus very similar at 0.7 microns, close to the peak of the minus end density.

Add more context. The results are not discussed in context of existing literature, including relating results to

i) super–resolution studies, e.g. the twist and handedness of HeLa spindles (Tolic papers, twist appears weak or absent in spindles #1–3),

ii) live cell imaging such as the EB tracking (Yamashita et al. 2015). The detected EB spots are far lower than the number of MTs detected here. Can this be explained? How many + ends (away from poles) have you in the spindle bulk, not associated with KTs.

iii) There is also no discussion of bridging fibres; this has been in the literature for a decade and a confirmation/quantification (number MTs) would be very welcome. This of course would entail examining non–KMTs, as suggested above.

iv) You have 40% of minus ends away from the spindle poles. It would be good to have your interpretation of the processes that nucleate MTs, both nonKMT and KMT, away from the poles? Does this number make sense or is this evidence of poor MT tracking?

Data release.

Tomography data: It is fantastic that the tomography data is released. I would however suggest that there is guidance on which free software can load the Amira files, or release in alternative formats.

Tracks: it would be good if the MT traces were released as well. This would enable many people to analyse the networks without the high cost of analysing the tomography data themselves.

[Editors' note: further revisions were suggested prior to acceptance, as described below.]

Thank you for resubmitting your work entitled "Three–dimensional structure of kinetochore–fibers in human mitotic spindles" for further consideration by *eLife*. Your revised article has been evaluated by Anna Akhmanova (Senior Editor) and a Reviewing Editor.

The manuscript has been greatly improved but there are some remaining points raised by the reviewers that the authors should consider before acceptance:

*Reviewer #1 (Recommendations for the authors):*

Kiewisz et al. have done a heroic job of revising their manuscript for resubmission. I appreciate the labor involved in recalculating all the images and interaction distances. Although the distances didn't change much, the images are a significant improvement, and the new data on near–neighbor MT–MT spacings are now more likely to stand the test of time. The revised neighbor distribution curves that are normalized for MT density are also a real improvement. All in all, this paper is now an excellent contribution to the literature and should be published as soon as possible. Below I mention a few issues that still strike my eye, but most of them are simple clarifications and so little work will be needed to make this manuscript a classic.

Critique

Ln 1300 Figure 8 is potentially an important figure, but I find two problems with its current version: the gray MTs on a black background are essentially invisible, and the dots in panels C and E are unexplained. This may be my display, but the authors should consider the points.

Figure 8 supplements. The lines representing MTs are so thin that they are hard to see on my screen, even when I blow each sub–image up to fill the screen. In your displays of whole spindles, it is clear why you must use narrow lines, but in these figures with only a few MTs displayed, you could increase the clarity of the images by just making each line a bit wider. Again, dots on the graphics are unexplained.

In 8Sup1, all I am seeing is KMTs in different colors. There don't seem to be any non–KMTs, and there are no sites of interaction displayed. I gather that this figure shows simply the trajectories of the KMTs that do interact, not the interactions themselves. Do you think this is the best way to show your point?

In 8Sup3, similar to Sup1. I think these figures would be more informative if you would show the interacting non–KMT and particularly the site of interaction. Simply coloring the whole MT erases that information.

Ln 331 You are right, of course, that the major peak in Figure 8, Sup2 G shows a major peak near the pole, like all the other associations, but the secondary peak farther from the pole is very noticeable in this display, and given the evidence from other groups that this kinetochore–proximal region may be an important site for interactions between the ends of non–KMTs and the walls of KMTs, I think this really deserves mention.

Figure 9, these MT–representing lines barely show up on my screen. A little thicker?

The legends for the Figure 9 supplement Figures need attention. The boxes are not referred to or explained, and again the sites of interaction are not clear. Supple. 2 legend text, "Number of KMTs plotted against the number of associations with other MTs in the spindle per individual KMT" doesn't match the labeling on those axes, so it's not clear what is being shown in these graphs.

Ln 371 and following, you state that the number of associations was 10.6, but it is not clear what this number means. Number/KMT, per k–fiber or what?

Ln 384 and following. I understand that the reworking of your images and numbers to accommodate the reviewers' suggestion of expanding the collapsed sections was a major amount of work, and probably pretty painful/annoying to do. Thus, adding this paragraph to the discussion is understandable, but as a reader coming to this version with a fresh point of view, I don't think it is necessary. You describe what you are going to do near the beginning of the paper, and you describe very nicely what you did in the methods section. It seems to me mentioning it again in the discussion over–emphasizes the point to, but this is for the authors to decide.

Ln 516 The statement that these and related data show that KMTs nucleate predominantly at kinetochores is true only for metaphase cells, and this limitation must be stated, otherwise it is misleading.

The addition of https://cfci.shinyapps.io/ASGA_3DViewer/ as a viewer tool is a major step in data sharing. Many thanks to the authors for developing this, because the Amira package is expensive. This addition will make their hard work much more useful to others.

*Reviewer #2 (Recommendations for the authors):*

The authors provide a substantially improved manuscript. We support publication. However, a number of issues (most small) remain and should be addressed before publication.

– Intro: The authors describe the three models of KMT–pole connections, but should situate known mammalian spindle structure work in the context of these models. More was known prior to this work than what was acknowledged.

– Figure 5B, 5S2B: legend and plot in 5B says centrosome interaction area was defined as a half–width of a peak from the center, but text lines 195 and 408 say "two fit half–widths from the center of the fit peak".

– Figure 5D: How can the MT–centrosome interaction area be highlighted in gray, if this distance is 1.7 µm but the P–KT length of each k–fiber is different? (i.e. the x–axis of this plot is a relative measurement, but MT–centrosome interaction distance is absolute).

– There is no Figure 5E.

– Figure 5G: Discrepancy with Figure 5A and 5G legend, where P2 is at a relative position of 1.00 vs. 1.2.

– Figure 4B, 4S1B, 5B, 5F, 5G, 9S2B, 9S2D: What criteria were used to decide which nKMTs to include? The total number from all three spindles is 16,256, based on Table 1, but these plots all include only 9957 (more than any individual spindle but far less than all three).

– Figure 5S4B legend does not match plot.

– Line 1254: What is meant by "A tortuosity of 1.1 is length."?

– Line 1255: What is meant by "The gray line indicates indicated by a dashed line."?

– Figure 6E–F: Color codes are reversed. Red corresponds to higher tortuosity in E but lower tortuosity in F, and vice versa with gray.

– Lines 231–236: A different motivation is needed for the tortuosity analysis. Tortuosity reports on microtubule curvature, but NOT twist–it doesn't tell us anything about 3D helicity that is consistent along a microtubule's length. It also seems that Figure 6 should have its own heading in the Results text, rather than being combined under the heading "K–fibers are broadened at spindle poles," since the correlation between tortuosity and spindle axis position is very weak and there is no other evidence in Figure 6 to support this statement.

– Figure 7S1D: They did not address our comment that the polygon area analysis is never motivated or explained, and is unclear.

– Figures 8–9: They did not change figures 8–9 (formerly 7–8) very much, as we suggested. Expanding their introduction to these figures in the text helps a little, but it's still hard to tell how these figures and sub–figures are making unique points.

– Figure 9A: This cartoon should be moved to Figure 9S2, because interaction length is not quantified in Figures 9 or 9S1.

– Figure 9S1: It's unclear what is being shown here or how it relates to Figure 9. Line 1375 says "This 3D model illustrates the association of KMT lattices with other KMT lattices minus ends". What are all the non–KMTs shown in these images? Are we looking at associations with lattices, with minus ends, or microtubules that interact for extended distances? Line 1376 says "The types of interactions are shown by color–coding," but what is the color code?

– Lines 497–498: What correlations is this sentence referring to? No correlations are shown in Figures 8 or 9.

---

## [Author Response]

Essential revisions:1) The authors do not take account of the collapse in section thickness that occurs when plastic–embedded samples are treated with the beam of an electron microscope. This shape–change is not isotropic, because the film upon which the sections sit shrinks little, if at all, supporting the dimensions of the embedded material in two dimensions. The dimension perpendicular to the section plane is, however, unsupported, and the section collapses along that axis by about 40%. There is confidence that this situation pertains for the current data for two reasons. The full metaphase spindles, seen in cross–section, appear either elliptical or flattened, whereas both EM cross–sections of HeLa metaphases and light microscopic images of living spindles viewed down their axes are circular. Second, each kinetochore appears elliptical in this study, whereas in previous work with spindle cross–sections, human kinetochores are circular. The ellipticity of both the kinetochore and the metaphase plates is just about the degree expected, given the 30 – 40% collapse that others have measured for ET of plastic embedded material. The potential error caused by this distortion must be resolved in the paper. One way to address this would be to apply the asymmetric expansion to their model necessary to make it a better representation of an uncollapsed spindle, as other have previously done, then recalculate all their near–neighbor distributions on the data that have been brought into true. While we do not insist on this for this quite large amount of work, the authors must resolve this artefact in their reconstruction.

Thank you for pointing this out. We are absolutely aware of this collapse in Z, and there is no doubt that there is a mass loss during imaging. We were, however, not completely sure about the correct factor that has to be applied to the collapsed sections. Following the request of reviewer #1 and referring to the publication by O’Toole et al. (2020), we have applied a Z-factor to each of our three data sets (Z-factors were: 1.3 for spindle #1, 1.4 for spindle #2 and 1.42 for spindle #3). We have mentioned this now in the main text (in the results, see new Figure 2-figure – supplement 2). This procedure is also described in the experimental procedures section. The corrected models are now shown in the new figures and supplied videos. After this expansion of the 3D models, we re-did our complete quantitative analysis of the spindle MTs. Essentially, all numbers only marginally changed, and all trends and conclusions remained unaltered. As suggested, we are also presenting the previously obtained quantifications in Tables 12-13, thus giving the interested reader the possibility to compare the data obtained before and after Z-extension of the 3D reconstructions.

2) Another issue that should be considered is that the preference for KMTs to have non–KMTs as neighbors near the pole, which they report, is uncorrected for two structural issues: the increased density of MTs near the poles, which is a natural result of the essentially spherical shape of the metaphase spindle, and the fact that there are many more non–KMT in this region of the spindle than KMTs, so naturally KMTs have more non–KMTs as neighbors. This situation says nothing about the possibility of MT–MT interactions unless the proximity data are normalized for these two considerations.

Taking the higher density of MTs at spindle poles into account, we have normalized the MT-MT interactions against the density of non-KMTs along the pole-to-kinetochore axis. The new figures show the normalized length distribution plots. Our claim that KMTs preferentially interact with non-KMTs at spindle poles remains the same.

3) When the observer is excising the tiny wafer of plastic that contains the cell for study from the thin layer of plastic in which it is embedded, it is all too easy to bend the plastic. This distorts the biological material in the plastic and may contribute to the torsion observed in the 3D reconstructions of spindle MTs. This small but important issue deserves mention.

There is a misunderstanding here about the preparation of our samples for electron tomography. Tomograms were not obtained after a re-mounting of thin layers of plastic. Here, the cells were grown and frozen on sapphire discs. The sapphire discs were removed after polymerization of the resin. Therefore, a bending of a thin layer of plastic prior to a re-mounting of the samples on dummy blocks is not an issue here. The observed torsion of the spindles is therefore not related to sample preparation!

4) The "textbook" models proposed and challenged in the text are not representative of the common understanding in the state of the field and the authors do not do enough to discuss the implications of their results or fit their results in the context of previous literature. Previous EM studies in mammalian spindles have already shown that k–fibers form semi–direct connections to the pole. The authors could instead emphasize other key findings in the paper, or provide more context as to why their main point about a semi–direct connection between kinetochores and poles in human cells is novel and exciting.

Thank you for pointing this out. After reading this comment, we realized that the semi-direct connection is not the main issue here as this has been published before our study. We only briefly mention this now in the introduction (and we clearly cite previous work on this). However, we think that the broadening of the k-fibers is indeed an unrecognized feature that needs to be reported. In the discussion we favor the interpretation that this broadening facilitates the interaction of KMTs with non-KMTs at positions close to the spindle pole. In the light of considering the mitotic spindle a liquid crystal with an intrinsic order, we discuss the possibility now that a broadening of the k-fibers could be directly related to a lower nematic order for the spindle poles compared to the kinetochore regions. In addition, we also link the minus-end morphology of the KMTs to our interpretation that KMTs grow out from the kinetochore towards the spindle poles. This way, we directly link the two submitted papers and provide context obtained from our parallel collaborative study.

5) HeLa spindle lengths were previously reported to be ~12 µm, however, here they are reported to be significantly shorter in all spindles analyzed. The authors should clarify the origin of this discrepancy.

We have used HeLa (Kyoto) cells obtained from the Gerlich lab (Vienna, Austria) that have been used previously for a cytokinesis study (Guizetti et al., 2001). This information has been added to the Materials and methods section. The discrepancy in spindle length reported for LM versus EM data is related to a dehydration step that is unavoidable in sample preparation for electron microscopy. Embedding of dehydrated samples is an issue related to all previous studies using plastic samples for either TEM or SEM. To minimize the effect of dehydration, we have performed a ‘mild dehydration’ (i.e., freeze substitution at the temperature of -80ºC). In addition, we have plotted the length distribution of KMTs against the relative position on the pole-to-kinetochore axis, so a change in the absolute number does not really change the interpretation of obtained EM data.

6) Figures 5, 7, and 8 should be better motivated. It would also help to discuss implications of findings, and to add summaries/models to the text or figure.

Thank you for pointing this out. We have tried to better motivate the studies associated with the figures mentioned above. We also have added aspects, related to these figures, in the discussion to better point out the implications of this work.

7) The paper lacks any discussion or estimation of potential errors –– tracking MTs or erroneous end declared/missed. An estimation/discussion of the errors should be provided so that the reader knows how much confidence there should be in this analysis.

This is a very important issue, thanks for bringing this up. We should have added a paragraph on this topic to the initially submitted manuscript. For this work, we applied a fully established image analysis pipeline, which includes an automatic segmentation of MTs, manual correction of false positive and negative ‘hits’ and an automatic stitching of MTs at section borders (modeling about section borders is now illustrated in the new Figure 3—figure supplement 1). Again, a paragraph of possible sources of errors should have been added, and we do so now in our revised manuscript. However, we would like to point out that we have published a number of large-scale reconstructions (for instance see, Redemann et al., 2017), in which we have discussed possible mistakes and errors already in more detail. We didn’t want to be repetitive here. Concerning the automatic stitching of MTs (as discussed in Weber et al., 2012) and the stitching of MTs at section borders (as discussed in depth in Lindow et al., 2021) we have added a new paragraph to Materials and methods. In addition, we also describe the requested expansion of our models in z (see also comments above and the new Figure 2-figure – supplement 2).

8) The authors need to provide a better explanation of the use of the term K–fiber. The paper uses the term K–fiber to refer to the bundle of KMTs. This is making an unjustifiable assumption.

We only partially agree with this comment. There is no unambiguous, universally agreed upon definition of a k-fiber. For the purpose of this paper, we first needed to ‘fish out’ KMTs from our 3D spindle reconstructions. For this, we were in need of a ‘hard criterium’ to do so and we did this by defining MTs with their putative plus ends associated with the chromosomes as KMTs. We consider these bundled KMTs as the ‘core’ of the k-fibers, and we clearly state this now in the main text of the manuscript. In a next step, we considered the non-KMTs in the vicinity of the KMTs. For further analysis and to avoid any bias here, we have chosen different ‘interaction distances’ to analyze the association of non-KMTs with the KMTs. We invested quite some time to detect a pattern of KMT/non-KMT association. Using the newly provided computer tool, the reader is encouraged to visualize the complex interaction pattern of KMTs with non-KMTs. We also provide three new figures to illustrate this issue (Figure 8—figure supplement 1 and 3 and Figure 9—figure supplement 1). However, we cannot and don’t exclude the interaction of KMTs with other neighboring non-KMTs. We mention this aspect now under cold-stabilization of k-fibers and ‘KMT maturation’ in the discussion.

The paper demonstrates that the KMTs often do not retain a tight cluster in going from KT to pole through premature termination of the KMTs and splay. Is this then the K–fiber? Is this consistent with the known structural integrity (eg under laser cutting) and high stability (cold stable) of K–fibers?

We don’t see how the core of KMTs versus a KMT/non-KMT bundle would necessarily show different data after laser cutting or cold stability. A number of MAPs can be considered here to achieve a stabilization of KMTs that run in parallel. In addition, a local modification of the KMT’s tubulin can’t be excluded as a possibility to provide stability. We consider these issues now in the discussion. In the future, it would be certainly interesting to look at the ultrastructure of cold-stabilized MTs and to further analyze the association of bundled KMTs with associated non-KMTs. However, we are convinced that this needs to be subject to a follow-up publication.

An alternative model would be that the K–fiber is a tight MT bundle that continues to the pole with a changing composition of MTs along its length. The loss of KMTs from the bundle is then simply MT exchange with the spindle. The analysis is unclear on this point – proximity to MT lattices of the KMTs has weak dependence on distance along the spindle axis, suggesting bundles are present, but this needs more analysis.

Certainly, this is a model to consider, and we have added this aspect to a new paragraph on the nature of k-fibers in our discussion. We now explicitly mention such an ‘exchange’ of MTs within the k-fiber in the discussion. However, it is unclear what objective criteria can be used to determine if particular MTs are or are not to be considered as being in a bundle. Again, we have added a new supplementary figure to illustrate the complexity of non-KMTs around KMTs. What makes a bundle a bundle? Certainly, MTs can not only be considered as bundles when they run in parallel. However, objective algorithms for such a non-KMT analysis are currently not available, and we plan to present a detailed analysis of non-KMT organization in a separate publication. Despite a detailed manual inspection of surrounding non-KMTs, we have so far been unable to see clear, qualitative indications of bundling non-KMTs. Because of a lack of unambiguous data on this point, we hesitate to make a definitive statement about this in our paper and to comment in an original publication on results obtained by other optical methods and in other labs! Finally, we don’t see why a weak dependence of KMTs with MT lattices along the spindle axis, suggests that non-KMT bundles are present. We believe that it would be best to address these issues in a future manuscript.

9). How are so many minus ends generated in the spindle – 50% for KMTs, 30% for non–KT MTs. These terminated MTs are presumably crosslinked, and the fast dynamics of astral MTs suggests they must be nucleated in the spindle itself. This should be discussed.

A short comment first, a crosslinking of MTs in the spindle is not visible in our electron tomograms obtained from plastic sections. Any statement on this point would be rather speculative. In addition, a given position of a minus end in the spindle is not necessarily the place of MT nucleation. MTs might be transported within spindles along tracks of neighboring MTs. So, we simply want to be careful here about statements on the sites of MT nucleation in spindles. However, it is not surprising that MTs are nucleated within the spindle and we comment now on this observation in our revised discussion and cite the appropriate papers. We also refer to our parallel collaborative work presented by Conway et al., in which the biophysics of KMT dynamics in the spindle is discussed in detail.

10) Inclusion of more EM images to back up and support the conclusions.

We have added more images to better illustrate the morphology of spindles in mitotic HeLa cells. We have added a new figure to illustrate the spindle poles and kinetochore MTs (for instance, see Figure 2—figure supplement 1). In this new figure, we present tomographic slices from our tomograms. However, we would like to point out here that these images are only very thin tomographic slices through whole spindles, so the reader can’t really appreciate the 3D information that is given by our 3D reconstructions. The ‘crowded environment’ of the spindles is not obvious in these thin slices showing only parts of whole KMTs and non-KMTs. To better illustrate the 3D complexity of the metaphase spindle in HeLa cells, we have introduced a platform that allows the interested readers to interactively look at various spindle features. We have added 3 new figures (Figure 8—figure supplement 1 and 3 and Figure 9—figure supplement 1) to illustrate the association of the KMTs of selected k-fibers with surrounding non-KMTs. We certainly hope that the interested reader will explore this new tool to appreciate the high density of MTs in spindles.

Please also take note of specific points raised by individual reviewers below.Reviewer #1 (Recommendations for the authors):Abstract: While K–fibers do connect kinetochore directly with the poles in many cells, in others the k–fiber connects with other spindle MTs, rather than the material at the pole. Even in cells with clearly defined poles, connection with other MTs may contribute to spindle mechanics (as is shown in this study). Therefore, it might set the reader's mind on the right course by replacing the first sentence with something like, "… provide a physical linkage between the chromosomes and the rest of the spindle."

Thank you, we have changed this in the abstract.

Ln 112 I think you mean "Figure 2–movie supplements 1–3".

This has been corrected. Sorry for this mistake.

Ln 114 The observations of 14% of the MTs being KMTs, but those MTs accounting for 24% of the tubulin mass make an interesting discrepancy. I presume the difference in these numbers comes from the fact that KMTs are, on average longer than non–KMTs, but whatever the reason, it should be explained.

Thank you for pointing this out. As given in the text, we have measured the average length of all KMTs and non-KMTs in our spindle reconstructions: average length is 3.87 ± 1.98 µm for KMTs and 2.0 ± 1.76 µm (mean ± STD; n = 9957) for non-KMTs. So, KMTs are on average almost twice as long as non-KMTs. We have shifted this issue to the discussion.

Figure 2 Supplement 1 Note that the legend to this figure is a repeat of the legend for figure 2, and there is no legend for this Figure

Sorry for this mistake. We solved this issue.

In Figure 3B you call these kinetochores sisters 1 and 2. If you are confident that they are a pair, it might be more meaningful to your readers to call this pair of kinetochores sisters #1 and call the individual kinetochores 1A and 1B. I certainly don't insist on this nomenclature change, but it might help readers to follow you.

We agree, this is a good suggestion and will help the reader to follow. We have changed this in the legend for Figure 3B.

Figure 2 Supplement 2 In the legend (ln 884) you use the term "cross–section" to refer to tomographic slices that cut through a MT parallel to its axis. I find this confusing, because the cross–section of a MT is usually a slice cut perpendicular to the MT axis. Rephrase?

Yes, this is not precise enough. We have changed this in the figure legend.

Ln 116 Were the Kinetochore–proximal ends of ALL KMTs flared, as shown? In previous publications many KMT plus ends were flared, but not all.

We will report on the end morphologies of KMTs in mitotic HeLa cells in a separate publication. We decided to analyze the MT ends in the context of depletion of the MT minus-end regulator, MCRS1. This is a collaboration with the lab of Isabelle Vernos (Barcelona, Spain). This parallel publication (Laguillo-Diego et al.) will be submitted also to *ELife* very soon.

Ln 119 O'Toole et al. have examined the pole–proximal ends of KMTs in *C. elegans* and found that about half of these ends are open, half appeared to be capped, perhaps by the γ–tubulin ring complex. I wonder of your data would allow you to make comparable observations on KMTs in Hela cells? Mammalian spindle MTs flux toward the spindle pole, whereas comparable data for the C.e. embryonic spindle are (to my knowledge) not available. It would be very interesting to know if KMT minus ends in HeLa are capped or not. If capped, how can they flux?

Yes, a similar analysis can be done. In fact, we did this already in the context of the parallel study (Laguillo-Diego et al.) on the morphology of MT minus ends in untreated and MCRS1-depleted HeLa cells. Indeed, KMT minus ends are heterogeneous showing either open or closed morphologies.

Ln 120 Since there is no legend for Figure 2 Sup 1, I am making an inference about panel C on that figure, but it looks as if you use the 3D positions of each KMT associated with a single kinetochore to find a 3D average position, then call that the position of the kinetochore. This is not strictly true, because the plus ends of the KMTs are not AT the kinetochore, they commonly end outside the outer plate in the corona material. It would be valuable to find a wording that states more explicitly what you are doing, because the distances you present for the spacings of sister kinetochores are likely to be bigger than those measured by others with different definitions of the kinetochore.

Formally, this is true. And yes, the presented distances for the spacings of sister kinetochores are bigger. We were interested in a comparison of the distance between the plus-ends of sister k-fibers. We now determine the ‘distance

between the median position of KMT plus-ends between pairs of sister k-fibers’. We hope that this is clearer now. Sorry again,

Ln 128 "Site facing the pole" is imprecise and undefined.

We have changed this to pole-proximal end.

Ln 134 and following. If I have understood you correctly, you estimated the size of each kinetochore by the area occupied by the KMTs that ended at approximately on region of the spindle because you did not see extra staining density in the chromatin or some other structural indicator of the kinetochore itself. That's OK, but you then go on to calculate the correlation between the number of KMTs and the size of the area they occupy and get a great correlation. But this is not meaningful. Of course, you get a good correlation because you are estimating the size of the kinetochore by the area occupied by the MTs. You then say that this is in good agreement with Cherry's observations on HeLa by LM, without saying what Cherry saw. I find this set of statements unhelpful for understanding K size.

We agree that we have to be clearer here! We were not primarily interested in the absolute size of the kinetochores in our spindles. This indirect measurement of KMT area was done to calculate the density and spacing of KMTs at the outer kinetochore (and along the length of k-fibers, later in the paper). We deleted the comparative statement about kinetochore size as measured by light microscopy and in other systems. We hope that this point is clearer now.

Ln 902 There is something funny about the font for G of Graph and then P of Plot. Similar problems throughout this legend and others, so check the PDF carefully before it is posted.

We checked the font size in all our graphs. Thank you for pointing this out.

Ln 903 Panel B does not show the distribution mentioned in the legend. It shows a related fact, which is important, but the distribution of minus ends is the subject of Figure 4. Have you computed it relative to a pole–to–pole axis, or the like? Perhaps relative to distance from the pole? The fact that so many KMTs do actually reach the pole is important and surprising to me. Based on K–fibers in PtK cells, I would have thought that more KMTs would have ended farther from the pole. Interesting.

Sorry that we caused some confusion here. To be clearer here, we have changed this figure. In panes A, we now show the length distribution (i.e., the measurements on the length) of KMTs and non-KMTs in a separate figure (Figure 4). In Figure 5A we now illustrate two measurements: First, we show the measurement of the absolute distances of the KMT and the putative non-KMT minus ends to the center of the nearest mother centriole. This is indicated by black arrows. The absolute values are given in µm. Second, we show the relative position of the KMT and of the putative non-KMT minus ends on the pole-to-kinetochore and on the pole-to-pole axis, respectively. As for the relative position of the KMTs on the pole-to-kinetochore axis, each KMT plus end was defined as position 1. We hope this is clear now in the new Figure 5 (and also in the new Figure 5—figure supplement 1 showing both spindle poles). In panel B, we explain how we determined the MT-centrosome interaction area. This was done to introduce a clear ‘border’ for an annotation of KMT minus ends as either directly or indirectly associated with the spindle poles. We have added a paragraph about this MT-centrosome interaction area to the discussion.

Ln 910 As indicated by my comments relevant to the text derived from this figure, I don't see much value in this analysis, given the way you defined the position and size of the kinetochores.

Is this comment related to the centrosomes instead of the kinetochores? Hopefully, our definition of the centrosome border is clearer now.

Ln 160 and following. Your definition of the distance from the pole within which you call MT ends pole–interacting is clear, but your reasons for choosing this distance are not. In a sense this is a discussion point, and so how either you or Steffie decided this was the right distance might best be presented in that part of the paper. However, something should be said about it here, because in its current form, the distance chosen sounds completely arbitrary. Perhaps add a discussion paragraph and say here that this point is clarified in the discussion?

We agree on this. The reason for choosing the border for this interaction area needs clarification and more explanation in the text. We tried to be clearer about this in the Results section (also in the figure legend) and have added a paragraph to the discussion. As also obvious in our previous publications on MTs in the early *C. elegans* embryo (O’Toole at al., 2003; Weber et al., 2012; Redemann et al., 2017), it is impossible to define a clear boundary for the membrane-less spindle poles in EM images. As a solution to this problem, we plotted the number of all putative non-KMT minus ends to the distance of theses minus ends to the nearest mother centriole (absolute values given in µm; these are not relative distances!). The border of the MT-centrosome interaction area (Figure 4B, right dashed line) was then determined by the bend in the histogram, where the peak plateaued (arrow). We hope this is clearer now.

Ln 919 The end markers are circles, not spheres.

Of course, you are right. Sorry, we have checked and changed this throughout our manuscript.

Ln 923 You map non–KMT end positions onto the pole–to–pole axis, which makes sense. You say that you map KMT end positions onto a kinetochore–to–pole axis, but that implies that you used a different axis for every kinetochore. Is that true? If so, you should say how these axes were determined, and a figure to show exactly what you mean should be added.

We agree, we need to better explain this, and it is also a good idea to add a figure here to show how we defined the kinetochore-to-pole axis. We defined this axis for each kinetochore. The spindle pole is always position 0, while the kinetochore is always position 1 on the half-spindle axis.

Figures 4D and G: Because you say "distance" in the text and on the abscissa, I presume this is what you have previously called "Euclidian distance", as suggested in 4A. Please be specific here because it is important for thinking about your data. In G, you again say distance, but here, I presume it is distance along the pole–to–pole axis, since that is how you describe it above. Thus, in your current labeling, the word distance means different things in these two panels. If I have this right, you should clean up the terminology. Note that again there are font problems at several places in the PDF.

Again, sorry for being unclear here. In Figure 4 (now Figure 5), we are talking about two types of measurements (see our comments above): first, about absolute distances; second, about relative positions on either the pole-to-kinetochore or the pole-to-pole axis. We have cleaned up the terminology. We hope that this is clearer now.

Ln 942 Here again distance of KMT minus ends to poles is referred to simply as "distance". If this is a 3D Euclidian distance, as previously implied please change wording. Now in panel C, you are presenting distances as mapped onto the spindle axes, so these should be directly comparable to the distances you present for nonKMTs. If I'm right about this, I think you should say so explicitly. For example, Panels C and E are directly comparable, and the penetration of KMTs beyond the position of the mother centriole is nicely demonstrated. But now you state that E represents real distances (Euclidian?) and F is position on the axes. This implies that you measured distances from minus ends of nonKMTs to the centriole, which has not previously been mentioned. This needs clarification and explicit description with clearer words. Moreover, the shapes of these plots shown in Figure 4 sup1 F are remarkable for their symmetry, and this deserves comment and explanation.

As mentioned above, we have modified Figure 4. In the new Figure 4, we talk about the length distribution of both KMTs and non-KMTs. In the new Figure 5, we now talk about our analysis of both KMT and non-KMT minus ends. Hopefully, this new presentation of the data is much clearer now. In addition, we have added a comment on the symmetry of the plot in Figure 4—figure supplement 1F (now Figure 5—figure supplement 2D).

Ln 180 This is the only reference to Figure 4 supple. 2, which is an interesting figure. I wonder why this information doesn't get more attention in the text.

We mention this point now in the text.

Also relevant to this figure: here better than anywhere, one sees the elliptical shape of the cross sections of your spindle reconstructions. Given what one knows from light microscopy, this is almost certainly an artifact of electron tomography. I presume it is a result of the collapse of section thickness in the beam of the microscope. Some mention and explanation of this feature of your reconstructions belongs in the paper somewhere.

Again, we took care of this collapse and present the z-corrected figures in the revised version of our manuscript.

Ln 196 I find this title to be a bit of an exaggeration. Splay implies really flaring out. Yes, the area of the k–fiber encompassing polygons increases toward the pole, but I would describe it as a widening, not flaring or splaying.

As requested, we now report on a ‘broadening’ of the pole-proximal ends of the k-fibers.

Ln 225 You are using polygon here and below as an adjective. Should be polygonal.

Thanks, we have changed this.

General comment on the study of MT associations. Some of these statements and graphs need a bit more thought. Figures 7B and 7D show that the greater the distance between an MT end and an MT wall that is considered to be an "interaction", the greater the number of interactions there are. This point is almost trivial; it must be true. I don't see that it is worth two graphs. Said another way, the looser the criterion use for MT–MT interaction, the more interactions are seen. I don't think this needs to be demonstrated.

We agree, this must be true. The point here is to show how the association patterns change by increasing in the interaction area. All in all, we have to say that the analysis of MT-MT interactions turned out to be very complex, and we decided to keep Figures 7 and 8 to explicitly show these complex interaction patterns. We fully agree that additional analysis will be necessary in the future.

Ln 245 It might be worth mentioning MT–interacting proteins that are not motor enzymes, since some of these have been shown to be important for spindle structure and function.

Thanks, this is a good point. We have added this information.

Ln 255 The term association distance is introduced without any intellectual background. This may be a bit cryptic for people who have not been reading papers about spindle MTs and the ways structural studies have tried to identity MT–MT interactions. Perhaps add a sentence that rationalizes your interest in "interaction distance"? Perhaps a reference to others who have used this approach?

Thanks, we have added a sentence here to better explain what we did here.

Ln 261 You say that most of these interactions occur near the pole, and you also say that many of the KMTs end before getting near the pole. Thus, the 41.4% of KMTs that have an end interacting with the wall of a non–KMT might be quite a large fraction of the KMTs that get into the polar area and are therefore in a position to have this kind of interaction. Can you look at this and clarify the point? For example, if you find a distance from the mother centriole that defines a hemisphere in which MT–MT distances are small, what fraction of the KMTs that get into that region have a tip interaction with a non–KMT.

We do not understand this comment. However, we think that a similar analysis is presented in the new Figure 8F and G.

Ln 270 and following. The observation that most of the MT–MT interactions (i.e., sites of close proximity) occurred near the spindle pole might be important, but a qualitative look at the 3D reconstructions suggests that the MT density is greater near the poles than farther away. By this I mean in a given spindle cross–section, how many MTs are there/unit area. If this number is higher near the poles, then the MTs must be closer to one another, so the increase in the number of MT proximities is a necessity, not a novel fact. This issue should be considered in the presentation of these data. Perhaps you could compute the MT density as a function of position along the spindle, then use this number at each position to normalize the distributions of interactions, thereby asking whether there is something in the data to indicate that the MTs are binding with one another, rather than simply being packed in. Also, the mechanical significance of KMT ends near KMTs is not clear to me.

Thank you. This is indeed an issue that needs to be considered. As suggested, we normalized the distribution of the MT-MT interactions.

Ln 276 You say, "as a result", but that phrase could be omitted. You are just describing what you saw.

Agreed, we have taken this phrase out.

Ln 290 Note that the number of non–KMTs/spindle cross–section exceeds the number of KMTs/cross section, and this is particularly true near the spindle poles. Thus, the fact that there are more associations between KMT ends and non–KMTs than with KMTs may simply be a consequence of this population difference. This could be calculated on the bases of the relative densities of the two kinds of MTs.

We took care of this aspect and normalized our graphs against the MT densities.

Ln 295 The fact that association frequencies are higher near the pole regardless of association distance may again be based simply on MT density.

We have considered this aspect by normalizing our data against the density of the surrounding MTs.

Ln 296 In this context, I'm particularly interested in Figure 7, supple. 1E. This distribution of interactions between the walls of KMTs and the ends of non–KMTs does show its biggest peak at the pole, as stated, but there is a good indication of a second peak, not far from the kinetochores. This peak would be clearer if the data were presented as a running average, since the fluctuations seen make the bar graphs very spikey. Moreover, if the normalization, based on MT density were applied, it would reduce the polar peak but not the rise in frequency that is seen nearer to the kinetochores. As you know, this position along the spindle was identified structurally by O'Toole et al. as a place where the minus ends of overlapping MTs from the interzone (a subset of your non–KMTs) made association with the walls of KMTs. It is also the place where two groups using microbeams have identified places where KMTs receive a push from the spindle interzone. Thus, the gentle rise in your data may be very significant for the functional interactions between different classes of spindle MTs.

Thank you for pointing this out. We have added the ‘moving average’ to this graph.

Ln 297 This discussion sounds just like what Mastronarde and others have called pairing length in their studies of yeast spindles. It might be well to use similar terminology to minimize confusion. Also, I think there might be a clearer way to describe what you have done. Here is my effort: "To measure the paraxial length over which MTs interacted, we defined 20 nm along the axis as a minimal length of interaction. For each pair of MTs that lay within a chosen interaction distance, we measured the number of 20 nm segments over which they retained this proximity."

Thank you for your elegant suggestion. We have changed this in our manuscript accordingly.

Ln 305 I'm a little uncertain about what the words here mean. Are the "peaks of association" peaks in the number of 20 nm paring lengths that describe the pairing distances? Please clarify.

Yes, this is related to the number of associations with a 20 nm-pairing length. We have corrected this.

Figure 8 It seems as if the legends to 8B and C are switched.

Thanks, we checked this and changed it accordingly.

Ln 305 and following. These words don't do justice to the very interesting distribution graphs you show in Figure 8. In the text, you don't discriminate between the classes of MTs that you break out in your graphs. For example, the words here describe what is shown in Figure 8B but not 8D. Please be specific about which MTs your words refer to.

Thank you. We have cleaned this up.

Ln 309 You see no "hotspots" in your data for interactions between non–KMTs, but in Figure 8D there is a significant peak near the spindle midplane. (The label on the ordinate of this graph says non–KMTs interacting with all other MTs. I presume that you mean both other non–KMTs and KMTs. If this is not right, please be more specific.) Granted, the peak is higher for greater interaction distances, but studies in fission yeast suggest that the mean distance between MTs in the zone of overlap is greater than near the poles, and this the region where kinesin motors generate pushing forces. This peak may be of functional significance, and I think deserves mention in the text.

Agreed, we have mentioned this in the text now.

Ln 327 The term "spindle MTs" appears here without definition. Please clarify.

We changed this to “any MT in the spindle”.

Ln 369 In the text and in Figure 9 it seems to me that you are under–representing the extent to which some KMTs reach the polar area. Your summary diagram shows only one MT of five making to the polar region, whereas both the images you show in the several movies for Figure 3 and your earlier text suggest that the fraction is closer to one–half. This diagram is important, because it is likely to be what gets cited and maybe even reaching textbook figures. Make sure it shows what you have actually seen.

Very good point. Thanks! We agree that the last schematic drawing should be a visual abstract/representation of our paper. We worked on this graph and made sure that 50% of the shown KMTs reach the pole. We also worked on the paragraph of the discussion covering this issue!

Ln 383 This statement about strong indications of KMTs preferring to interact with non–KMTs is subject to the concern raised above that your data are not corrected for the greater number of non–KMTs. Make sure to do this normalization before making this claim.

As suggested, we present out data now as normalized graphs, but the overall interpretation of our data remains the same.

Ln 472 No mention is made in the methods section of coping with the fact that plastic sections collapse during the early stages of exposure to the electron beam. Although there is some shrinkage in the plane of the section, the collapse of section thickness can be as much as 40%, and this affects 3–D geometry of the reconstruction. In my opinion, the fact that both the over–all shapes of your spindles and the shapes of each kinetochore fiber are oval, not circular, is a result of this collapse. Some labs expand the dimension perpendicular to the section plane to correct for this collapse. Was this correction applied here? If not, do you think it has impact on the distances observed between spindle MTs? Some discussion would be appropriate.

We completely agree that there is a collapse of the plastic sections during imaging. As stated in our comments to the ‘essential revisions’ (see above), we have corrected this collapse by applying a z-factor to our full 3D reconstructions. We have added a new figure illustrating this point (new Figure 2—figure supplement 2). We have also added a new paragraph to the Experimental Procedures, where we talk more about how this z-expansion was achieved. Basically, we followed the procedure as described in O’Toole et al. (2020). Essentially, the numbers in our analyses changed only marginally! For clarity, we are showing the data obtained before and after z-expansion in Table 12 and 13.

Ln 535 "was found within the interaction distance" would be clearer.

We changed this to “were positioned“.

Ln 1099 The words, "This video shows" could be omitted from each of these legends with no loss.

We have deleted these words from all video legends.

Reviewer #2 (Recommendations for the authors):Figure 1–The authors should include the semi–direct connection between kinetochores and poles, since this has previously been shown.

We agree. As requested, we have added this to Figure 1.

Figure 2– Figure legend 2C should indicate that microtubule number is in the upper right corner.

We have indicated this in the figure legend.

Figure 2–supplement 1.– The figure legend is missing.

Thank you for pointing this out. This is now Figure 2—figure supplement 3. We have added the appropriate legend for this figure. Sorry for this mistake.

– Figure 2–supplement 1B indicates pole–to–pole distances of > 8µm for each spindle, with an average around 10.5 µm. However, Table 1 reports spindle pole distances of ~7, 10, and 9 µm. The authors should clarify this discrepancy.

We have corrected this.

Figure 3– Line 140 in the text reports a KMT packing density of 0.07 KMT/µm2, and Table 3 reports similar KMT packing densities (0.06–0.09 KMT/µm2). However, Figure 3–supplement 1E reports KMT densities in the range of 25–75 KMTs/µm2. The correlation coefficient on lines 137–8 also doesn't match the correlation coefficient on Figure 3–supplement 1D.

We have corrected this.

– Authors should clarify the motivation for comparing inter–kinetochore distance and the number of MTs at the kinetochore, and what previous evidence might suggest a correlation (Figure 3C).

Our motivation here was simply to check if a stretch between kinetochore pairs has an impact on the number of attached KMTs as this would influence the average number of KMTs per k-fiber. We just wanted to rule this out. This point should be clearer now in the revised version of our manuscript.

– Authors should define "neighborhood KMT–KMT" in the figure legend for Figure 3–supplement 1F.

This refers to the average KMT-to-KMT distance at the kinetochore in each k-fiber. We have changed this in the figure legend.

Figure 4– The authors make several conclusions about the composition of k–fiber bundles, for example, that k–fiber bundles are made up of varying KMT lengths. However, they only present gross KMT data. To directly conclude that individual bundles are made up of varying KMT lengths, they should show their KMT data grouped by k–fiber bundle.

We agree on this point and have added a new figure (Figure 4—figure supplement 2) in which we show the length distribution of the KMTs for the analyzed k-fibers. We have also grouped the k-fibers according to their position on the metaphase plate. Essentially, all k-fibers of all three groups show a high variation in KMT length.

– Relating to Figure 4–supplement 2N, the authors could explain the significance of finding more KMT minus–ends at spindle poles in central k–fibers compared to peripheral k–fibers, expanding on what the motivations were for this analysis and what might it say about global k–fiber organization. The authors could also consider plotting KMT length versus kinetochore region.

Yes, this is a good suggestion. We did this additional analysis for individual k-fibers by showing the variation in the length distribution of KMTs (Figure 4—figure supplement 3). In addition, we present this analysis in the context of the defined three kinetochore regions (now called central, intermediate and peripheral).

– The color of the outer kinetochore region does not match the colors of the images or plots, similarly in Figure 5.

We have corrected this and adapted the color coding. Thanks for pointing this out.

Figure 5– The authors could expand on their motivations for studying tortuosity of KMTs, and speculate on what their findings could imply about spindle organization.

Thanks. We realized that we should elaborate on this point. Briefly, the tortuosity of the spindles was previously reported by the Tolic lab. We simply wanted to check whether this bending of the MTs as observed by LM is also obvious in our 3D reconstructions, which we could confirm by EM. A simple explanation for this tortuosity in spindles could be that most of the internal bending is caused by MTs that align relative to each other, possibly by some combination of molecular motors and/or passive crosslinkers (see also our parallel study by Conway et al. for our consideration of spindles as liquid crystals). We don’t think that the tortuosity of spindles is directly related to the segregation of the chromosomes.

Figure 6– The polygon area analysis shown in Figure 6–supplement 1D is unclear and difficult to follow. The authors could clarify why this method was necessary.

Our intention is to calculate the cross-section area of k-fibers. We quickly realized that such an analysis can’t be done by simply fitting a circle, and this is why we thought that the polygon area measurements would be much more precise. We added a comment on this in the results.

– Line 229 confusingly states that the polygon area of the k–fiber was 5–fold lower at the pole–facing end of the fibers. It is unclear whether this is relative to the mid–position or to the kinetochore.

Sorry for being unclear here. This refers to the mid position. We have clarified this issue in our revised version.

Figure 7 and Figure 8– Some of the text in Figure 7 is too small to read.

Good point. We have changed this and checked all figures for clarity and readability.

– The authors could condense and clarify these figures, as well as compare their findings to previous work on KMT and non–KMT minus–end distributions (O'Toole et al. 2020).

We have intentionally focused our presentation on the patterns of the KMT/non-KMT interactions. Another goal was to show the complexity of the 3D data. Essentially, we did not find signs of bundles of non-KMTs or other clear patterns of non-KMTs in the vicinity of k-fibers. In our paper, we simply want to be careful about this proposed bundling of non-KMTs as proposed by the Tolic lab. The fact that we didn’t find such bundles in our analyses is certainly not a proof for their nonexistence. Our interpretation, however, is that the KMT/non-KMT interactions are statistically distributed all over the spindle but this needs additional analysis. We believe that this is an interesting extension of the present work, which would be best suited for a future publication. However, the complex 3D pattern should be presented here. To illustrate this complexity, we applied our new 3D tool to show the interaction of selected k-fibers with surrounding non-KMTs. We have added new supplementary figures here (Figure 8—figure supplements 1 and 3; Figure 9—figure supplement 1). In this context, we also mention the data presented by O’Toole et al., 2020. Thanks for this hint.

– The authors could discuss more on the key differences found between kMTs and non–kMTs, specifically how the distributions of these two populations are related to their lifetime, organization in the spindle, and mechanistic differences.

We now discuss the implications of our work for the model of spindle organization that has been explained in more detail in the accompanied publication by Conway et al.

– This overall section would benefit from a general summary in the text and model figure(s).

Is this comment related to the paragraphs covering Figures 7 and 8? We have expanded the discussion on this issue.

Reviewer #3 (Recommendations for the authors):I would recommend a number of additions. Potentially too many for this paper. The most important are the error analysis and the K–fiber identification issue, whilst the interpretation of Figure 8 needs clearing up. Adding more context/discussion and showing tomograph images to support your analysis would be very helpful to your readers.

We have added a paragraph on error analysis in the Experimental Procedures. While we agree that information on this issue should be added to this particular publication, we would like to point out that our 3D reconstruction pipeline has been described in detail in previous publications. An extended analysis on the efficiency and accuracy of template matching for MT segmentation has been published in Weber et al., 2012. Recently, we have also presented a tool to evaluate the accuracy of MT stitching across serial section borders (Lindow et al., 2021). In this new paragraph we consider issues of possible errors during 3D reconstruction and mainly refer to the appropriate publications from our lab. We also worked on the presentation of Figure 8 (now Figure 9), re-wrote the discussion and showed more tomographic slices (see new Figure 2—figure supplement 1).

Additions to support analysis claims:1. Measurement error analysis and discussion. The manuscript fails to address any issue with respect to errors – all biological data has errors, the question is how much and if it is biased (errors on average not zero). A discussion of accuracy and cautions for interpretation is thus essential.

As mentioned above, a number of error analyses have been done in the context of previous publications, and we didn’t want to be repetitive here. The stitching of MTs over section borders has been described in detail in Lindow et al., 2021. However, we have added a new paragraph to the Experimental Procedures – *Error analysis*, in which we mention the accuracy in MT segmentation and stitching across section borders. We have also added a new figure (Figure 3—figure supplement 1) that shows the exact tracing of MT across several sections and also visualizes the distribution of stitched MTs over all serial section interfaces.

The key issues are whether:a. MTs are tracked accurately and along their full length. In particular are microtubules cut prematurely. This could be addressed by analysis of the location of identified MT ends in the sections, in particular errors are more likely at the boundaries of the 300nm slices. Thus, if end density is uniform across the section this provides some reassurance that sectioning (and image morphing) is not causing an error in declaring a MT end when there is none. Since sectioning of the 3 cells is likely to have different orientations with respect to the spindle axis, error rates across sections could be assessed with regard to MT orientation.

As for MT segmentation, an analysis specifically on the ends at section borders has been done extensively in a previous publication (see: Automated stitching of microtubule centerlines across serial electron tomograms, Weber et al., 2014). Using exactly the same computational approach, it is stated: “In our experiments, 95% of the computed connections agreed with an expert's opinion, but 4% (*X. laevis*) and 1% (*C. elegans*) connections disagreed.” We would like to point out here that each automatic stitching event in our reconstructions (caused by ‘ends’ of the same MT at section borders, termed ‘MT connection’ in Weber et al., 2014) is checked and (if necessary upon visual inspection) corrected manually.

In our manuscript we now write: "We expect to find approximately the same density of MT endpoints along the Z-direction of each serial-section tomogram. This distribution is visualized in the Serial Section Aligner tool previously presented (Lindow et al., 2021)."

Secondly, can MTs be lost between sections – what is the invisible volume width between sections?

There is indeed a ‘smeared region’ at the top and the bottom of each tomogram. These regions, however, were generated during reconstruction of the tomograms, later used for flattening and then removed before stitching. These regions do not contain MTs. As for the accuracy in the stitching of MT ends, please see our comment above.

Thirdly, examine both the + and – ends and ascertain if they are consistent with MT ends in EM. Since + ends at KTs are curved, it would be good to determine if all + ends in the spindle are curved; otherwise, there may be a signature of an erroneously declared end. Please show examples from the EM. Why wasn't end structure used to determine which are the + and – ends?

In parallel to this study, we have analyzed the end-morphologies of MT ends in the context of untreated and MCRS1-depleted HeLa cells. This analysis will be published as an additional study. The data will also be submitted to *ELife* and will be available via bioRxiv very soon. Of course, the end morphologies as visualized by electron tomography match exactly the end structures as observed previously by thin-section EM. We have shown this in a number of earlier publications (e.g., see O’Toole et al., 2003; Redemann et al., 2017)! Briefly, the plus ends of the KMTs at kinetochores are all open (mostly) flared, and we see both open (blunt) and closed (capped) minus ends at the spindle poles. Only by visualizing the end morphology, it is not possible for all MTs to annotate ends as either plus or minus. This is particularly true for the blunt ends. Therefore, we refer here to pole-proximal and -distal ends, thus implying that we talk here about MT minus and -plus ends, respectively. This is in full agreement with our previous publications.

b. Are MT ends located accurately, and can you estimate the accuracy?

We are very confident that MT ends are located accurately in our reconstructions. Each MT (including each KMT) in our tomographic reconstructions was manually checked for a correct segmentation at both ends! In Amira, we always check the segmentation of each MT end in 3D (allowing an inspection of each end in an optimal orientation for viewing). We were very careful about this, as the accuracy in the segmentation process is absolutely crucial to this ultrastructural study.

c. Are there lost volumes within sections, or rips, where the sample was destroyed/distorted by the slicing procedure. This would give blind spots in the analysis. What proportion of cell volume is lost and how is it handled in the analysis? Are MT ends called near these problematic volumes?

Of course, we used only perfectly sectioned ribbons of serial sections for our large-scale reconstructions! Serial sectioning is routinely established in our EM lab.

2. Confirm basic statistics with other modalities. To reassure the reader, quantities such as intersister distances, spindle length, KT number, sister–sister twist (wrt metaphase plate), metaphase plate width, kinetochore size, etc could be shown to be consistent with live imaging of the Kyoto HeLa cell line. Perhaps in a Table. It was good to see in Conway et al. the polarisation microscopy comparison.

On purpose, we have moved all light microscopic observations to our parallel manuscript presented by Conway et al. To our mind, the best argument that the EM and LM data are in agreement is the fact that we could fully recapitulate the length distribution of the KMTs by combining both imaging modalities for biophysical modeling. We thought that the data obtained from electron tomography and LM combined with stochastic simulations show remarkable similarity. We strongly believe that both publications have to be read in parallel.

3. Show EM/tomographs to illustrate results. Given this data has 2nm resolution, there is a lack of illustrations with the imaging data. Only examples of KMT + ends are shown in Figure 2 Supplement 2. A sequence of cross sections through a 'K–fibre bundle' would be helpful, marking the KMTs and nonKMTs. This would also help visualise the 'area' and KMT density quantification. An image (sequence) showing an example MT along its length might also be useful. These would allow the reader to see the MT/bundle environment. High resolution sections and/or movies through the spindle moving from the metaphase plate to pole would be good to see, annotated with KMTs in pseudo–colour for instance. Adding this ability to their visualisation tool might be helpful, and taking an idea from Google maps – flip between tomograph section image and analysis would be great. This could be made 3D showing MT orientations as a vector field.

The problem with complex 3D data is that it is very difficult to simultaneously illustrate both the overall structure and the details of certain regions. However, we have added a gallery of tomographic slices to better illustrate the spindle (see new Figure 2—figure supplement 1). The spindle is extremely ‘crowded’ and can be analyzed best in 3D reconstructions. We have added a video (Figure 9-video 2) to illustrate the complexity of the 3D organization in the zone between the spindle pole and the metaphase plate. In this video, MT-MT interactions with an association distance of up to 35 nm are shown. Our primary motivation for presenting the new visualization tool was to give the interested reader the freedom to look at the reconstructions at any desired 3D feature of the spindle at any desired perspective/angle. Compared to the limitations accompanied with previously submitted supplementary videos, we think this is a significant improvement in presenting and visualizing the data.

The Google maps approach is certainly a good idea, and this has been on our to-do-list for quite some time. As a first step towards a better visualization of the tomographic data, we present here (as mentioned above) our new visualization tool. We thought that this is a step in the right direction. More improvements will follow in the near future but this requires much more work and has to be subject to future studies.

Additional analyses, which this data should be amenable to, and likely add to the paper's conclusions:4. Identify K–fiber bundles. You seem to identify the K–fiber with the bundle of KMTs. I don't think it can be used in this limited way. We know that K–fibres are mechanically distinct MT bundles attached to the KT, bent by forces created by KT attachments (rebounding when cut), MTs in the bundle are crosslinked and K–fibers are cold stable. Hence, can the authors identify nonKMTs that join the bundle (K–fibre) or demonstrate they are not there. These should be parallel to the KMTs with a long association length and be physically close to KMTs. You could use your transverse circle around KMTs to also compute the density of MTs – is that density constant despite loss of KMTs. Movies/image series along the bundle center line could then be constructed showing the bundle tomography image section with distance. Going beyond the KTs would also enable visualisation of bridging fibers, if any – this should complement the analysis of O'Toole et al. on mcMTs. You could then separate MTs into KMTs, mcMTs, and neither (you mention another paper on interdigitating MTs but this paper is weakened without addressing this point). If you find bridging fibers, what is typical geometry of merger to K–fibre and distance from KT?

We only partially agree with this comment. What is the ‘correct’ definition of a k-fiber, and which of the numerous surrounding non-KMTs have to be considered as being associated with the KMTs to establish a k-fiber? For the purpose of this paper, we ‘fished out’ the KMTs from our spindle 3D reconstructions by defining MTs with their putative plus ends associated with the chromosomes as KMTs. We consider these bundled KMTs as the ‘core’ of the k-fibers, and we state this now in the main text of the manuscript. Following this initial characterization, we have considered the non-KMTs in the vicinity of the KMTs. For further analysis and to avoid any bias here, we have chosen different ‘interaction distances’ to analyze the association of the non-KMTs with the KMTs. Using the newly provided computer tool, we now present such interactions in additional supplementary figures (Figure 8—figure supplements 1 and 3; Figure 9—figure supplement 1). The interested reader is encouraged to visualize the interaction pattern of KMTs with non-KMT. Certainly, we don’t want to exclude non-KMTs from our analyses but we need a ‘hard criterium’ to extract those non-KMTs from the population of about two thousand MTs that built the k-fibers. We have added a paragraph on this issue to the discussion. We discuss this core structure of k-fibers now in the context of a maturing of the k-fibers.

5. Quantify + ends. There must be a substantial number of + ends of KMTs in the spindle. It would be good to confirm this. This could be used to examine if MT ends could be generated by breaking MTs – are there + and – ends in close proximity and along the spindle MT orientation, in particular do the KMTs – ends have a + end nearby?

Per definition, KMT plus ends have to be associated with the kinetochores, otherwise we have to classify them as non-KMTs. We are planning on a separate study to show the position of non-KMT plus ends in the spindle. This will be complemented with a detailed analysis of non-KMT length distribution in the context of spindles in which specific proteins, such as augmin and others, have been depleted. This is Best suited for a follow up study.

6. The analysis of peripheral and central KTs (I don't like the word inner and outer) you do for tortuosity can be extended to other statistics – length, fraction KMTs meeting the pole, splay etc. It seems evident that peripheral KMTs will be longer and more likely to fail to reach the spindle pole.

Of course, we could add more analyses on the tortuosity of k-fibers. In the interest of the length of this paper, we are afraid that additional analyses have to be subject to a future publication.

7. You could analyse the angles of interaction of MT end–MT lattice interactions (within 35 nm). Your length association would suggest these are parallel.

Again, in the interest of the length of this paper, we didn’t want to add even more analyses to this paper.

8. You don't account for the change in MT density as you move towards the spindle poles. For instance, the increased MT associations towards the pole can reflect simply this increased MT density. It might also be useful to ascertain the angular dependence of the MTs around the spindle pole (at 1.53 microns). Is it homogeneous as one moves from the pole–pole axis and behind the pole – I suspect not? Is there evidence of clusters (possibly use a statistical test here). It might also be helpful to illustrate the density of MTs. I calculated that MTs would be separated at the spindle pole sphere on average by 148nm (if homogeneous), whilst your tightest bundling of KMTs is 60nm at KTs. Distances are thus very similar at 0.7 microns, close to the peak of the minus end density.

As mentioned in our response to reviewer #1, we are now considering the density of the surrounding MTs in our analysis. We now present data that is normalized against the MT density.

Add more context. The results are not discussed in context of existing literature, including relating results toi) super–resolution studies, e.g. the twist and handedness of HeLa spindles (Tolic papers, twist appears weak or absent in spindles #1–3),

We have added more context to the observation of spindle tortuosity. We have added a section to the discussion.

ii) live cell imaging such as the EB tracking (Yamashita et al. 2015). The detected EB spots are far lower than the number of MTs detected here. Can this be explained? How many + ends (away from poles) have you in the spindle bulk, not associated with KTs.

Light microscopy analysis of EB1 typically only counts well defined, easily visualized “spots”. This will only include EB1 comets that remain in focus for extended periods of time and which are well resolved from other EB1 comments in the spindle. We believe that this is the primary reason that light microscopy misses so many of the MTs in the spindle. The fact that EB1 only associates with growing ends may also be a contributing factor.

iii) There is also no discussion of bridging fibres; this has been in the literature for a decade and a confirmation/quantification (number MTs) would be very welcome. This of course would entail examining non–KMTs, as suggested above.

On purpose, we avoided a statement on this issue. A possible bundling of non-KMTs similar to KMTs and thus visible as ‘bridging fibers’ (as proposed in the Tolic papers) is not obvious in our large-scale tomograms in metaphase. We just wanted to be careful here. The fact that we didn’t find such a bundling does not exclude such a pattern per se. We simply didn’t find evidence for it. BTW, we did a number of computational experiments to analyze patterns of non-KMTs with KMTs (in fact, we have two figures on this). We certainly need additional ways to analyze such bundling of non-KMTs before a strong statement can be made about this issue. However, the existence of bridging fibers should be much more obvious in anaphase spindles when these proposed fibers are proposed to associate specifically with k-fibers, thus promoting the segregation of chromosomes. This, however, is also on the list of future analyses on the ultrastructure of mitosis.

iv) You have 40% of minus ends away from the spindle poles. It would be good to have your interpretation of the processes that nucleate MTs, both nonKMT and KMT, away from the poles? Does this number make sense or is this evidence of poor MT tracking?

The issue raised above is certainly not related to the accuracy in MT tracing (see also our comments on the accuracy of MT segmentation and the manual correction of tracings). The correlation of MT nucleation and minus-end positioning is tricky as MTs can be moved within spindles. In the manuscript submitted in parallel (Conway et al.) we argue that the KMTs are nucleated at the kinetochore and that the growing MTs are guided towards the spindle poles by the surrounding non-KMTs. A more detailed analysis of the non-KMTs will be the subject of a future manuscript.

Data release.Tomography data: It is fantastic that the tomography data is released. I would however suggest that there is guidance on which free software can load the Amira files, or release in alternative formats.

We can certainly add additional information about our released data sets. We released all data sets in “.am“ Amira format. This file format allows to open such big files (also on older PCs). All tomographic data is also now available in “.tif“ format, which can be opened with either the ImagJ Fiji or the IMOD open-source software packages. However, these data sets are huge (50-80Gb each), therefore we cannot guarantee that everyone one will be able to open these files.

Tracks: it would be good if the MT traces were released as well. This would enable many people to analyse the networks without the high cost of analysing the tomography data themselves.

The MT-track files containing information about the segmented MTs were already released in binary and ASCII format, allowing anyone to read them. To make this task easier to potential users, the ASGA open-source software, which is part of this publication, is supplied with small scripts written in R language, which allows users to read the ASCII format into array (https://github.com/RRobert92/ASGA/blob/main/R/bin/Utility/Load_Amira.R).

[Editors' note: further revisions were suggested prior to acceptance, as described below.]

The manuscript has been greatly improved but there are some remaining points raised by the reviewers that the authors should consider before acceptance:Reviewer #1 (Recommendations for the authors):Kiewisz et al. have done a heroic job of revising their manuscript for resubmission. I appreciate the labor involved in recalculating all the images and interaction distances. Although the distances didn't change much, the images are a significant improvement, and the new data on near–neighbor MT–MT spacings are now more likely to stand the test of time. The revised neighbor distribution curves that are normalized for MT density are also a real improvement. All in all, this paper is now an excellent contribution to the literature and should be published as soon as possible. Below I mention a few issues that still strike my eye, but most of them are simple clarifications and so little work will be needed to make this manuscript a classic.

Thank you for your nice comments on our revised manuscript. We are happy that we could address all critical issues. We have addressed your additional points as follows:

CritiqueLn 1300 Figure 8 is potentially an important figure, but I find two problems with its current version: the gray MTs on a black background are essentially invisible, and the dots in panels C and E are unexplained. This may be my display, but the authors should consider the points.

Yes, we do agree with this statement. We addressed it in the modified new figure.

Figure 8 supplements. The lines representing MTs are so thin that they are hard to see on my screen, even when I blow each sub–image up to fill the screen. In your displays of whole spindles, it is clear why you must use narrow lines, but in these figures with only a few MTs displayed, you could increase the clarity of the images by just making each line a bit wider. Again, dots on the graphics are unexplained.

As stated above, we fixed this issue in the supplements to Figure 8.

In 8Sup1, all I am seeing is KMTs in different colors. There don't seem to be any non–KMTs, and there are no sites of interaction displayed. I gather that this figure shows simply the trajectories of the KMTs that do interact, not the interactions themselves. Do you think this is the best way to show your point?

As suggested, we have thickened the lines representing the MT trajectories. The whole purpose of this figure is to demonstrate the power of ASGA 3Dviewer software and to motivate the reader to display MT interactions by using this newly developed software package.

In 8Sup3, similar to Sup1. I think these figures would be more informative if you would show the interacting non–KMT and particularly the site of interaction. Simply coloring the whole MT erases that information.

Thank you for pointing this out. Again, the visualization as shown here was generated by using the ASGA_3DViewer. Again, this supplementary figure was added to encourage the reader to use the new software. Due to the interactive nature of this software tool, it should be easy to observe directly in the 3D models where MTs interact with each other. In the current version, the visualization of the interaction zones is not possible. However, this is a very good suggestion, and we will work on incorporating of interaction zones to have this feature in the next release of the software.

Ln 331 You are right, of course, that the major peak in Figure 8, Sup2 G shows a major peak near the pole, like all the other associations, but the secondary peak farther from the pole is very noticeable in this display, and given the evidence from other groups that this kinetochore–proximal region may be an important site for interactions between the ends of non–KMTs and the walls of KMTs, I think this really deserves mention.

We included this information in the revised version of our manuscript.

Figure 9, these MT–representing lines barely show up on my screen. A little thicker?

We have changed this according to your suggestion. Thank you for raising this point.

The legends for the Figure 9 supplement Figuresneed attention. The boxes are not referred to or explained, and again the sites of interaction are not clear. Supple. 2 legend text, "Number of KMTs plotted against the number of associations with other MTs in the spindle per individual KMT" doesn't match the labeling on those axes, so it's not clear what is being shown in these graphs.

Thank you for pointing this out. We have corrected this.

Ln 371 and following, you state that the number of associations was 10.6, but it is not clear what this number means. Number/KMT, per k–fiber or what?

We clarified this in the text. This number referred to an average number of individual interactions with KMT.

Ln 384 and following. I understand that the reworking of your images and numbers to accommodate the reviewers' suggestion of expanding the collapsed sections was a major amount of work, and probably pretty painful/annoying to do. Thus, adding this paragraph to the discussion is understandable, but as a reader coming to this version with a fresh point of view, I don't think it is necessary. You describe what you are going to do near the beginning of the paper, and you describe very nicely what you did in the methods section. It seems to me mentioning it again in the discussion over–emphasizes the point to, but this is for the authors to decide.

We removed this as suggested.

Ln 516 The statement that these and related data show that KMTs nucleate predominantly at kinetochores is true only for metaphase cells, and this limitation must be stated, otherwise it is misleading.

Thanks for pointing this out. We have added this to the manuscript.

The addition of https://cfci.shinyapps.io/ASGA_3DViewer/ as a viewer tool is a major step in data sharing. Many thanks to the authors for developing this, because the Amira package is expensive. This addition will make their hard work much more useful to others.

Thanks for this comment. This was exactly our intention: an AMIRA-independent software tool for viewing of our 3D models. The plan is to add more and more spindles to this visualization platform.

Reviewer #2 (Recommendations for the authors):The authors provide a substantially improved manuscript. We support publication. However, a number of issues remain and should be addressed before publication.

Thank you for supporting the publication of our manuscript. We have addressed your additional points as follows:

– Intro: The authors describe the three models of KMT–pole connections, but should situate known mammalian spindle structure work in the context of these models. More was known prior to this work than what was acknowledged.

As requested, we have added more references to each model of the KMT-pole connections.

– Figure 5B, 5S2B: legend and plot in 5B says centrosome interaction area was defined as a half–width of a peak from the center, but text lines 195 and 408 say "two fit half–widths from the center of the fit peak".

Thank you for pointing this out. We have corrected this.

– Figure 5D: How can the MT–centrosome interaction area be highlighted in gray, if this distance is 1.7 µm but the P–KT length of each k–fiber is different? (i.e. the x–axis of this plot is a relative measurement, but MT–centrosome interaction distance is absolute).

We apologize for being unclear here. As for the relative position of the MT-centrosome interaction area, this area indicates an approximated interaction zone on the pole-to-kinetochore axis. This is now addressed in the main text, in the Experimental Procedures and also stated in each figure legend.

– There is no Figure 5E.

Thank you for pointing this out. We have corrected this.

– Figure 5G: Discrepancy with Figure 5A and 5G legend, where P2 is at a relative position of 1.00 vs. 1.2.

Thank you for pointing this out. We have corrected this.

– Figure 4B, 4S1B, 5B, 5F, 5G, 9S2B, 9S2D: What criteria were used to decide which nKMTs to include? The total number from all three spindles is 16,256, based on Table 1, but these plots all include only 9957 (more than any individual spindle but far less than all three).

This discrepancy is related to the fact that some MTs (KMTs and non-KMTs) were not fully covered in the volume. For instance, if a single KMT from a k-fiber could not fully be reconstructed, we removed this whole fiber from the length distribution analysis. Similarly, non-KMTs not entirely reconstructed in the volume were marked as uncompleted and not considered in our analyses of length distribution, end distributions, curvature, etc. In the case of the MT-MT interaction analysis, this was indeed not correct. We apologize for this mistake. The text and the figure legends have been corrected to include now 16,256 non-KMTs.

– Figure 5S4B legend does not match plot.

Thank you. We have corrected the figure legend.

– Line 1254: What is meant by "A tortuosity of 1.1 is length."?

We deleted this phrase.

– Line 1255: What is meant by "The gray line indicates indicated by a dashed line."?

This line of text was rewritten in for a clarity.

– Figure 6E–F: Color codes are reversed. Red corresponds to higher tortuosity in E but lower tortuosity in F, and vice versa with gray.

We are sorry about this. We have corrected this both in the text and in the figure.

– Lines 231–236: A different motivation is needed for the tortuosity analysis. Tortuosity reports on microtubule curvature, but NOT twist–it doesn't tell us anything about 3D helicity that is consistent along a microtubule's length. It also seems that Figure 6 should have its own heading in the Results text, rather than being combined under the heading "K–fibers are broadened at spindle poles," since the correlation between tortuosity and spindle axis position is very weak and there is no other evidence in Figure 6 to support this statement.

Correct, this was confusing in our previous version. Therefore, we re-wrote the introduction to this paragraph. paragraph. Hopefully, our motivation for the tortuosity analysis is clearer now.

– Figure 7S1D: They did not address our comment that the polygon area analysis is never motivated or explained, and is unclear.

Thank you for raising this point. As suggested, Figure 6 now has its own heading.

– Figures 8–9: They did not change figures 8–9 (formerly 7–8) very much, as we suggested. Expanding their introduction to these figures in the text helps a little, but it's still hard to tell how these figures and sub–figures are making unique points.

As requested, we have changed (condensed) figures 8 and 9 and tried to better explain our motivation to perform the analyses on MT tortuosity. Hopefully, this is clearer now.

– Figure 9A: This cartoon should be moved to Figure 9S2, because interaction length is not quantified in Figures 9 or 9S1.

Cordially, we don’t agree with this specific point. We show this schematic drawing to simply illustrate how interactions in the spindles were detected, specifically how the number and length of interactions was measured. Although Figure 9 is focusing only on the number of interactions, we wanted to help the reader to understand our analysis. It is correct that Figure 9 sup2 then shows the quantification of the length of interactions. We think that this schematic drawing should remain to be shown in Figure 9.

– Figure 9S1: It's unclear what is being shown here or how it relates to Figure 9. Line 1375 says "This 3D model illustrates the association of KMT lattices with other KMT lattices minus ends". What are all the non–KMTs shown in these images? Are we looking at associations with lattices, with minus ends, or microtubules that interact for extended distances? Line 1376 says "The types of interactions are shown by color–coding," but what is the color code?

We apologize for this error. We fixed this and address the above comment in the figure legend. Following the suggestion from Reviewer #1, we also thickened the MT trajectories.

– Lines 497–498: What correlations is this sentence referring to? No correlations are shown in Figures 8 or 9.

We agree with this comment. This correlation refers to the correlation now shown in Figure 9—figure supplement 2C-D. We address this in the text.